

# Comparing calculated microphysical properties of tropical convective clouds at cloud base with measurements during the ACRIDICON-CHUVA campaign

*Ramon Campos Braga[1], Daniel Rosenfeld[2], Ralf Weigel[3], Tina Jurkat[4], Meinrat O. Andreae[5,9], Manfred Wendisch[6], Mira L. Pöhlker[5], Thomas Klimach[5], Ulrich Pöschl[5], Christopher Pöhlker[5], Christiane Voigt[3,4], Christoph Mahnke[3], Stephan Borrmann[3], Rachel I. Albrecht[7], Sergej Molleker[8], Daniel A. Vila[1], Luiz A. T. Machado[1], and Paulo Artaxo[10]*

[1]Centro de Previsão de Tempo e Estudos Climáticos, Instituto Nacional de Pesquisas Espaciais, Cachoeira Paulista, Brasil

[2]Institute of Earth Sciences, The Hebrew University of Jerusalem, Israel

[3]Institut für Physik der Atmosphäre, Johannes Gutenberg-Universität, Mainz, Germany

[4]Institut für Physik der Atmosphäre, Deutsches Zentrum für Luft- und Raumfahrt (DLR),

Oberpfaffenhofen, Germany

[5]Multiphase Chemistry and Biogeochemistry Departments, Max Planck Institute for Chemistry, 55020 Mainz, Germany.

[6]Leipziger Institut für Meteorologie (LIM), Universität Leipzig, Stephanstr. 3, 04103 Leipzig, Deutschland

[7]Instituto de Astronomia, Geofísica e Ciências Atmosféricas, Universidade de São Paulo, São Paulo, Brazil

[8]Max Planck Institute for Chemistry (MPI), Particle Chemistry Department, Mainz, Germany

[9]Scripps Institution of Oceanography, University of California San Diego, La Jolla, California 92037, USA

[10]Instituto de Física (IF), Universidade de São Paulo (USP), São Paulo, Brazil

*Correspondence to*: Ramon C. Braga (ramonbraga87@gmail.com)

**Abstract:** Reliable aircraft measurements of cloud microphysical properties are essential for understanding liquid

convective cloud formation. In September 2014, the properties of convective clouds were measured with a Cloud Combination Probe (CCP), a Cloud and Aerosol Spectrometer (CAS-DPOL), and a cloud condensation nuclei (CCN) counter on board the HALO (High Altitude and Long Range Research Aircraft) aircraft during the ACRIDICON-CHUVA campaign over the Amazon region. An intercomparison of the cloud drop size distributions (DSDs) and the cloud water content derived from the different instruments generally shows good agreement within

the instrumental uncertainties. The objective of this study is to validate several parameterizations for liquid cloud formation in tropical convection. To this end the directly measured cloud drop concentrations ($N_d$) near cloud base were compared with inferred values based on the measured cloud base updraft velocity ($W_b$) and cloud condensation nuclei (CCN) vs. supersaturation ($S$) spectra. The measurements of $N_d$ at cloud base were also compared with drop





concentrations ($N_a$) derived on the basis of an adiabatic assumption and obtained from the vertical evolution of cloud

drop effective radius ($r_e$) above cloud base. The results demonstrate agreement of the measured and theoretically

expected values of $N_d$ based on CCN, $S$, $W_b$ at cloud base, and the height profile of $r_e$. The measurements of $N_{CCN}(S)$

and $W_b$ did reproduce the observed $N_d$. Furthermore, the vertical evolution of $r_e$ with height reproduced the observation-based nearly adiabatic cloud base drop concentrations, $N_a$. Achieving such good agreement is possible only with

accurate measurements of DSDs. This agreement supports the validity of the applied parameterizations for continen-

tal convective cloud evolution, which now can be used more confidently in simulations and satellite retrievals.

## 1. Introduction

The understanding of cloud formation and its influence on the global hydrological cycle and radiation budget is

fundamental for improving weather and climate forecasting models (Ten Hoeve et al., 2011; Jiang and Feingold,

2006; Kohler, 1999; Rosenfeld et al., 2008; Stephens, 1984). Cloud microphysical models pursue to reproduce

atmospheric processes based on physical relationships developed from field experiments and remote sensing

observations in different parts of the globe (Silva Dias et al. 2002; Machado et al. 2014; Fan et al. 2014; Rosenfeld

et al. 2014b). Data from aircraft probes provide opportunities to validate and improve cloud models.

An assessment of the validity of data from cloud probes is essential before the results can be implemented into cloud

models. According to previous studies, the number concentration of cloud droplets ($N_d$) expected at cloud base

mainly depends on atmospheric conditions just below cloud base, i.e., updraft wind speed and the supersaturation

($S$) activation spectra of cloud condensation nuclei [$N_{CCN}(S)$] (Pinsky et al., 2012; Reutter et al., 2009; Twomey,

1959). From cloud condensation nuclei counter (CCNC) measurements across a range of supersaturations ($S$), the

parameters $N_0$ and $k$ are estimated from Twomey's formula (Twomey, 1959):

$$N_{CCN} = N_0 \cdot S^k \qquad (1)$$

where $N_0$ is the cloud condensation nuclei (CCN) concentration at $S=1\%$ in cm$^{-3}$, and $k$ is the slope parameter

(Twomey, 1959). Equation 1 is an analytical representation of the observational data within the measured range of $S$,

which in our case represents the observed CCN spectrum from 0.2 to 0.55 %. Note, however, that Eq. 1 does not

allow a reliable extrapolation of $N_{CCN}(S)$ beyond this range (Pöhlker et al., 2016).

The parameters $N_0$ and $k$ are estimated from data measured below cloud base along with updrafts wind speed

measurements at cloud base ($W_b$). The values of $W_b$, $N_0$, and $k$ are used for calculating the theoretical cloud droplet

concentration from Eq. 2 (Twomey, 1959) below:

$$N_{dT} = 0.88 \cdot N_0^{\frac{2}{k+2}} \cdot \left(0.07 \cdot W_b^{1.5}\right)^{\frac{k}{k+2}} \qquad (2)$$

where $N_{dT}$ are the estimated cloud base drop concentrations in cm$^{-3}$. Here we compare the measured $N_d$ to $N_{dT}$ by

substituting in Eq. 2 the measured $N_{CCN}(S)$ in the form of $N_0$ and $k$, along with the measured $W_b$.

Equations 1 and 2 are a rather simplistic parameterization. More advanced methods, using the hygroscopicity

parameter $\kappa$ (kappa) are more accurate to represent the CCN spectrum (Pöhlker et al., 2016). However, in this study,





using Twomey´s parameterization is advantageous, because the CCN measurements were performed within the range of 0.2-0.55 %, where the estimation of the $N_0$ and $k$ parameters using Eq. 1 does not incur significant errors in comparison with more advanced methods (Pöhlker et al., 2016). Furthermore, Twomey's parameterization also allows calculating the effects of updraft wind speed on $N_{dT}$ as a function of $N_0$ and $k$.

Another approach to estimate the number concentration of CCN that are expected to nucleate as droplets at cloud base is through the use of the $k$-Köhler model (Petters and Kreidenweis, 2007). Based on a given dry aerosol particle size distribution (ASD), the $k$-Köhler model with prescribed $W_b$ simulates the expansion and cooling of air as well as the resulting changes in relative humidity and the related hygroscopic growth of aerosol particles and further condensational growth of cloud droplets. The input to this approach depends strongly on the measured ASD and $\kappa$ (Reutter et al., 2009).

Measurements of ASD by a Passive Cavity Aerosol Spectrometer Probe (PCASP) and Ultra-High-Sensitivity Aerosol Spectrometer (UHSAS) probes were available during the ACRIDICON (Aerosol, Cloud, Precipitation, and Radiation Interactions and Dynamics of Convective Cloud Systems) - CHUVA (Cloud processes of tHe main precipitation systems in Brazil: A contribUtion to cloud resolVing modeling and to the GPM [Global Precipitation Measurements]) campaign (Wendisch et al., 2016). However, calculating $\kappa$ from the combined CCN, PCASP, and UHSAS measurements below cloud resulted in unreasonably low $\kappa$ values (not shown), which could only be explained by hygroscopic swelling of the aerosols at ambient humidity by a large factor of up to more than two. A possible reason for this behavior in measurements over the Amazon is that the effective hygroscopicity parameters describing water uptake at sub-saturated conditions can be substantially lower than at supersaturated conditions (Mikhailov et al., 2013). The analysis of this effect on the ASD measurements from PCASP and UHSAS below cloud base requires considerable efforts, which are beyond the scope of this paper. Also, in the case of our flight missions, a major obstacle to the use of the use of the $k$-Köhler approach is the fact that measuring the CCN(S) spectrum requires a much longer time than the aerosol spectrum, thus the two measurements are not representing the same aerosol sample. This was evident from the variability of the CCN concentrations measured at fixed $S$ with one CCNC column, while measuring the CCN(S) spectrum with the other column during the flights. The lack of these important analyses prevents the use of $k$-Köhler model estimates for comparison with $N_d$ measurements from cloud probes in the present study.

An estimation of the cloud base droplet concentrations is also possible via the calculation of the maximum supersaturation ($S_{max}$) at cloud base, relying on the measured $N_d$ and $W_b$ according to Eq. 3 (Pinsky et al. 2012) below:

$$S_{max} = C \cdot W_b^{\frac{3}{4}} \cdot N_d^{-\frac{1}{2}} \qquad (3)$$

where $C$ is a coefficient that is determined by cloud base temperature and pressure. Since the combination of $N_{CCN}(S)$ and $W_b$ determines $N_d$ and $S_{max}$, it is possible to compare the measured and theoretical relationships. Additionally, the estimation of adiabatic cloud droplet concentrations ($N_a$) from measurements of the vertical profile of cloud drop effective radius ($r_e$) is another alternative to evaluate the number of droplets nucleated at cloud base (Freud et al., 2011). The definition of $r_e$ is:





$$r_e = \frac{\int N(r) \cdot r^3 \, dr}{\int N(r) \cdot r^2 \, dr} \qquad (4)$$

where N and r are the droplet concentrations and radii, respectively.

Rosenfeld et al. (2014a) have shown that the effective number concentration of droplets at cloud base ($N_d{}^*$) can be expressed by a single number, which depends on the effective updraft speed at cloud base ($W_b{}^*$). To evaluate whether the measured $N_d{}^*$ represents the theoretically expected $N_d{}^*$ based on the independent measurements of $N_{CCN}(S)$ and $W_b$, it is necessary to find the range of measured $W_b{}^*$ and $N_d{}^*$ that fulfills best the closure between the

measured and indirectly calculated values. Cloud models represent the number of droplets at cloud base by a single number (Pinsky et al., 2012). Therefore, from a set of $N_d$ measurements at cloud base, an 'effective' number of droplets, $N_d{}^*$, can be derived, which represents the measurements for a set of clouds formed in the same thermodynamic condition.

The droplet size distribution (DSD) spectrum from clouds, i.e. the DSD variability, depends on the stage of cloud

development. After nucleation, the cloud droplets in rising cloud parcels grow with height mainly by condensation. Raindrops start forming when $r_e$ reaches 13-14 µm and coalescence becomes efficient (Freud and Rosenfeld, 2012; Rosenfeld and Gutman, 1994). Accurate documentation of the vertical evolution of cloud and rain DSDs is essential for analyzing these types of microphysical processes within clouds. Assessing the quality of DSD measurements by the aircraft probes is thus a necessary task. This assessment can be achieved via comparisons between the cloud

water content (CWC) calculated from cloud probe DSDs and the direct measurements of CWC with a hot-wire device (CWC$h$) for cloud penetrations at different heights (Freud et al., 2008; Rosenfeld et al., 2006). This is done in section 3 while accounting for the dependence of the measurement efficiency of the hot-wire on drop size.

Three cloud probes measured the DSDs on board the HALO aircraft during the ACRIDICON-CHUVA campaign (Wendisch et al., 2016). In addition, CWC was measured by a King hot-wire probe (King et al. 1978) installed in the

Cloud and Aerosol Spectrometer (CAS-DPOL) probe.

Figure 1 illustrates the HALO flight patterns in convective cloud clusters performed in three steps:

    a.    Flying below cloud base for measuring $N_{CCN}(S)$;

    b.    Flying through cloud base for measuring $W_b$ and DSD;

    c.    Conducting vertical profiles in growing convective towers close to their tops, to avoid precipitation that

130          may fall from above. The cloud penetrations during this phase are made in vertical steps of several hundred
          meters when possible, from cloud base to the anvils.

The availability of these measurements collected by the same aircraft provides a unique opportunity to compare the data with model predictions and to test the sensitivity of the results to the differences between the measurements by the cloud probes. Particularly, the validation of physical parameterizations for cloud base of convective clouds over

the Amazon basin are on focus in this study. This is the first study that tests the parameterizations against each other with the same data set.  In addition, the distributions of the updrafts and $N_d$ at cloud base and their relationships in the context of convective clouds parameterization is account for in this study.

Different approaches are illustrated in the next sections. Section 2 discusses the instrumentation and database used



for this study. Section 3 gives an overview on the cloud probe measurements and discusses consistencies and disagreements between the measurements. Section 4 describes the methodologies applied to compare measurements and model results at cloud base.

## 2. Instrumentation

The HALO flights during the ACRIDICON-CHUVA campaign were performed over the Amazon region during
September 2014 under different conditions of aerosol concentration and land cover, as shown in Fig. 2 (from Wendisch et al., 2016). This region was chosen for documenting cloud microstructure and precipitation-forming processes during the dry season with high concentrations of CCN, and to contrast these measurements against cleaner conditions that could be found within flight range, as documented previously (Andreae et al., 2004; Artaxo et al., 2002). Additionally, we made use of the fact that Manaus city is located in the central Amazon (3.11 ºS; 60.02
ºW), and that therefore the aerosol perturbation from the Manaus urban plume may increase CCN concentrations by one to two orders of magnitude above the pristine conditions in the background air (Kuhn et al., 2010). This study is done in collaboration with the Green Ocean Amazon experiment – GoAmazon (Martin et al., 2016), which also addressed the aerosol influences on cloud microphysical properties, with special focus on the Manaus urban plume. A comprehensive introduction to airborne instrumentation is given by Wendisch and Brenguier (2013), and in
particular of the microphysical instruments involved in this study by Brenguier et al. (2013).

### 2.1 Cloud condensation nuclei (CCN) measurements

CCN number concentrations were measured on board HALO during ACRIDICON-CHUVA using a two-column CCNC (CCN-200, Column A and B), a continuous-flow longitudinal-thermal-gradient instrument manufactured by
Droplet Measurement Technologies (DMT) (Roberts and Nenes, 2005). It measures the CCN number concentration as a function of water vapor supersaturation ($S$) at a time resolution of 1 Hz. In the instrument, the sampled aerosol particles are exposed to a set supersaturation, and adsorb water depending on their size and chemical composition. Those particles that grow to droplets larger than 1 μm in diameter are counted as CCN at that $S$. The instrument was calibrated between flights following Rose et al. (2008).

Sample air for the aerosol measurements was obtained from two different inlets: (i) the HALO aerosol submicron inlet (HASI), and (ii) the HALO counterflow virtual impactor (HALO-CVI) (Wendisch et al., 2016). The CCN-200 provides the possibility to measure in parallel from both inlets or at two different values of $S$. In this study, only the aerosol measurements from the HASI inlet have been used. The measurements were done with one column at a constant S=0.55 %, while the other was cycling $S$ between 0.2 and 0.55 % with steps every 100 seconds.


### 2.2. Cloud probe measurements

Three cloud probes were operated on board HALO during the measurements in the ACRIDICON-CHUVA campaign. This study focuses on the CAS-DPOL and CCP-CDP probes. The third probe, NIXE-CAS-DPOL was of identical type as CAS-DPOL and is thus not used in this study. The probes' range of measurements is shown in





Table 1. In this study, cloud particle concentrations are counted at diameters larger than 3 µm to avoid measurements of haze droplets. This is also in accordance with the similar lower limits of the bins sizes of the CCP-CDP. Details about the cloud probe measurement characteristics are described in the following sections (see also Brenguier et al., 2013).

*2.2.1 CCP-CDP and CCP-CIP measurements*
The Cloud Combination Probe (CCP) combines two detectors, the Cloud Droplet Probe (CDP) and the greyscale Cloud Imaging Probe (CIPgs). The CDP detects forward scattered laser light from cloud particles as they pass through the CDP detection area (Lance et al., 2010), and represents an advanced version of the Forward Scattering Spectrometer Probe (FSSP) (Baumgardner et al., 1985; Dye and Baumgardner, 1984; Korolev et al., 1985; Wendisch
et al., 1996). The CIPgs records 2-D shadow-cast images of cloud elements that cross the CIPgs detection region. The overall particle detection size range is 2 to 960 µm when measuring with the CCP. The highest temporal resolution of the CCP measurements is limited to 1Hz. Recent findings concerning the measurement uncertainties of the underwing cloud probes at the comparatively high HALO flight velocities (well above 170 m s$^{-1}$) provide correction procedures to be applied to the measured raw data to further improve the data quality of the ambient
cloud particle number concentrations (Weigel et al., 2016). The robust performance of the specific CCP instrument used in this study, even under extreme conditions, was demonstrated by earlier investigations in tropical convective outflow (Frey et al., 2011), Polar Stratospheric Clouds (PSC) (Molleker et al., 2014), and low-level mixed-phase clouds in the Arctic (Klingebiel et al., 2015). For the CDP sample area of 0.22 mm² was used additionally considering an uncertainty of about 10 % (Molleker et al., 2014). The sizing accuracy of the CDP is estimated to be
about 10 % for spherical particles and correctly assumed refractive indices.

        *2.2.2 CAS-DPOL measurements*
The CAS-DPOL measures particle size distributions between 0.5 and 50 µm at 1Hz time resolution (Baumgardner et al., 2001). Its measurement principle is developed based on the FSSP-300 (Baumgardner et al., 1985, Korolev et al.,
1985), which has been used previously to study the particle size range in ice clouds (Voigt et al., 2010, 2011; Schumann et al., 2011; Jeßberger et al., 2013). The intensity of forward scattered light in the angular range of 4 – 12 ° is detected and sorted into 30 size bins. Assuming Mie scattering theory, additional binning into 15 size bins is employed to rule out ambiguities. Polarized backward scattered light is detected to investigate the sphericity and phase of the particles (Baumgardner et al., 2005; Gayet et al., 2012; Järvinen et al., 2016). Number concentrations
are derived using the probe air speed measured by the probe.).. The distribution of time intervals between single particles, recorded for the first 290 particles in each second, did not provide indications of droplet coincidence up to a time resolution of 0.8µs or a number concentration of 2200 cm$^{-3}$. After the campaign, the sampling area (SA) which is used to derive the number concentration of particles was characterized by a high-resolution scan with a droplet generator. 250 water droplets of a known, quasi constant size of about 40 µm were dropped at and around
the sensitive region perpendicular to the laser beam. The resolution of the droplet generator scan was 25 µm perpendicular to the laser beam and 50 µm along the laser beam. According to the scan, the area of the measured SA



for particle diameters above 3 µm was 0.27 mm², which is 8% higher than the initially reported SA by the manufacturer. The fringe of the area, a region where particles are counted but with low efficiency was about 0.032 mm² which represents an uncertainty of 15% of the total SA. Additionally, we estimate an uncertainty of the particle velocity in the CAS sampling tube of 15%, taking into account that particle velocities in the sampling tube may be slowed down or accelerated compared to open path instruments or the Pitot tube velocities at the CAS. This results in a combined uncertainty of the number concentration of 21%.

Calibrations with glass beads of four different sizes (2, 5, 20 and 42 µm) were performed between the flights to monitor the stability of the size bin classification. Difference in the refractive index can be accounted for using the method of e.g. (Rosenberg et al. 2012). The size calibration was stable over the whole campaign. For the purpose of this study mainly the effective diameter range between 10 and 26 µm was evaluated, which employed mainly the lowest amplifier gain stage. For particles up to 20 µm the size calibration did not show any size deviations from the expected values. Larger particles with diameters > 40 µm were shifted towards lower sizes by about 5 µm. We therefore estimate an uncertainty in particle size for particles diameters above 40 um on the order of 13 to 15 % and less below. The instrument had been installed previously on HALO and the DLR Falcon aircraft during the ML-CIRRUS (Voigt et al., 2016), ACCESS-II, ECLIF, and DACCIWA campaigns.

### *2.3 Hot-wire CWC measurements*

The hot-wire instrument is a King Probe type device that measures the bulk liquid water content (LWC) from 0.01 to 3 g m⁻³ in the droplet diameter range of 5 to 50 µm by detecting the power (current) required to maintain a heated wire at a constant temperature of 125 °C. The sensitivity of the instrument is reduced for droplets below 10 µm, since smaller particles follow more closely the streamlines around the hot-wire. The instrument was mounted on the CAS-DPOL probe. The accuracy of the King Probe LWC measurement is estimated to be 5 % at 1 g m⁻³ and decreases down to 16 % at 0.2 g m⁻³, with a sensitivity of 0.02 g m⁻³ (King et al., 1978). For this study, mainly CWC in the range up to 1 g m⁻³ was used.

### *2.4 Vertical wind speed measurements*

The HALO aircraft was equipped with a new meteorological sensor system (BAsic HALO Measurement And Sensor System - BAHAMAS) located at the nose of the aircraft (Wendisch et al., 2016). Measurements of updraft speeds during cloud base penetrations during the ACRIDICON-CHUVA campaign have shown maximum vertical wind speeds in the range of 5 m s⁻¹. In these conditions, the uncertainties of $W$ measurements are less than 0.2 m s⁻¹ (Mallaun et al., 2015). For a long sequence of measurements at cloud base (> 20 s) these uncertainties become negligible.

### *3. Cloud probe intercomparison*

### *3.1 Method*

The validation of convective clouds parameterization requires reliable cloud probe measurements. In this section, we discuss quantitatively the difference in estimated and directly measured CWC and DSDs of the two cloud probes





CAS-DPOL and CCP-CDP as well as the hot-wire instrument.

For comparisons between the CWC estimated from the cloud probe DSDs and hot-wire measurements (CWC$h$), we distinguish between spectra that are dominated by condensational growth, and spectra where coalescence becomes important, too. These spectra are separated by the threshold of $r_e$ for significant coalescence, which varies as a function of the drizzle water content (DWC) for 1 second cloud passes (Freud and Rosenfeld, 2012). In addition,

droplets with diameters < 10 μm are not captured efficiently by the hot-wire probe, resulting in an underestimation of CWC$h$. The hot-wire device was installed on the CAS-DPOL probe; therefore a better statistical agreement is expected for this probe in comparison with the CCP-CDP. The CCP-CDP was mounted on the other wing, about 15 m away from the hot-wire device (Voigt et al., 2016; Wendisch et al., 2016). Only cloud passes at temperatures greater than 0 ºC are considered in this intercomparison, to avoid uncertainties of the measurement due to freezing

of droplets.

*3.2 CWC comparison between cloud probe and hot wire measurements*

Comparison of different techniques of cloud water content measurements are challenging regarding the individual instrumental differences, like time resolution, dependence of sensitivity on size and with respect to their target of

interest e.g. inhomogeneous, turbulent convective cloud.

For this study we use the hot-wire instrument as a reference to the scattering spectrometer probes since its total water content is derived from a smaller set of physical parameters with an overall uncertainty of maximal 16% as compared to ~ 30% uncertainty when derived from DSDs.

The calculation of CWC is performed separately with CAS-DPOL and CCP-CDP probes droplet concentrations as

follows:

$$CWC = \frac{4\pi}{3}\rho \int N(r)r^3 dr \qquad (5)$$

where $N$ is the droplet concentration in m$^{-3}$, $r$ the droplet radius in m and $\rho$ is the water density (1 g cm$^{-3}$). The calculation of DWC is done similar to CWC but with different cloud probe and particle size ranges. The DSDs from CCP-CDP and CAS-DPOL are used to calculate the CWC, defined here as the mass of the drops integrated over the

diameter range of 3–50 μm. Similarly, DSDs from CCP-CIP are used to calculate the DWC, defined here as the mass of the drops integrated over the diameter range of 75–250 μm (Freud and Rosenfeld, 2012).

Figure 3 shows the dependency of calculated $r_e$ as a function of altitude for cloud passes during flights over different conditions of aerosol concentrations (AC13- very polluted, AC18- polluted and AC19 – clean). The probability of rain due to collision and coalescence processes are indicated with dashed lines. It is assumed that rain formation

starts when calculated DWC exceeds 0.01 g m$^{-3}$ (Freud and Rosenfeld, 2012). Overall, the figure shows that $r_e$ values increases with altitude. In addition, it shows the effects of aerosol loading, which in higher concentration nucleate a larger number of droplets at cloud base that grows slower as a function of height via condensation. Also, for $r_e$ values < 9 μm the probability of coalescence of droplets is very small and starts to be significant for $r_e$ > 11 μm. There is little concern that raindrops precipitate from above when flying near the tops of growing convective

clouds (as illustrated at Fig. 1).

The comparison of CWC estimated from the cloud probe data and CWC$h$ measured with the hot-wire was



performed as a function of $r_e$, because the measurement efficiency of the hot-wire probe depends on drop size. This type of analysis also provides information about the differences between the two cloud probes regarding the estimated CWCs. Strapp et al. (2003) show that large differences between actual CWC and hot-wire measurements

occur when larger drops ($\sim r > 20$ µm) contribute to the cloud water content above 1 g m$^{-3}$. We therefore limit our analysis to the effective diameter range of 5 µm $< r_e <$ 13 µm and compare CWC$h$ with CWC estimated from the cloud probe DSD only for CWC up to 1 g m$^{-3}$.

The comparison between the mean CWC estimated from the cloud probe DSDs and mean CWC$h$ are shown as a function of $r_e$ in Fig. 4. The ratio between the CWC$h$ from the hot-wire measurements and the probe estimates

(CWCr) is also shown (in red color). As the hot-wire has reduced sensitivity for particles with $r_e <$ 5 µm, the analysis is performed only for $r_e >$ 5 µm.

The mean values of CWC estimated from the probes from flights AC08 and AC20 and altitudes between 600 m and 5,000 m generally show an increase with increasing $r_e$. The CWC uncertainty calculated with CAS-DPOL (CCP-CDP) DSDs is about 22% (10 %) for all measurements. In addition, the uncertainty associated with $r_e$ calculations

with CAS-DPOL (CCP-CDP) DSDs is about 14 % (9 %). Within their statistical variability, the CAS-DPOL CWC agrees well with the hot-wire CWC$h$ over the whole effective radius range (upper panel). The CWCr for CAS-DPOL (CCP-CDP) is around 1 ± 0.1 (0.8 ± 0.05) for almost all $r_e$ sizes. The comparisons of the CWC$h$ with the CWC estimated from the CCP-CDP probe (lower panel) shows that CCP-CDP is systematically higher by about 21%. The difference is larger than the standard deviation of the individual measurements. The overall systematic

difference (mean of the ratio) in the cloud probe CWC in comparison to CWC$h$ are 0.04 g m$^{-3}$ (6% in percentage) for CAS-DPOL and 0.11 g m$^{-3}$ (21% in percentage) for CCP-CDP higher than the hot-wire measurements. However, considering the uncertainty of the measurements, all three CWC measurements agree within the uncertainty range (16% and 30%).

In summary, the CWC$h$ from the hot-wire agrees better with the CWC derived from CAS-DPOL DSDs. The fact

that CCP-CDP was mounted on the opposite wing, while the measurements were performed in very inhomogeneous conditions may account for some of the larger spread between the two instruments (e.g. through the choice of $r_e$) compared to the CAS-DPOL - hot-wire comparison but cannot explain the systematic offset of the CCP-CDP. In the next sub-section we discuss input parameters for the CWC estimated from the cloud probes like number concentration and size to find an explanation for the observed differences.


### 3.3 Comparing cloud probe Nd and DSDs

Figure 5 shows the mean $N_d$ values measured by CAS-DPOL and CCP-CDP (solid line) and the systematic uncertainties of measurements (dashed lines) as a function of $r_e$ for values greater than 5 µm (left panel) and the standard deviation of the two cloud probe $N_d$ measurements (right panel). The data is the same used with hot-wire

intercomparison. Both probes measure a decrease in number concentration with effective radius, related to coagulation processes in the cloud. Taking into account the increase in CWC with $r_e$, a reduced number of larger droplets contribute to the enhanced CWC at larger $r_e$. In general, CAS-DPOL mean $N_d$ agree well (difference lower than 1 %) with mean $N_d$ as CCP-CDP for effective radii between 7 and 11 µm. Statistical significant differences are





observed for $r_e$ smaller than 7 µm and above 11 µm. Both probes have similar standard deviation (STDEV) for different $r_e$ sizes. The STDEV decreases with increasing $r_e$, varying from ~20 cm$^{-3}$ to ~10 cm$^{-3}$.

The two $N_d$ measurements agree within the combined statistical variability and the systematic uncertainties of the two probe measurements (21% for CAS-DPOL and 10% for CCP-CDP). However, in order to explain the difference in CWC, we point towards the difference of mean droplet number at $r_e > 11$µm. Lower number concentrations of the CAS-DPOL at larger $r_e$ may be related to the shift in droplet radii for particles above 40 µm to smaller sizes that shift the effective radius and the CWC to smaller $r_e$ and smaller CWC. On the other hand, the difference in the size binning of the two probes may artificially shift particles from higher sizes to lower sizes just by the choice of the bin boundaries. For the CAS-DPOL, larger bin sizes were chosen in order to avoid ambiguities based on Mie-Lorenz theory.

The differences in $N_d$ at larger $r_e$ correspond to the enhanced CWC in Fig. 4 and may explain most of the differences in CWC between the probes. The higher number concentration at $r_e < 7$ µm may be explained by the higher sensitivity of the CAS-DPOL at smaller sizes. The instrument was built to particularly measure the full spectrum of aerosol and cloud particles in the size range where aerosols are activated into cloud droplets.

Figure 6a-d shows the mean droplet concentration and CWC as a function of droplet diameter from the cloud probes. The distributions are shown for four different effective radii to give an impression on the evolution of particle size and CWC with altitude for the two cloud probes. For $r_e$ between 5 and 6 µm and 8 and 9 µm (Figures 6a-b), where collision and coalescence processes are negligible (see Fig. 3), CCP-CDP DSDs are somewhat below the CAS-DPOL DSDs, revealing an enhanced sensitivity of the CAS-DPOL for smaller particles. For larger $r_e$ (Figures 6c-d), where coalescence starts and raindrops may be present, CCP-CDP shows slightly larger droplet concentrations and CWC for diameters > 15 µm in comparison to CAS-DPOL. This may be related to larger droplets that enter the open path instrument sampling area of the CCP-CDP easier than the closed path sampling area of the CAS-DPOL by falling vertically into the measurement area.

These results suggest that CAS-DPOL and CCP-CDP generally measure similar droplets concentrations in the size range between 3-50 µm. The observed deviations between the probes could be caused by different inlet configurations or measurement principles of the two probes, each with individual advantages depending on the measurement target and related size range. However, the differences in DSDs are within the uncertainties of the measurement and show a much better agreement as compared to earlier measurements at similar conditions (Lance et al., 2012; Rosenberg et al.,2012).

## 4. Methodology

The following analysis is performed in four steps. Section 4.1 presents the analyses of CCN measurements below cloud base. Assuming the relation between $N_{CCN}$ and $S$ is given by Eq. 1, the parameters $N_0$ and the slope $k$ are calculated from the measurements below cloud base. Section 4.2 describes the estimation of maximum $S$ at cloud base ($S_{max}$) based on the measured $N_d$ and $W_b$ there. The co-variability of $N_d$ and $W_b$ is used to estimate the CCN concentration ($N_{dCCN}$) by calculating $S_{max}$ according to Eq. 1. This is repeated for the two $N_d$ spectra that were obtained from the two cloud droplet probes. In addition, $N_d$ is estimated by application of the measured $W_b$ spectrum





to Eq. 2 and comparing against the directly measured $N_d$ from the two cloud probes. Section 4.3 outlines the methodology of calculating the effective number of droplets at cloud base from cloud probe measurements ($N_d$*). This is done using theoretical considerations based on the estimated values of $N_{dT}$ and $N_{dCCN}$ at cloud base ($N_{dT}$* and $N_{dCCN}$*, respectively). The exact definitions of all parameters are provided in Section 4.2. Section 4.4 explains the

calculation of the estimated adiabatic cloud droplet concentration ($N_a$), as obtained from the measured vertical profile of cloud drop size distributions.

### 4.1 CCN measurements below cloud base as a function of S

The measurements of $N_{CCN}$ and $S$ can be parameterized by Eq. 1 and provide $N_0$ and $k$ (Pruppacher et al., 1998). The

typical values of $N_0$ are about 100 cm$^{-3}$ for pristine conditions, and range from 500 cm$^{-3}$ to several thousand cm$^{-3}$ for polluted continental regions at different levels of aerosol loading. The values of the slope parameter $k$ vary from about 0.3 to 1 in clean and polluted air, respectively (Andreae, 2009).

Two types of CCN measurements were performed: (i) measuring CCN concentration at fixed $S$ (~0.55%) [hereafter referred to $S_1$ with the corresponding CCN concentration referred as $CCN_1$] and (ii) measuring CCN concentration at

variable $S$ (ranging from 0.2 % to 0.55 %) [hereafter referred to $S_2$ with the corresponding CCN concentration referred as $CCN_2$]. Since the $CCN_2$ measurements were performed at varying $S_2$ (generally modified every 100 seconds during the flights; hereafter referred as time step), the mean values of these measurements for each time step are used to calculate the $N_0$ and $k$ parameters in Eq. 1. The flight period of measurements below cloud base in a specific region consisted of several CCN time steps and covered at least one full $N_{CCN}(S)$ spectrum, and is defined as

a group of measurements (hereafter referred as a group).

To achieve accurate measurements of $CCN_2$ as a function of $S_2$, a weighting factor calculated from the $CCN_1$ measurements is applied, as specified in the steps below. Because $CCN_1$ measures at a fixed supersaturation ($S_1$), its variability is caused only by changes of total CCN concentration (from aerosol loading) along the flight track (assuming constant size distribution and composition during the measurement group). This is used to correct the

$N_{CCN}(S)$ as measured by $CCN_2$ for these changes of total concentration. The procedure for this analysis is:

1. The mean values of $S_1$, $S_2$, $CCN_1$ and $CCN_2$ measurements ($mS_1$, $mS_2$, $mCCN_1$ and $mCCN_2$, respectively) are calculated for each time step below cloud base;

2. A factor of aerosol loading ($FA$) for measurements during a full cycle of $S$ is calculated as follows:

$$FA = \frac{mCCN_1}{TmCCN_1}$$

where $TmCCN_1$ is the mean of all $CCN_1$ measurements for the group of $S$ cycling. $FA$ provides the

deviation of aerosol concentration from the mean for a specific time step in the group;

3. The $mCCN_2$ values for each group are weighted by $FA$ generating normalized $mCCN_2$ values ($NCCN_2 = mCCN_2 / FA$). Then, the $NCCN_2$ are used in combination with $mS_2$ to fit a power-law-function equation for each group of measurements. From this fit, the values of the parameters $N_0$ and $k$ in the Twomey equation ($N_{CCN}=N_0 \cdot S^k$) are obtained.






### 4.2. Estimating $S_{max}$, $N_{dCCN}$, and $N_{dT}$

The number of CCN that nucleate into cloud droplets ($N_d$) reaches its maximum value near the $S_{max}$ height in the cloud (Pinsky et al., 2012). This level is observed between cloud base and a height up to a few tens of meters above

it. The value of $S_{max}$ depends on the vertical velocity at cloud base and on $N_{CCN}(S)$. Therefore, $N_d$ can be used to achieve a closure for $N_{dCCN}$ estimates. $N_d$ is measured with the cloud probes CCP-CDP and CAS-DPOL (*Ncdp* and *Ncas*, respectively). The $S_{max}$ at cloud base was then estimated from $N_d$ and $W_b$ measurements from Eq. 3.

The $N_0$ and $k$ values that were calculated from measurements below cloud base (as described in Section 4.1) are substituted in Eq. 1 and Eq. 2 for calculating $N_{dCCN}$ and $N_{dT}$, respectively.

The comparisons between $N_{dCCN}$, $N_{dT}$ and $N_d$ from the cloud probes are discussed in Section 5.2. Measurements of $N_d$ for each probe are considered only for concentrations greater or equal 20 droplets per cubic centimeter, to focus on the convective elements and avoid highly mixed and dissipating portions of the clouds. The time and distance differences between the measurements below cloud base and at cloud base have maximum values of 1 hour and 30 km, respectively. With this consideration, we assume that the $N_d$ measurements at cloud base pertain to the same

region as the CCN measurements below cloud base.

According to Twomey (1959), the $N_d$ that should be observed at cloud base increases with $W_b$ (assuming a constant CCN concentration; see Eq. 2). However, at cloud base the variability of $W_b$ and $N_d$ measurements is high due to air turbulence. Since a cloud parcel moves as an eddy with a local $W_b$ that produces a given $N_d$ at cloud base, its continued movement as a turbulent eddy within the cloud adds a large random component to the individual

realizations of $W_b$ for a given $N_d$. These turbulent characteristics greatly reduce the confidence that a given measured $W_b$ within cloud has produced the corresponding measured $N_d$, and therefore, these measurements are often not well correlated. A suitable method to analyze the relationship between $W_b$ and $N_d$ measurements is the 'probability matching method' (PMM) (Haddad and Rosenfeld, 1997). For a set of measurements of $W_b$ and $N_d$ at cloud base, it is expected that larger $W_b$ would produce larger $N_d$ for a given $N_{CCN}(S)$. In a PMM analysis, the same percentiles of updrafts are matched to the same percentiles of $N_d$ (or $N_{dCCN}$ and $N_{dT}$). As $N_d$ must be produced by positive updrafts

(Eq. 2), negative (positive) values of $W_b$ are associated with lower (higher) $N_d$. This procedure allows identifying the role of $W_b$ (positive) in producing $N_d$ in a set of cloud base measurements. The results of PMM analysis from cloud probes $N_d$ versus $W_b$, and for estimated $N_{dCCN}$ with $N_{dT}$ are discussed in Section 5.2.1.

### 4.3. Estimating $W_b^*$, $N_d^*$, $N_{dT}^*$ and $N_{dCCN}^*$


The formulation of an effective updraft speed at cloud base ($W_b^*$) is a useful approximation of the updraft spectrum (Rosenfeld et al., 2014a). $W_b^*$ and $N_d^*$ are given in Eqs. (6) and (7):

$$W_b^* = \frac{\int W_b^2}{\int W_b} \qquad ; where\ W_b > 0 \qquad (6)$$

$$N_d^* = N_d[percentile\ (W_b^*)] \qquad (7)$$





where $N_d$* represents the spectrum of $N_d$ at cloud base that matches the same percentile of $W_b$*. Figure 7 shows an illustration and example of the estimated value of $W_b$* and $N_d$* from the CCP-CDP probe for flight AC17. In this case the calculated $W_b$* has a value of 1.83 m s$^{-1}$, which represents the 86th percentile of total measurements at cloud base when sorted by $W_b$ measurements, including negative values. The corresponding percentile of $N_d$* (when sorted by $N_d$) in this case is 1207 cm$^{-3}$. Another approach for $N_d$ retrieval is the calculation of $N_{dT}$* considering $W_b$*

as the updraft wind speed in Eq. 2. In addition, $S_{max}$ can be estimated by applying the calculated values of $W_b$* and $N_d$* to Eq. 3. Then, applying the obtained $S_{max}$ to Eq. 1 yields $N_{dCCN}$*. The values of the calculated $N_{dT}$* and $N_{dCCN}$* in this case are 1,175 cm$^{-3}$ and 915 cm$^{-3}$, respectively.

### 4.4. Estimating $N_a$

Another approach for estimating $N_d$ is through the calculation of the adiabatic cloud droplet number concentration, $N_a$ (Freud et al., 2011). The $N_a$ is calculated from CWC and the mean volume droplet mass ($M_v$) calculations from the cloud probe DSDs obtained during the cloud profiling measurements. This behavior is the outcome of the almost completely inhomogeneous mixing behavior of the clouds with the ambient air (Burnet and Brenguier, 2007; Freud et al., 2011). Recently, Beals et al. (2015) wrote that their "*measurements reveal that turbulent clouds are*

*inhomogeneous, with sharp transitions between cloud and clear air properties persisting to dissipative scales (<1 centimeter). The local droplet size distribution fluctuates strongly in number density but with a nearly unchanging mean droplet diameter*". The dominance of inhomogeneous mixing diminishes when the drops become very large ($r_e$>15 μm) and their evaporation rate becomes more comparable to the mixing rate. This is most evident in those cloud passes where CWC is greater than 25 % of the adiabatic CWC (Freud et al., 2011). The measurements during

cloud profiling flights were aimed at penetrating the tops of growing convective towers (as shown at Fig. 1). This was done successfully in the data selected for analysis, as verified by examination of videos recorded by the cockpit camera of HALO. The cloud penetrations occurred mainly near the tops of growing convective cumulus, where mixing is expected to be rather inhomogeneous and little precipitation can fall from above. The validity of this expectation will affect the agreement between $N_d$ and $N_a$. The $N_a$ is calculated from the slope of CWC and $M_v$

measurements and provides an estimate of the maximum $N_d$ that should be observed within clouds (i.e., the maximum $N_d$ observed at cloud base of growing cumulus clouds). However, this methodology does not account for cloud mixing losses from droplet evaporation and the $N_a$ estimates commonly overestimate the expected $N_d$ by 30 % (Freud et al., 2011). Therefore, in calculating $N_a$ we applied this 30 % correction.

**5. Results**

### 5.1 CCN measurements below cloud base

The estimation of the $N_0$ and $k$ parameters in Eq. 1 is made from CCN and $S$ measurements below cloud base. Figure 8 illustrates CCN and $S$ measurements below cloud base for flight AC17 over a deforested region in the central Amazon. The cloud base is located at a height of about 2,300 m. The values of $S_1$ are constant at ~0.55 % and the

values of $S_2$ range from 0.2 % to 0.55 %. During these measurements, $CCN_1$ showed higher values than $CCN_2$, which is in agreement with its larger S, and the difference between $CCN_1$ and $CCN_2$ increases with decreasing $S_2$



(e.g., at time ~ 19:45 UTC, where $CCN_2$ values are around 300 cm$^{-3}$ and $CCN_1$ values are around 700 cm$^{-3}$). The $mCCN_1$, $mCCN_2$, and $NCCN_2$ for this group of measurements are shown in Fig. 9. In addition, the power fit equation from $NCCN_2$ and $mS_2$ measurements is shown and the values of $N_0$ and $k$ are 1015 cm$^{-3}$ and 0.54, respectively. The
error estimates of these calculations are shown in Table 2.

This procedure was applied to all cloud profiling flights with measurements of $N_{CCN}(S)$ with variable $S$ below cloud base. The $N_0$ and $k$ slope parameters for all groups of measurements during the campaign are shown in Fig. 10. The measurements show that for the less polluted conditions, the values of $N_0$ ($k$ slope) are near 1000 (0.5), while for more polluted conditions, values of $N_0$ ($k$ slope) greater than 2000 (0.9) are observed. Additionally, the correlation
coefficient values for almost all power fit equations are around 0.9. The estimated standard error (STDE) for the $N_0$ and $k$ parameters and CCN estimates were calculated (as described in Appendix A) for each flight segment and are shown in Table 2. The table shows that the STDE observed for the $N_0$ and $k$ parameters is lower than 5% of the mean values. For example, the maximum STDE observed for all $CCN$ estimates is ~70 cm$^{-3}$ in the most polluted case, where $N_0$ was 4145 cm$^{-3}$ (AC13).


### 5.2 Comparing estimated with measured $N_d$ near cloud base

Cloud base drop concentrations obtained in several different ways were compared. Appendix B summarizes the measurements and theoretical calculations at cloud base. Agreement between these different estimates constitutes a closure. Section 5.2.1 discusses comparisons between cloud probe individual $N_d$ measurements with the
corresponding theoretical estimations of $N_{dT}$ and $N_{dCCN}$. Section 5.2.2 describes the comparisons between estimated $N_d^*$, $N_{dT}^*$ and $N_{dCCN}^*$. Section 5.2.3 analyzes the agreement between $N_d^*$ and $N_a$.

### 5.2.1 Comparison between $N_d$ measurements with estimated $N_{dT}$ and $N_{dCCN}$

The PMM procedure was applied to the measured $W_b$ and $N_d$ for analyzing the spectrum of $N_d$, $N_{dT}$, and $N_{dCCN}$ values
near cloud base (as described in Section 4.2). This analytical method makes it possible to identify the role of $W_b$ in producing $N_d$. A perfect agreement of the values is not expected due to the turbulent nature of the clouds, but the statistical modes of the measurements should have similar values to the theoretical estimation of the same modes of $N_{dCCN}$ and $N_{dT}$, within their uncertainty range (calculations shown at Appendix C). Figures 11 and 12 show $N_{dCCN}$, $N_{dT}$, and $N_d$ values for the two cloud probes as a function of $W_b$ for the cases presented in Table 3. The uncertainties
regarding the $S_{max}$, $N_{dCCN}$ and $N_{dT}$ estimates for measurements at cloud base with both probes (CCP-CDP and CAS-DPOL) are about 20, 30 and 17 %. The values of $Ncas$ are with the range of the theoretical expectation of $N_{dT}$ ($N_{dCCN}$), with higher values of about 6 % (21%) for all flights analyzed (best accuracy in AC11, AC14, and the polluted portion of AC17; Fig. 11e). Slightly higher values (~15%) of $Ncas$ as compared to $N_{dT}$ are observed in the maximum values for flight AC14, probably associated with measurements in pollution plumes. Flight AC16 shows
measurements slightly lower (~25%) than expected theoretically. A similar behavior of measurements can be observed for the $Ncdp$ values (which presents higher values of about 8% in comparison with $N_{dT}$ estimates for all cases), with slight differences with respect to $Ncas$.

The PMM analysis provided the information that CAS-DPOL and CCP-CDP agree within 30 % to theoretical





estimates for different aerosol conditions, $N_{CCN}(S)$, below cloud base. These results support the analyses concerning

the $N_d$ measurements at cloud base that are presented in the next sections.

*5.2.2 Comparing estimated $N_d*$ with $N_{dT}*$ and $N_{dCCN}*$*

Assuming that $W_b*$ represents the updraft velocity for a set of cloud base measurements, the corresponding measured $N_d*$ from CAS-DPOL and CCP-CDP ideally should have similar values to the estimated $N_{dCCN}*$ and $N_{dT}*$.

The uncertainties of $N_{dCCN}*$ and $N_{dT}*$ are ~30 % and ~17 %, respectively. Figure 13a shows the values of $N_d*$ and $N_{dT}*$ for the different cloud base measurements shown in Figs. 11 and 12. The results indicate that most of $N_d*$ calculated from CAS-DPOL and CCP-CDP measurements are in agreement with theoretically calculated $N_{dT}*$ with differences up to 10% (except for measurements from flight AC17 – less polluted case). A similar behavior is observed for comparisons between $N_{dCCN}*$ and $N_d*$ (see Fig. 13b), but with differences of a factor of ~2 for some

cases (AC14 and AC17). Regarding the Twomey formulation (Eq. 2), Figs. 11 and 12 show that: CCP-CDP and CAS-DPOL agree closely with the calculations. Summarized, for this type of closure analysis, the CAS-DPOL and CCP-CDP measurements achieve agreement with theoretical estimates.

*5.2.3 Comparing estimated $N_d*$ with $N_a$*

Another possibility of cloud base closure is via comparison of $N_d*$ and $N_a$ estimates from $N_d$ measurements in pristine and polluted conditions. In these situations, the estimated values for these parameters must converge. Figure 14a shows the calculated $N_a$ with CCP-CDP probe results from cloud measurements during flight AC17. The estimated $N_a$ in this case is 1496 cm$^{-3}$, and, considering evaporation losses due to cloud mixing, the expected number of droplets at cloud base is 1047 cm$^{-3}$ after applying the correction by division by 1.3 (Freud et al., 2011). $N_d*$ for

the same flight segment is 1207 cm$^{-3}$, calculated from CCP-CDP data (see Fig. 7). A close relationship between $M_v$ and $r_e$ as a function of height is shown at Figure 14b. Similar results were found for cloud profile measurements during the other flights.

Values of $N_a$ and $N_d*$ were calculated for all profile flights and cloud probes, and the results are shown in Fig. 15. The uncertainties of $N_a$ ($N_d*$) estimates with CAS-DPOL and CCP-CDP are ~25% (21 %) and ~14% (10%),

respectively. The comparisons between the estimated $N_a$ and $N_d*$ show a linear correlation with correlation coefficients greater than 0.9 for all cloud probes. The linear regression between $N_d*$ and $N_a$ estimates shows a slope close to one for CAS-DPOL and CCP-CDP. The similarities of $N_a$, $N_d*$ values for these several cases supports the methodology to calculate from vertical profile of measured $r_e$ or $M_v$, (see Fig. 14a-b) the effective number of droplets observed at cloud base of convective clouds.

These results show good agreement with theoretical expectations when done with CAS-DPOL and CCP-CDP. The flights performed in near-pristine and polluted conditions can be distinguished based on the CAS-DPOL estimates from $N_d*$ and $N_a$ values. For example, in flight AC19 performed over the Atlantic Ocean in clean conditions, the CAS-DPOL estimated values of $N_d*$ and $N_a$ are ~270 cm$^{-3}$, whereas for flights AC07 and AC11 performed under polluted conditions, the values of $N_d*$ and $N_a$ are greater than 1000 cm$^{-3}$. The results from CCP-CDP also showed

good agreement between $N_d*$ and $N_a$ estimates.



For most polluted flights, e.g., AC08, $N_a$ is larger than $N_d*$ by a factor of ~20-30%. The vertical profiles of the $N_d$ measurements indicate that in these cases the $N_d$ measurements up to 2-3 kilometers above cloud base were larger than those at cloud base. A higher aerosol concentration at these greater heights was also observed in aerosol probe measurements (not shown), suggesting that secondary droplet nucleation was taking place on the most polluted flights. The $N_a$ calculation does not take into account the possibility of new nucleation above cloud base (Freud et al., 2011). Therefore, the assumption of adiabatic growth of droplets via condensation from cloud base to higher levels within cloud leads to an overestimation by ~20-30% of the number of droplets at cloud base when calculating $N_a$.

The results of Sections 5.2.1 and 5.2.2 highlight that a closure between cloud base measurements and theoretical models was achieved with the CCP-CDP and CAS-DPOL probes. The results from these probes support the methodology to derive $N_a$ based on the rate of $r_e$ growth with cloud depth and with respect to the nature of cloud mixing with the entrained ambient air (Freud et al., 2011). Overall, a perfect relationship between the cloud probe measurements and the theoretical models is not expected. However, a large bias in the measurements would result in a correspondingly large deviation from the calculated values, and closure would not be reached (e.g., when strong secondary nucleation of droplets occurs above cloud base).

## 6. Conclusions

This work was focused on testing for closures between cloud properties derived from observation-based theoretical estimates and measurements during the ACRIDICON-CHUVA campaign. In addition, liquid water content measurements from a hot-wire device were taken as a reference for the quality assessment of estimated CWC from cloud probe DSDs near cloud base. The intercomparison of the cloud drop size distributions (DSDs) and the cloud water content derived from the different instruments generally shows good agreement within the instrumental uncertainties. The values of $N_d$ near cloud base were compared with their inferred values based on the measured $W_b$ and $N_{CCN}(S)$ spectra. The measured effective droplet numbers ($N_d*$) at cloud base were also compared with the theoretically calculated $N_{dT}*$ and $N_{dCCN}*$. In addition, $N_d$ near cloud base was compared with $N_a$, obtained from the vertical evolution of cloud drop effective radius ($r_e$) above cloud base. Comparisons of estimated $N_d*$ with $N_{dT}*$ and $N_{dCCN}*$ from the measurements showed good agreement with measurements from CAS-DPOL and CCP-CDP. Similar agreement was observed for comparisons between $N_d*$ and $N_a$. The results support the methodology to derive $N_a$ based on the rate of $r_e$ growth with cloud depth and under the assumption that the entrainment and mixing of air into convective clouds is extremely inhomogeneous. In summary, the measurements of $N_{CCN}(S)$ and $W_b$ did reproduce the observed $N_d$. Furthermore, the vertical evolution of $r_e$ with height reproduced the observation-based adiabatic cloud base drop concentrations, $N_a$. Our study supports the validity of the parameterizations used, which now can be applied more confidently to simulations and satellite retrievals.

### Acknowledgements

The first two authors of this study were supported by project BACCHUS European Commission FP7-603445. The generous support of the ACRIDICON-CHUVA campaign by the Max Planck Society, the German Aerospace Center





(DLR), FAPESP (São Paulo Research Foundation), and the German Science Foundation (Deutsche Forschungsgemeinschaft, DFG) within the DFG Priority Program (SPP 1294) "Atmospheric and Earth System Research with the Research Aircraft HALO (High Altitude and Long Range Research Aircraft)" is greatly appreciated. This study was also support by EU Project HAIC under FP7-AAT-2012-3.5.1-1 and by the German Science Foundation within DFG SPP 1294 HALO by contract no VO1504/4-1 and contract no JU 3059/1-1. The first author also acknowledges the financial support from the Brazilian funding agencies CAPES and CNPq during his Ph.D. degree studies.

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

**Appendix - A**

*Calculating STDE CCNmax and STDE CCNmin*

The $N_0$ and $k$ parameters standard errors (STDE) are associated with the statistical uncertainty of the power law function fit. To compute the STDE for the CCN estimates the uncertainties of $S$ (~10%) are considered. Then, the maximum and the minimum STDE values expected for the CCN estimates are calculated as follows:

*Maximum STDE*

$$STDE\ CCNmax = \frac{[\ (N_0 + SD.N_0) \cdot (S_i \cdot 1.1)^{k+SD.k}] - N_0 \cdot S_i^k}{\sqrt{N}} \qquad (A1)$$

*where:*

The averaging is done on I=1:N.

N is the number of $NCCN_2$ cases for each group of measurements.

SD.$N_0$ is the statistical standard deviation of $N_0$;





SD.k is the statistical standard deviation of $k$;

S$_i$ is the supersaturation in each step, forced to have the maximum value (multiplied by 1.1).

*Minimum STDE*


$$STDE\ CCNmin = \frac{N_0 \cdot S_i^k - [\ (N_0 - SD.N_0) * (S_i \cdot 0.9)^{k-SD.k}]}{\sqrt{N}} \qquad (A2)$$

*where:*

The averaging is done on I=1:N.

N is the number of $NCCN_2$ cases for each group of measurements.

SD.$N_0$ is the statistical standard deviation of $N_0$;

SD.k is the statistical standard deviation of $k$;

S$_i$ is the supersaturation in each step, forced to have the minimum value (multiplied by 0.9).

**Appendix – B**

Summary of the measurements and theoretical calculations at cloud base:

1)   $N_d$ - based on probe measurement;

      2)   $N_a$ - based on vertical profile of $r_e$;

      3)   $S_{max} - S$ substituting $N_d$ and $W_b$ in Eq. 3.

      4)   $N_{dT}$ - Obtained from substituting in Eq. 2 $W_b$ and $N_{CCN}(S)$ parameters ($k$ and $N_0$);

      5)   $N_{dCCN}$ - Obtained from substituting $S_{max}$ and $N_{CCN}(S)$ parameters in Eq. 1.

6)   $W_b$* - Obtained from Eq. 5.

      7)   $N_d$*, $N_{dT}$*, $N_{dCCN}$* - $N_d$, $N_{dT}$, $N_{dCCN}$ that match $W_b$*.

**Appendix – C**

*Calculating uncertainty for $N_{dT}$*

The calculation is similar of *STDE CCNmax* and *STDE CCNmin* considering also the $W_b$ uncertainty (assumed the maximum; 0.2 m s$^{-1}$). Therefore, the maximum and the minimum uncertainty values of $N_{dT}$ are calculated as follow:

*Maximum uncertainty*

$$N_{dTmax} = \frac{0.88 \cdot [N_{0max}^{2/kmax+2} \cdot (0.07 W_{bmax}^{1.5})]^{\frac{kmax}{kmax+2}} - 0.88 \cdot [N_0^{2/k+2} \cdot (0.07 W_b^{1.5})]^{\frac{k}{k+2}}}{\sqrt{N}} \qquad (C1)$$

*Minimum uncertainty*

$$N_{dTmin} = \frac{0.88 \cdot [N_0^{2/k+2} \cdot (0.07 W_b^{1.5})]^{\frac{k}{k+2}} - 0.88 \cdot [N_{0min}^{2/kmin+2} \cdot (0.07 \cdot W_{bmin}^{1.5})]^{\frac{kmin}{kmin+2}}}{\sqrt{N}} \qquad (C2)$$



*where:*

$N_{0max} = N_0 + SD.N_0$

$N_{0min} = N_0 - SD.N_0$

kmax= k+SD.k

kmin= k-SD.k

$W_{bmax} = W_b + 0.2$

$W_{bmin} = W_b - 0.2$; for $W_b$ greater than 0.2

N is the number of $N_d$ measurements for each group of measurements.

*Calculating uncertainty for $N_{dCCN}$*

The calculation is similar of *STDE CCNmax* and *STDE CCNmin* considering also the $S_{max}$ uncertainty (assumed the maximum; 20 %). Therefore, the maximum and the minimum uncertainty values of $N_{dCCN}$ are calculated as follow:

$$N_{dCCNmax} = \frac{[N_{0max}^{2/kmax+2} \cdot (Smax \cdot 1.2)]^{\frac{kmax}{kmax+2}} - [N_0^{2/k+2} \cdot (Smax \cdot 1.2)]^{\frac{k}{k+2}}}{\sqrt{N}} \qquad (C3)$$


$$N_{dCCNmin} = \frac{[N_0^{2/k+2} \cdot (Smax \cdot 1.2)]^{\frac{k}{k+2}} - [N_{0min}^{2/kmin+2} \cdot (Smax \cdot 0.8)]^{\frac{kmin}{kmin+2}}}{\sqrt{N}} \qquad (C4)$$

*where:*

$N_{0max} = N_0 + SD.N_0$

$N_{0min} = N_0 - SD.N_0$

kmax= k+SD.k

kmin= k-SD.k

N is the number of $N_d$ measurements for each group of measurements.

**Figure captions**

Figure 1. Flight patterns below and in convective clouds during the ACRIDICON-CHUVA campaign.

Figure 2. HALO flight tracks during the ACRIDICON-CHUVA experiment. The flight numbers are indicated on the right (from Wendisch et al., 2016).

Figure 3. Cloud droplet effective radius ($r_e$) as a function of altitude for clouds over clean (Flight AC19 - blue color squares ), polluted (Flight AC18 – green color triangles) and very polluted (Flight AC13 – brown color diamonds)

environments. Dashed lines indicate the probability of rain from the coalescence process expressed in percentage on the top of the graphic.





Figure 4. Mean cloud water content from the hot-wire measurements and estimated from the cloud probes (CCP-CDP and CAS-DPOL from top to bottom, respectively) as a function of effective radius ($r_e$) size (left panel). The ratios between the hot-wire liquid water content and the cloud water content derived from each probe are shown in red (*CWCr*). The total uncertainty for each probe and the hot-wire measurements are shown by the dotted lines. The number of cases (black continuous line), hot-wire measurement standard deviations (dashed black line), and probe CWC standard deviations (dashed colored line) for each $r_e$ size are shown in the right panels.

Figure 5. Mean cloud droplet concentrations for CAS-DPOL and CCP-CDP as a function of effective radius ($r_e$) (left panel). The systematic error for each probes shown by the dashed line. The right panel indicates the standard deviation in cm$^{-3}$ of each probe concentration as a function of $r_e$. The probes are identified by colors as shown in the top of the panels. The sample for each probe is the same as shown in Figure 3.

Figure 6. (left) Mean cloud droplet concentration (solid lines) and (right) cloud water content as a function of droplet diameter in the left and right panels, respectively, for a) 5 µm < $r_e$ < 6 µm; b) 8 µm < $r_e$ < 9 µm; c) 11 µm < $r_e$ < 12 µm; d) 12 µm < $r_e$ < 13 µm. The probes are identified by colors as shown at the top of the panels. The dashed lines indicate the uncertainty range of mean cloud droplet concentration and cloud water content values as a function of droplet diameter.

Figure 7. a) Frequency histogram for vertical wind speed ($W_b$) from cloud base measurements on flight AC17 (labeled on the left ordinate). The blue line indicates the cumulative probability function of $W_b$ (labeled on the right ordinate). The cyan arrow indicates the value of $W_b$* (1.83 m s$^{-1}$), which represents the 86th percentile of the $W$ spectra; b) Similar for the cloud droplet concentrations measured with the CCP-CDP probe. The cyan line indicates the $N_d$* value (1207 cm$^{-3}$) at the 86th percentile in the $N_d$ spectra. The indicated time is in UTC and shows the time of the first cloud penetration at cloud base and the total number of 1-s measured cloud data points.

Figure 8. $CCN_1$ (red dots) and $CCN_2$ (black dots) measurements for a segment of flight AC17 on 27 September 2014. The abscissa shows the measurement time in UTC. The blue line indicates the altitude in meters above sea level and is labeled on the left ordinate (as well as $CCN_1$ and $CCN_2$). $S_1$ and $S_2$ measurements in % are indicated by the orange and green lines, respectively (both are labeled on the right ordinate). Cyan dots on the blue line indicate cloud penetrations (i.e., when cloud droplets concentrations are greater than 20 cm$^{-3}$). In this case, cloud base heights were observed around 2,300 meters above ground.

Figure 9. A comparison of the $CCN$ spectra derived from the two $CCN$ counter columns on board the HALO aircraft during flight AC17. Black (blue) smaller dots indicate $CCN_1$ ($CCN_2$) measurements for each second. Large diamonds in black (blue) indicate the $mCCN_1$ (mCCN$_2$) for each time step of measurements. The orange large diamonds indicate the $NCCN_2$ values, which are used to fit the power law equation of the group of measurements, which is shown at the lower right corner of the plot. The standard error for CCN spectra derived is shown at Table 2.

Figure 10. $CCN$ spectra as measured on board the HALO aircraft during cloud profiling flights. Diamonds indicate the $NCCN_2$ values, which are used to fit the power law equation of the group of measurements. The colors indicate the group of measurements and match the legend on the right side of the plot. The legend indicates the flight number; the initial time of group measurements; the period of measurements in seconds; the power law fit and the correlation coefficient of the data. The standard errors for each CCN spectra derived are shown at Table 2.





Figure 11a-f. $N_{dCCN}$, $S$, $N_{dT}$ and $N_d$ values are presented as a function of the cloud base updrafts ($W_b$). This plot is
based on the 'probability matching method' (PMM), using same percentiles for $W_b$ and $N_d$ ($N_{dCCN}$ or $N_{dT}$). The val-
ues of $N_{dCCN}$, $N_{dT}$ and $N_d$ are shown the left y-axis, those of $S$ on the right y-axis. The black dashed lines are the $N_{dT}$
uncertainties. The gray solid (dashed) lines are the $N_{dCCN}$ values (uncertainties). The effective updraft $W_b*$ for each
flight segment is shown by the cyan line. The data are based on the CAS-DPOL probe. The time, period of meas-
urements (sample size in seconds), and $CCN(S)$ equation are shown on the top of the figures

Figure 12a-d. Same as Figure 7 for the CCP-CDP probe. No data were available for flight AC16. The CCP-CDP
malfunctioned in flight AC13 during the cloud base measurements.

Figure 13. a) $N_d*$ versus $N_{dT}^*$ calculated with $W_b*$ from cloud base data shown in Figures 7-8. The CAS-DPOL
values are indicated by plus symbols (+) and the CCP-CDP values are indicate by circles (o). The colors indicate
each flight segment (legend in the right side of the plot). Error bars indicates the uncertainties of variables estimates.
Lines show the 1:1 and 1:2 relationships between $N_{dT}^*$ versus $N_d*$ for each probe (dotted line – CCP-CDP; solid line
– CAS-DPOL); b) Same for $N_d*$ versus $N_{dCCN}^*$.

Figure 14 a). Mean volume drop mass ($M_v$) versus liquid water content from the CCP-CDP measurements for
adiabatic fraction greater than 0.25 ($LWC_a$). Vaues are shown with different colors labeled as a function of height in
kilometers above sea level (indicated by the colorbar on the right side of the graphic). The slope of the linear
equation is the estimated $N_a$ (i.e., 1496 cm$^{-3}$); b) $M_v$ versus re as a function of height in kilometers above sea level
(indicated by the colorbar on the right side of the graphic).

Figure 15. $N_d*$ versus $N_a$ measured with CAS-DPOL and CCP-CDP (indicated on the top of panels) for profile
flights during the ACRIDICON-CHUVA campaign. The color of the dots is associated with the flight number shown
at the right side of the panels. Error bars indicates the uncertainties of variables estimates. The linear regression
equation and the correlation coefficient $R$ are shown in the top of each panel.

**Table captions**

Table 1. Cloud probe size intervals and central bin diameters during HALO flights.

Table 2. Estimates of $N_0$ and $k$ below cloud base and their standard error (STDE) for each case study. Maximum and
minimum STDE (STDE $CCNmax$ and STDE $CCNmin$, respectively) for the $CCN$ measurements are calculated
considering errors in the supersaturation measurements (~10%). The details about the calculation of these
uncertainties are given in Appendix A.

Table 3. List of case studies for measurements below cloud base. The duration of measurements is given in seconds,
starting at the initial time indicated. An asterisk indicates those flights where the two probes provided at least 20
seconds of measurements at cloud base. The data can be from different cloud passes in the same region of
measurements below cloud base.





**Tables**

Table 1. Cloud probe size intervals and central bin diameters during HALO flights.


| Cloud Probe | Size interval | Number of bins | Central bin diameter (μm) |
|---|---|---|---|
| **CCP-CDP** | 3-50 μm | 14 | 3.8, 6.1, 8.7, 10.9, 13.5, 17.1, 19.7, 22.5, 25.9, 28.3, 31.7, 36.6, 40.7, 44.2 |
| **CAS-DPOL** | 3-50 μm | 10 | 3.9, 6, 10.8 ,17.3, 22.3, 27.4, 32.4, 37.4, 42.4, 47.4 |

Table 2. Estimates of $N_0$ and $k$ below cloud base and their standard error (STDE) for each case study. Maximum and minimum STDE (STDE *CCNmax* and STDE *CCNmin*, respectively) for the *CCN* measurements are calculated

considering errors in the supersaturation measurements (~10%). The details about the calculation of these uncertainties are given in Appendix A.

| Flight | Time | $N_0$ | k | STDE $N_0$ | STDE k | STDE *CCNmax* [cm$^{-3}$] | STDE *CCNmin* [cm$^{-3}$] |
|---|---|---|---|---|---|---|---|
| AC11 | 14:58:21 | 1985 | 0.73 | 81.6 | 0.035 | 25.5 | 24.8 |
| AC11 | 17:38:20 | 2927 | 1.14 | 82.8 | 0.032 | 43.9 | 43.8 |
| AC12 | 15:56:00 | 1764 | 0.3 | 71.4 | 0.046 | 19.0 | 22.7 |
| AC13 | 16:29:01 | 4145 | 0.92 | 64.7 | 0.016 | 69.7 | 54.8 |
| AC14 | 15:21:40 | 1509 | 0.97 | 44.8 | 0.028 | 24.7 | 18.9 |
| AC15 | 13:33:35 | 2209 | 0.94 | 70.4 | 0.038 | 47.4 | 31.2 |
| AC16 | 20:21:40 | 1966 | 0.67 | 69.5 | 0.029 | 26.5 | 21.2 |
| AC17 | 16:50:50 | 2743 | 0.72 | 38.7 | 0.013 | 31.9 | 30.5 |
| AC17 | 19:38:20 | 1015 | 0.54 | 18.5 | 0.018 | 10.7 | 9.4 |





Table 3. List of case studies for measurements below cloud base. The duration of measurements is given in seconds,
starting at the initial time indicated. An asterisk indicates those flights where the two probes provided at least 20
seconds of measurements at cloud base. The data can be from different cloud passes in the same region of
measurements below cloud base.

| Measurements below cloud base | | | |
|---|---|---|---|
| **Flight** | **Date** | **Initial time (UTC)** | **Period of analysis (s)** |
| AC11 | 16/09/2014 | 14:58:21 | 593 |
| AC11* | 16/09/2014 | 17:38:20 | 710 |
| AC12 | 18/09/2014 | 15:56:00 | 440 |
| AC13* | 19/09/2014 | 16:29:01 | 722 |
| AC14* | 21/09/2014 | 15:21:40 | 800 |
| AC15 | 23/09/2014 | 13:33:35 | 555 |
| AC16 | 25/09/2014 | 20:21:40 | 550 |
| AC17* | 27/09/2014 | 16:50:50 | 831 |
| AC17* | 27/09/2014 | 19:38:20 | 840 |




**Figures**

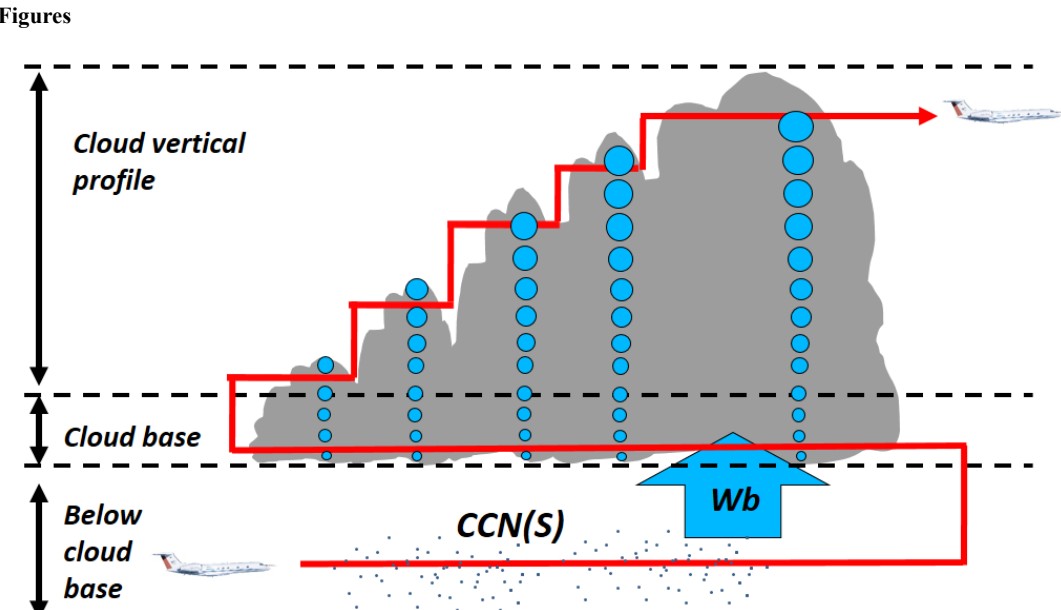


Figure 1. Flight patterns below and in convective clouds during the ACRIDICON-CHUVA campaign.









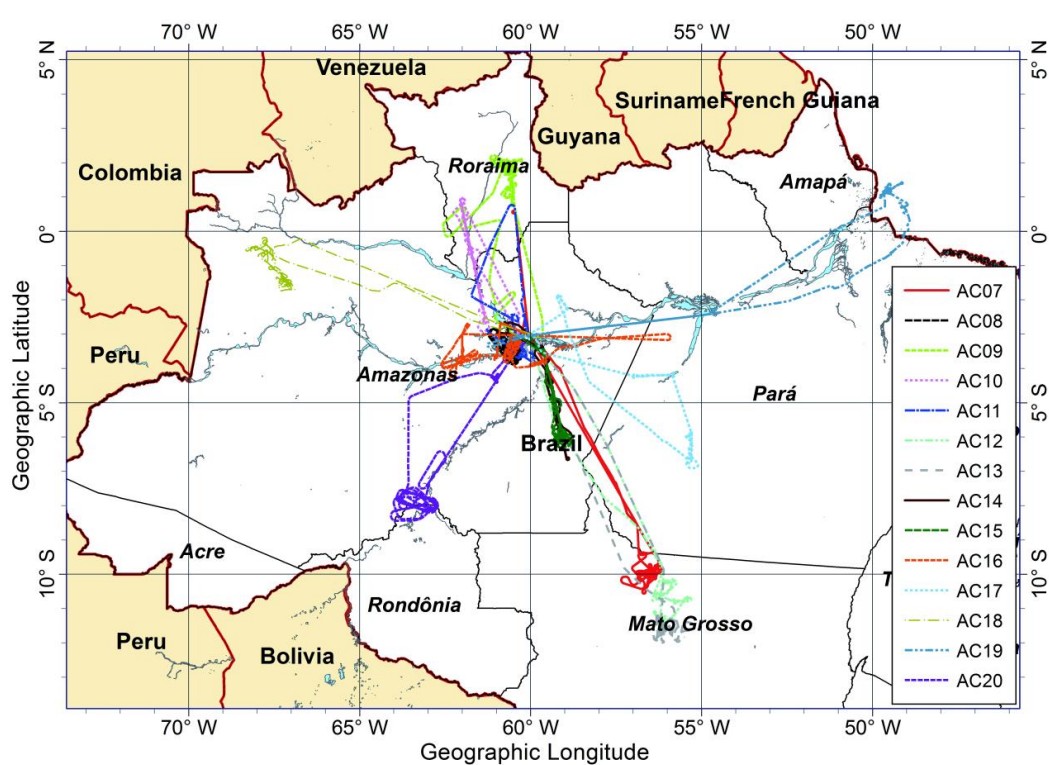

**Figure 2. HALO flight tracks during the ACRIDICON-CHUVA experiment. The flight numbers are indicated on the right (from Wendisch et al., 2016).**









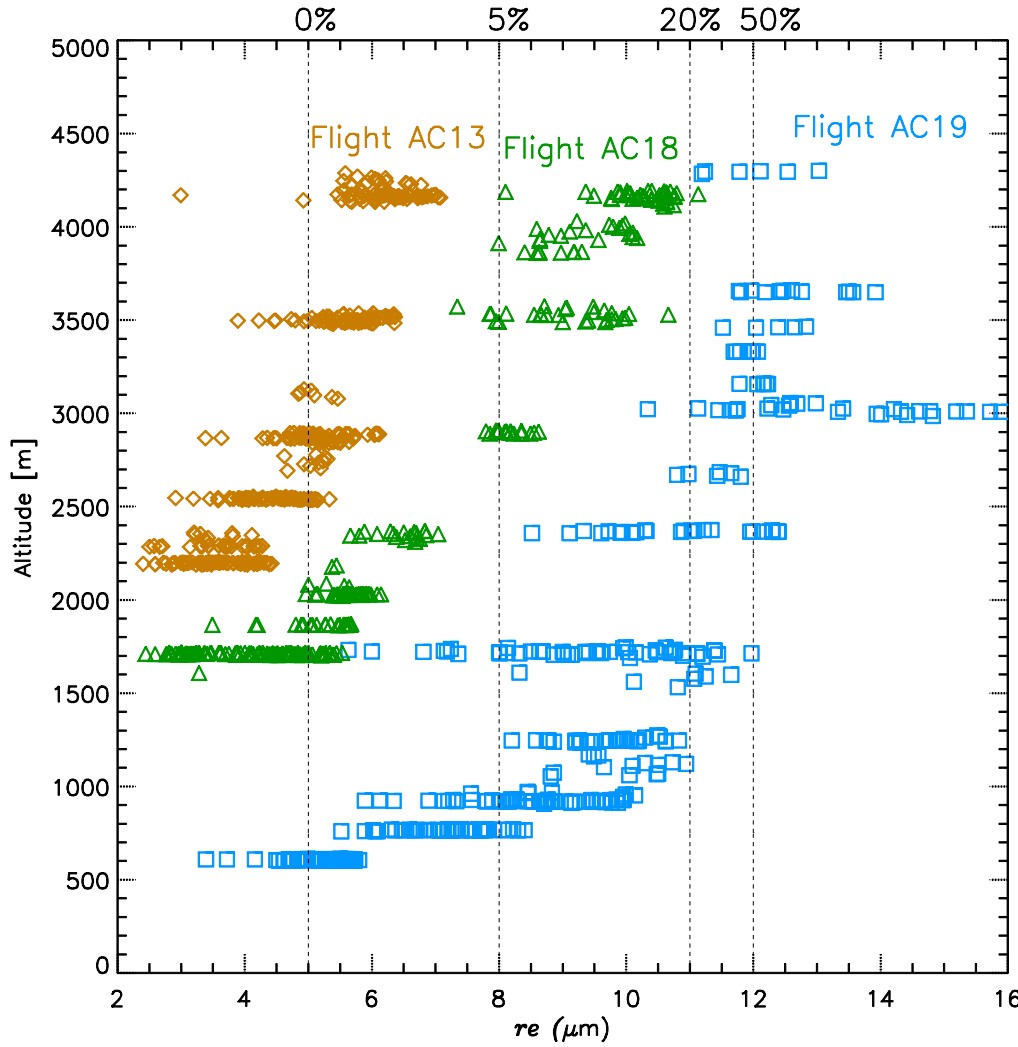


**Figure 3. Cloud droplet effective radius ($r_e$) as a function of altitude for clouds over clean (Flight AC19 - blue color squares ), polluted (Flight AC18 – green color triangles) and very polluted (Flight AC13 – brown color diamonds) environments. Dashed lines indicate the probability of rain from the coalescence process expressed in percentage on the top of the graphic.**





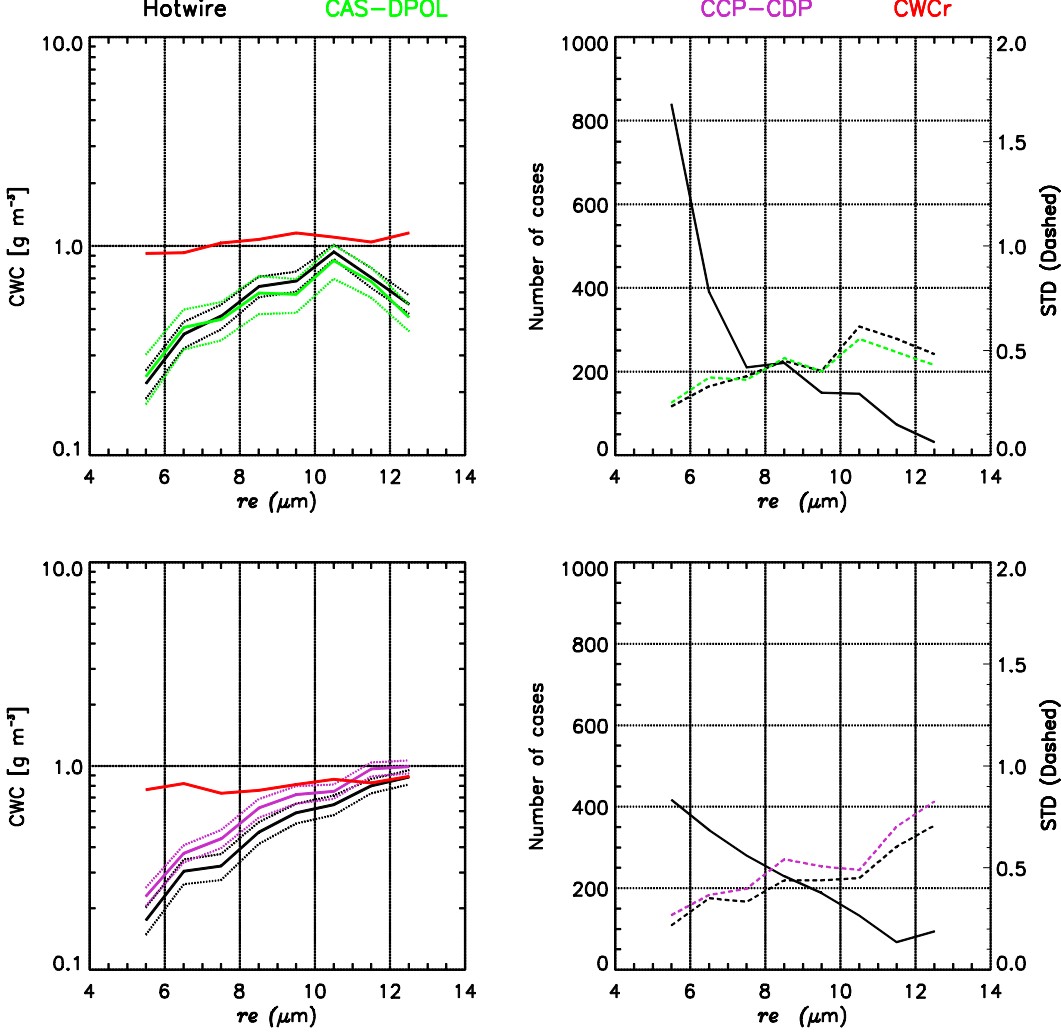

**Figure 4. Mean cloud water content from the hot-wire measurements and estimated from the cloud probes (CCP-CDP and CAS-DPOL from top to bottom, respectively) as a function of effective radius ($r_e$) size (left panel). The ratios between the hot-wire liquid water content and the cloud water content derived from each probe are shown in red (*CWCr*). The total uncertainty for each probe and the hot-wire measurements are shown by the dotted lines. The number of cases (black continuous line), hot-wire measurement standard deviations (dashed black line), and probe CWC standard**

**deviations (dashed colored line) for each $r_e$ size are shown in the right panels.**





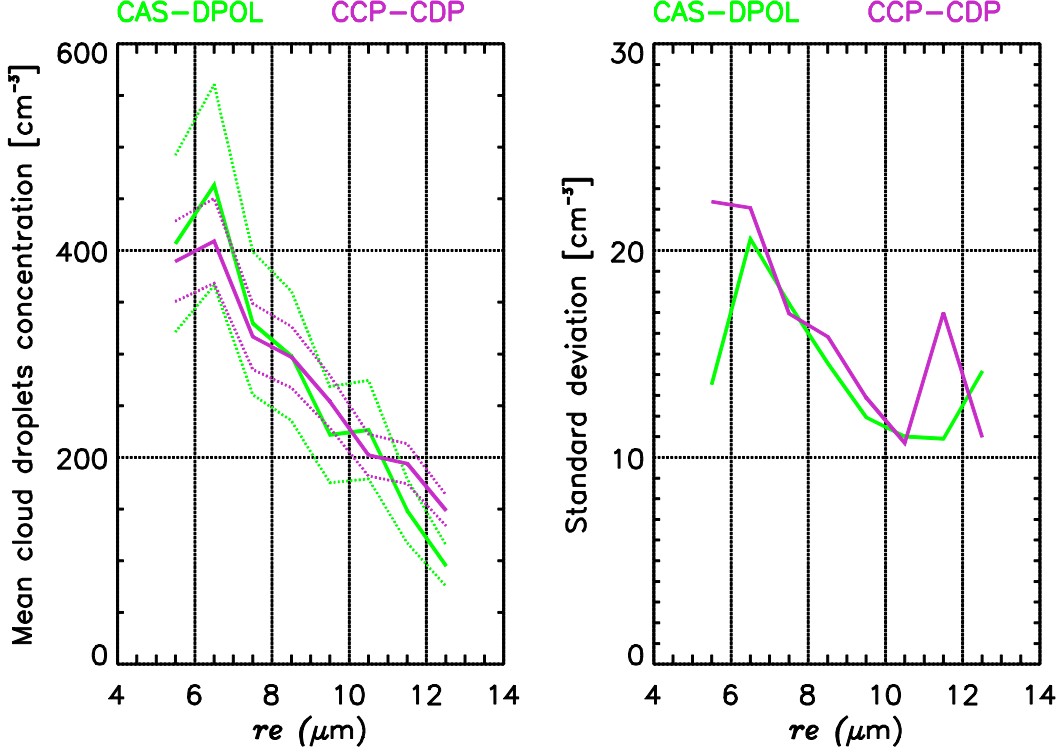

**Figure 5. Mean cloud droplet concentrations for CAS-DPOL and CCP-CDP as a function of effective radius ($r_e$) (left panel). The systematic error for each probes shown by the dashed line. The right panel indicates the standard deviation in cm⁻³ of each probe concentration as a function of $r_e$. The probes are identified by colors as shown in the top of the panels. The sample for each probe is the same as shown in Figure 3.**










**Figure 6.** (left) Mean cloud droplet concentration (solid lines) and (right) cloud water content as a function of droplet diameter in the left and right panels, respectively, for a) 5 μm < $r_e$ < 6 μm; b) 8 μm < $r_e$ < 9 μm; c) 11 μm < $r_e$ < 12 μm; d) 12 μm < $r_e$ < 13 μm. The probes are identified by colors as shown at the top of the panels. The dashed lines indicate the uncertainty range of mean cloud droplet concentration and cloud water content values as a function of droplet diameter.





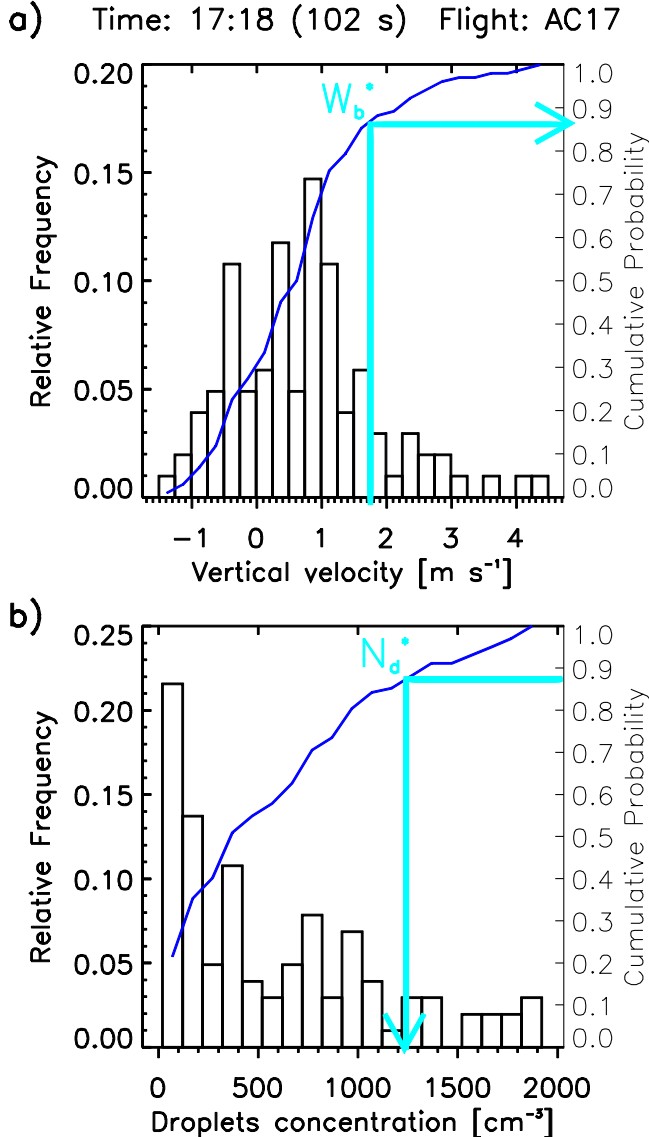

**Figure 7. a) Frequency histogram for vertical wind speed ($W_b$) from cloud base measurements on flight AC17 (labeled on the left ordinate). The blue line indicates the cumulative probability function of $W_b$ (labeled on the right ordinate). The cyan arrow indicates the value of $W_b$* (1.83 m s$^{-1}$), which represents the 86th percentile of the $W$ spectra; b) Similar for the cloud droplet concentrations measured with the CCP-CDP probe. The cyan line indicates the $N_d$* value (1207 cm$^{-3}$) at the 86th percentile in the $N_d$ spectra. The indicated time is in UTC and shows the time of the first cloud penetration at cloud base and the total number of 1-s measured cloud data points.**





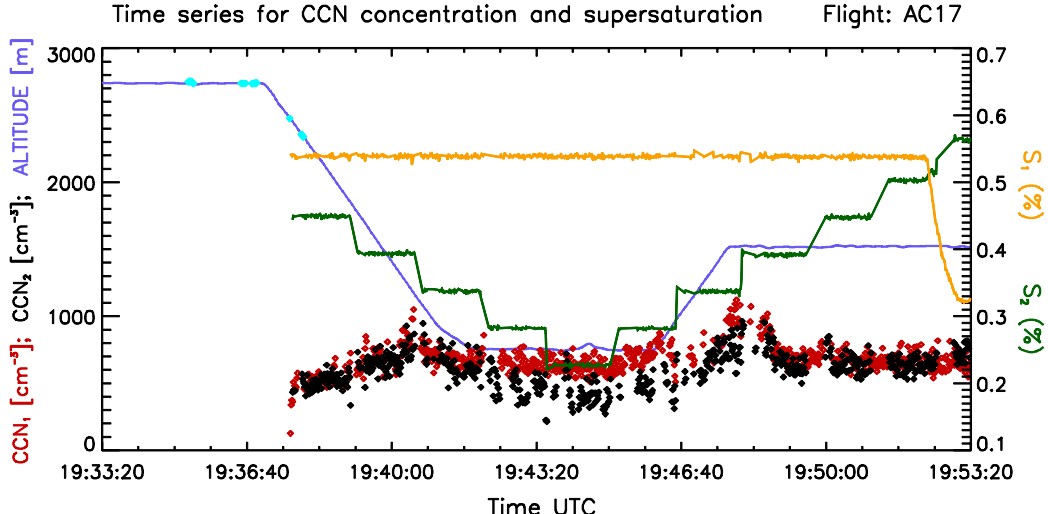

**Figure 8.** *CCN$_1$* (red dots) and *CCN$_2$* (black dots) measurements for a segment of flight AC17 on 27 September 2014. The abscissa shows the measurement time in UTC. The blue line indicates the altitude in meters above sea level and is labeled

on the left ordinate (as well as *CCN$_1$* and *CCN$_2$*). $S_1$ and $S_2$ measurements in % are indicated by the orange and green lines, respectively (both are labeled on the right ordinate). Cyan dots on the blue line indicate cloud penetrations (i.e., when cloud droplets concentrations are greater than 20 cm$^{-3}$). In this case, cloud base heights were observed around 2,300 meters above ground.







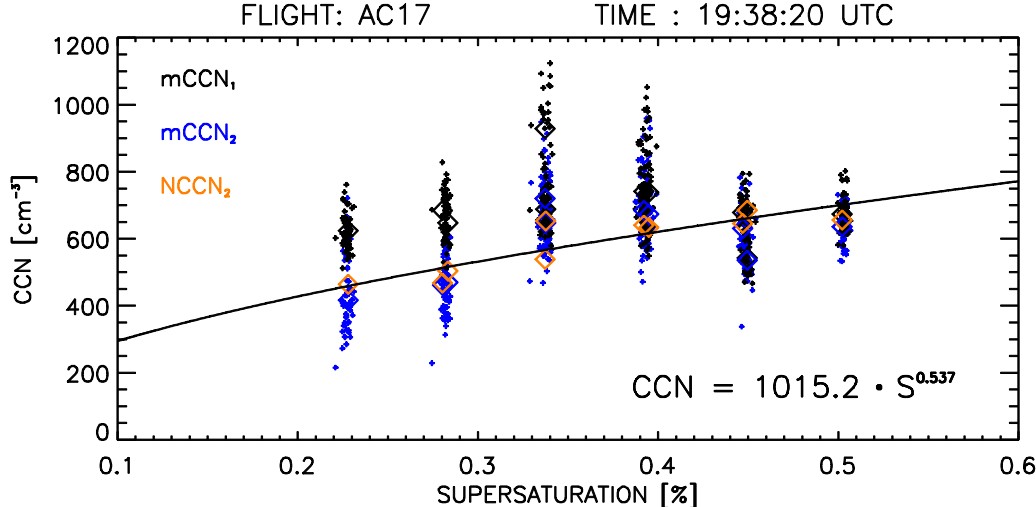


**Figure 9. A comparison of the *CCN* spectra derived from the two *CCN* counter columns on board the HALO aircraft during flight AC17. Black (blue) smaller dots indicate *CCN₁* (*CCN₂*) measurements for each second. Large diamonds in black (blue) indicate the *mCCN₁* (mCCN₂) for each time step of measurements. The orange large diamonds indicate the**

**NCCN₂ values, which are used to fit the power law equation of the group of measurements, which is shown at the lower right corner of the plot. The standard error for CCN spectra derived is shown at Table 2.**







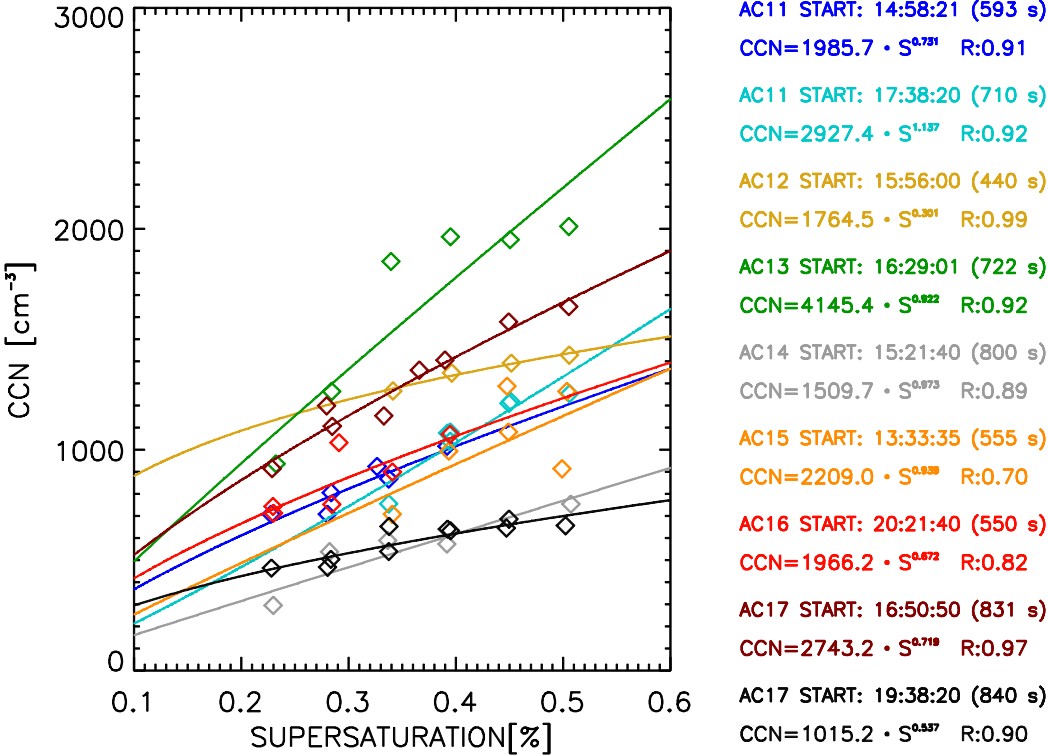

**Figure 10.** *CCN* **spectra as measured on board the HALO aircraft during cloud profiling flights. Diamonds indicate the** *NCCN₂* **values, which are used to fit the power law equation of the group of measurements. The colors indicate the group of measurements and match the legend on the right side of the plot. The legend indicates the flight number; the initial time of group measurements; the period of measurements in seconds; the power law fit and the correlation coefficient of the data. The standard errors for each CCN spectra derived are shown at Table 2.**





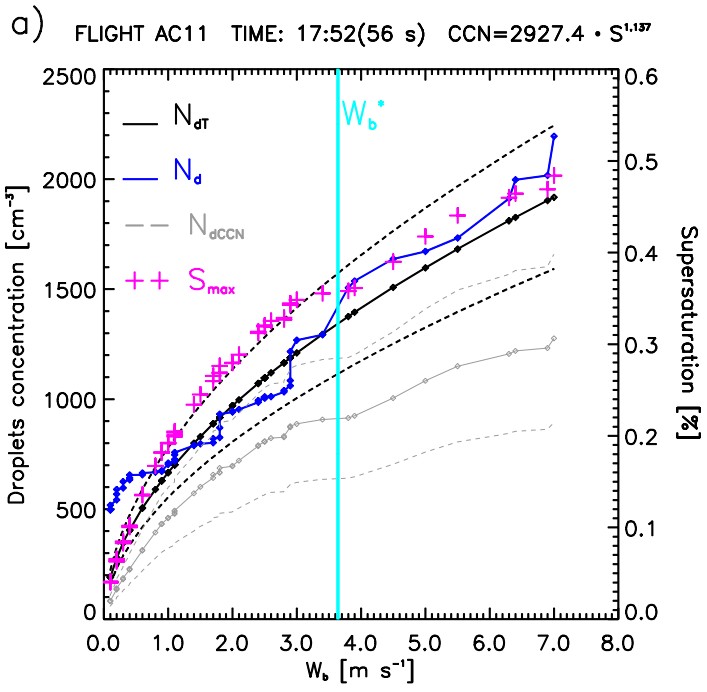

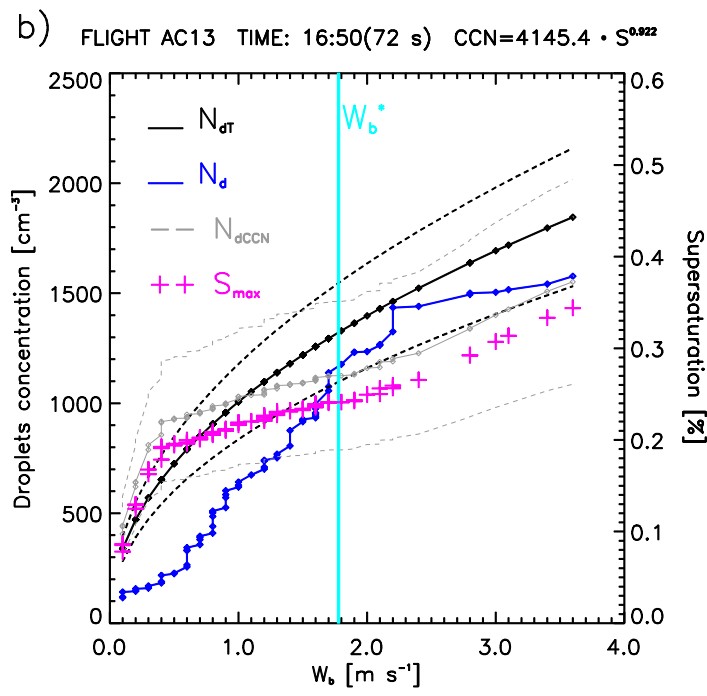



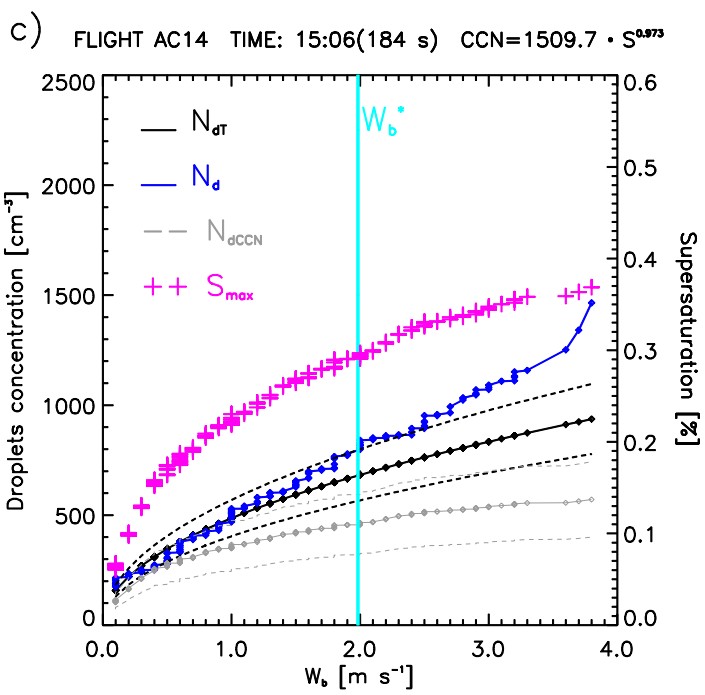

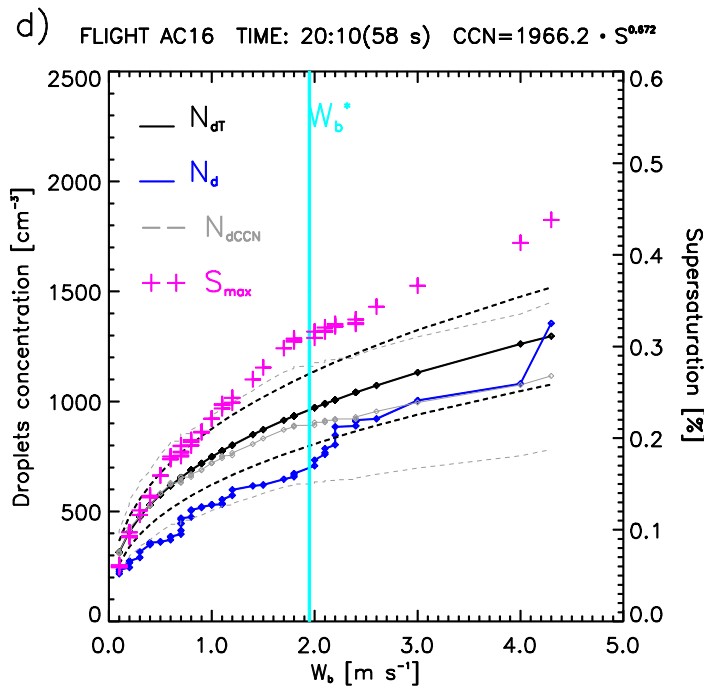



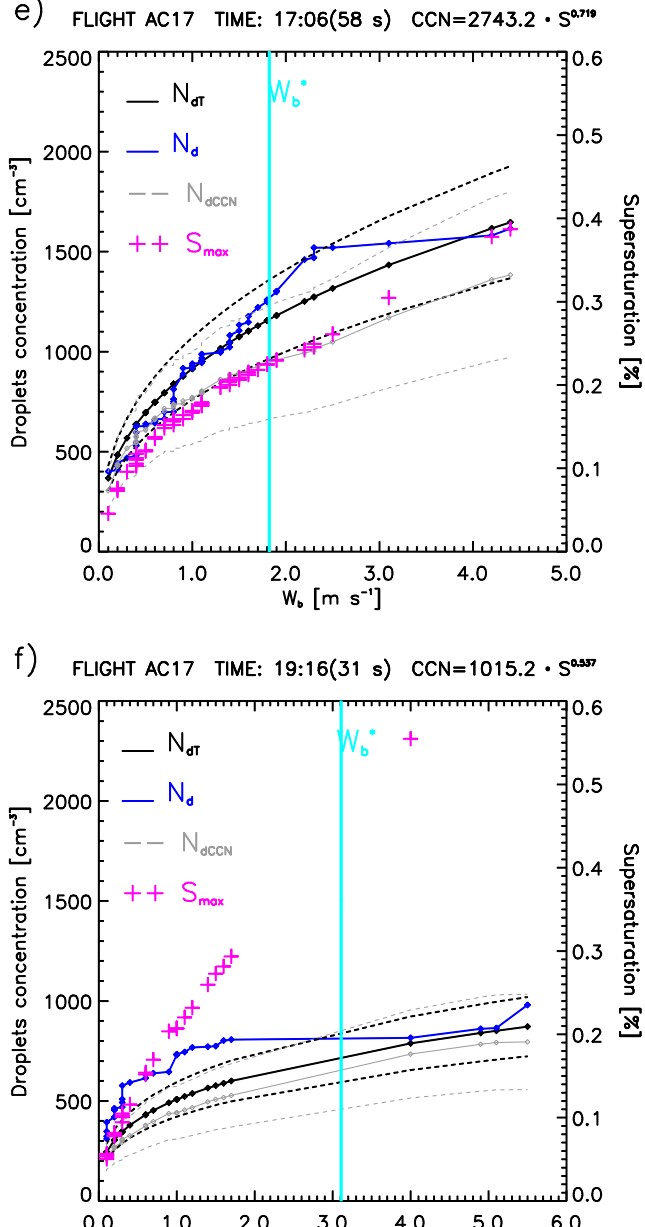

**Figure 11a-f.** $N_{dCCN}$, $S$, $N_{dT}$ and $N_d$ values are presented as a function of the cloud base updrafts ($W_b$). This plot is based on the 'probability matching method' (PMM), using same percentiles for $W_b$ and $N_d$ ($N_{dCCN}$ or $N_{dT}$). The values of $N_{dCCN}$, $N_{dT}$ and $N_d$ are shown the left y-axis, those of $S$ on the right y-axis. The black dashed lines are the $N_{dT}$ uncertainties. The gray solid (dashed) lines are the $N_{dCCN}$ values (uncertainties). The effective updraft $W_b$* for each flight segment is shown by the cyan line. The data are based on the CAS-DPOL probe. The time, period of measurements (sample size in seconds), and $CCN(S)$ equation are shown on the top of the figures.





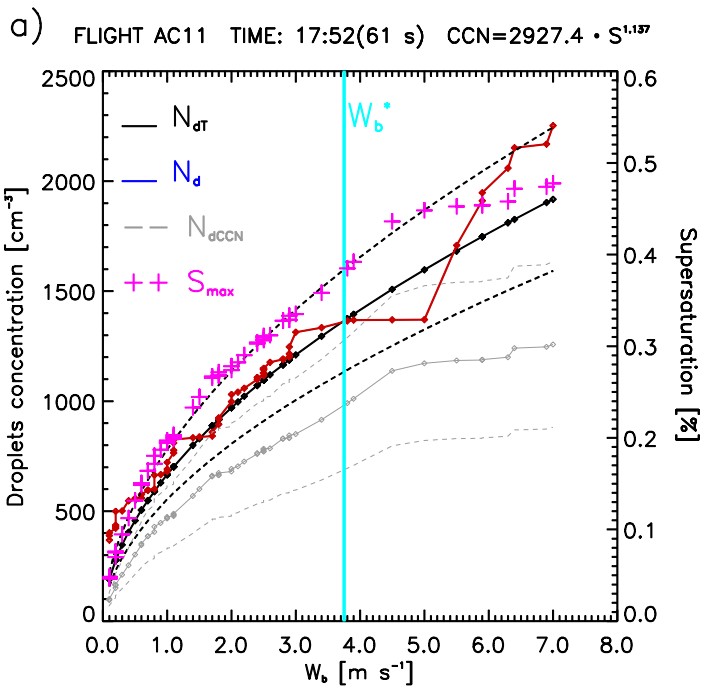

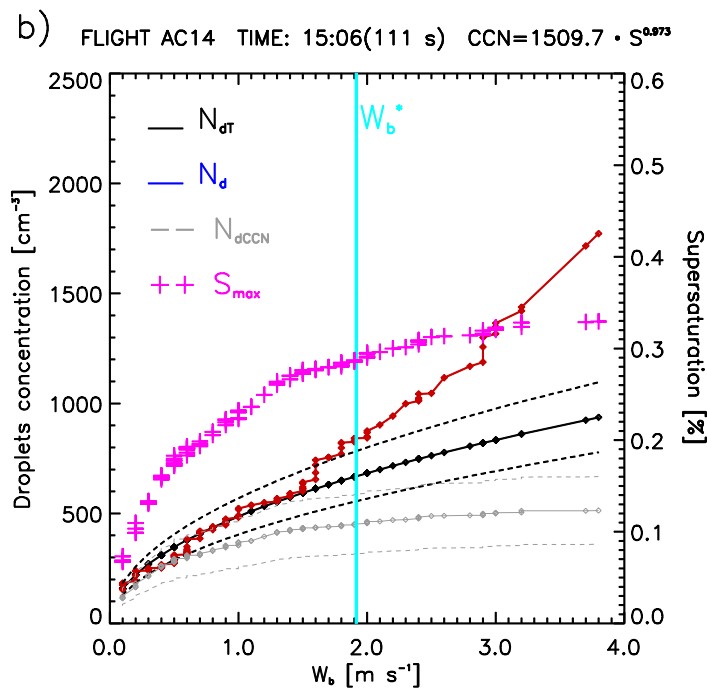



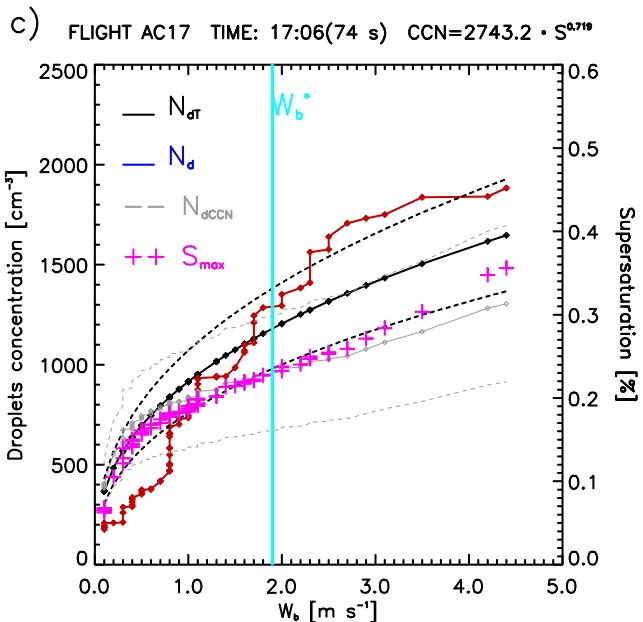


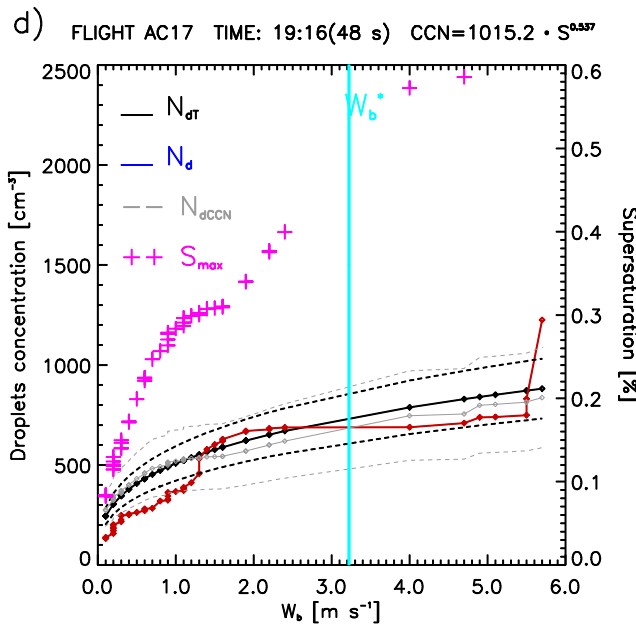

**Figure 12a-d. Same as Figure 7 for the CCP-CDP probe. No data were available for flight AC16. The CCP-CDP**
**malfunctioned in flight AC13 during the cloud base measurements.**



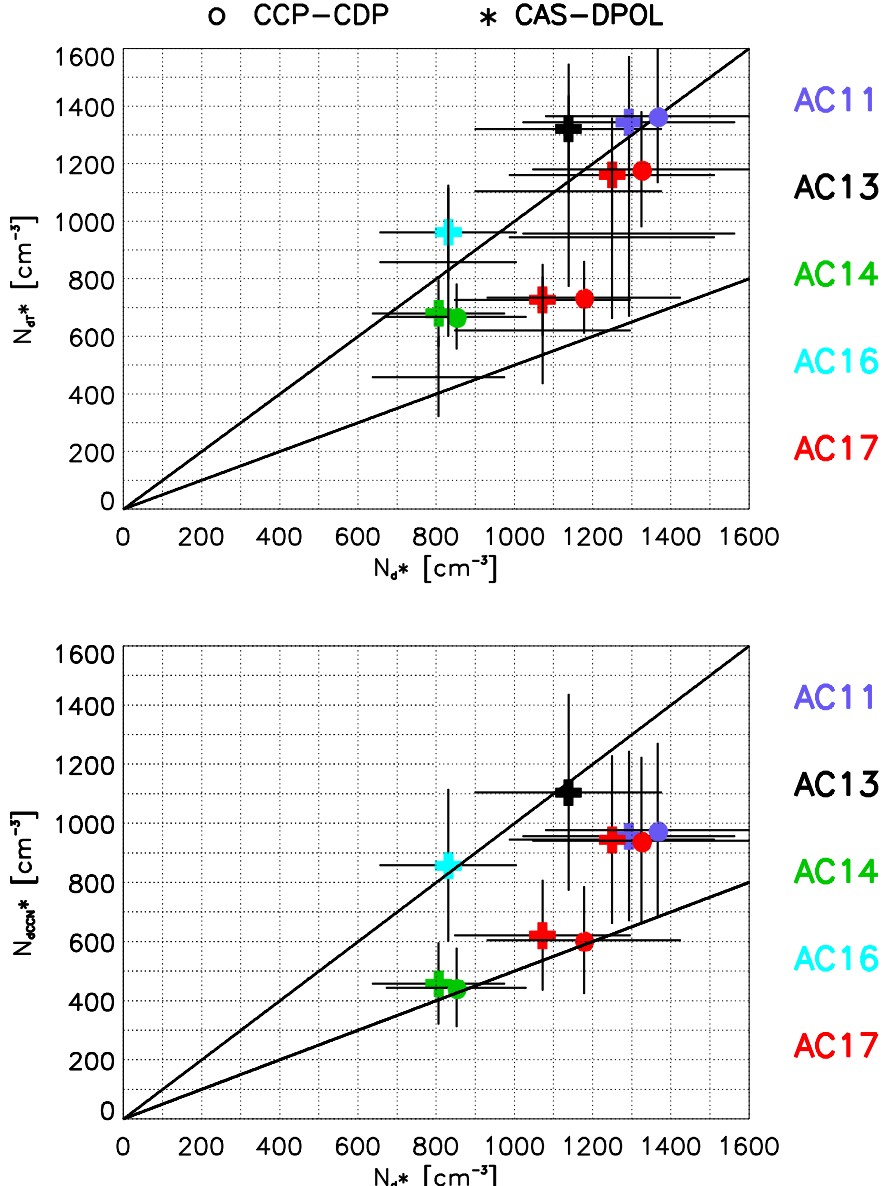

**Figure 13. a)** $N_d$* versus $N_{dT}$* **calculated with** $W_b$* **from cloud base data shown in Figures 7-8. The CAS-DPOL values are indicated by plus symbols (+) and the CCP-CDP values are indicate by circles (o). The colors indicate each flight segment (legend in the right side of the plot). Error bars indicates the uncertainties of variables estimates. Lines show the 1:1 and 1:2 relationships between** $N_{dT}$* **versus** $N_d$* **for each probe (dotted line – CCP-CDP; solid line – CAS-DPOL); b) Same for** $N_d$* **versus** $N_{dCCN}$*.



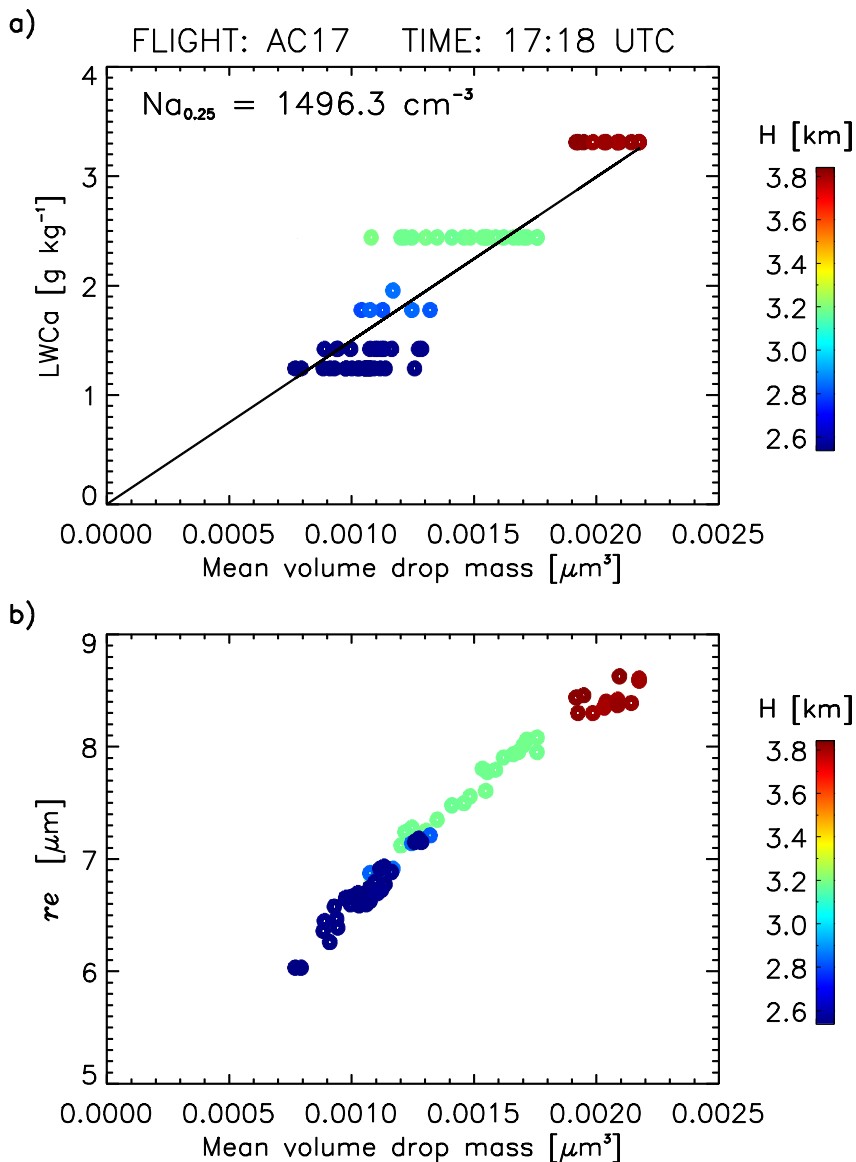


**Figure 14 a). Mean volume drop mass ($M_v$) versus liquid water content from the CCP-CDP measurements for adiabatic fraction greater than 0.25 ($LWC_a$). Vaues are shown with different colors labeled as a function of height in kilometers above sea level (indicated by the colorbar on the right side of the graphic). The slope of the linear equation is the estimated $N_a$ (i.e., 1496 cm$^{-3}$); b) $M_v$ versus re as a function of height in kilometers above sea level (indicated by the colorbar on the right side of the graphic).**




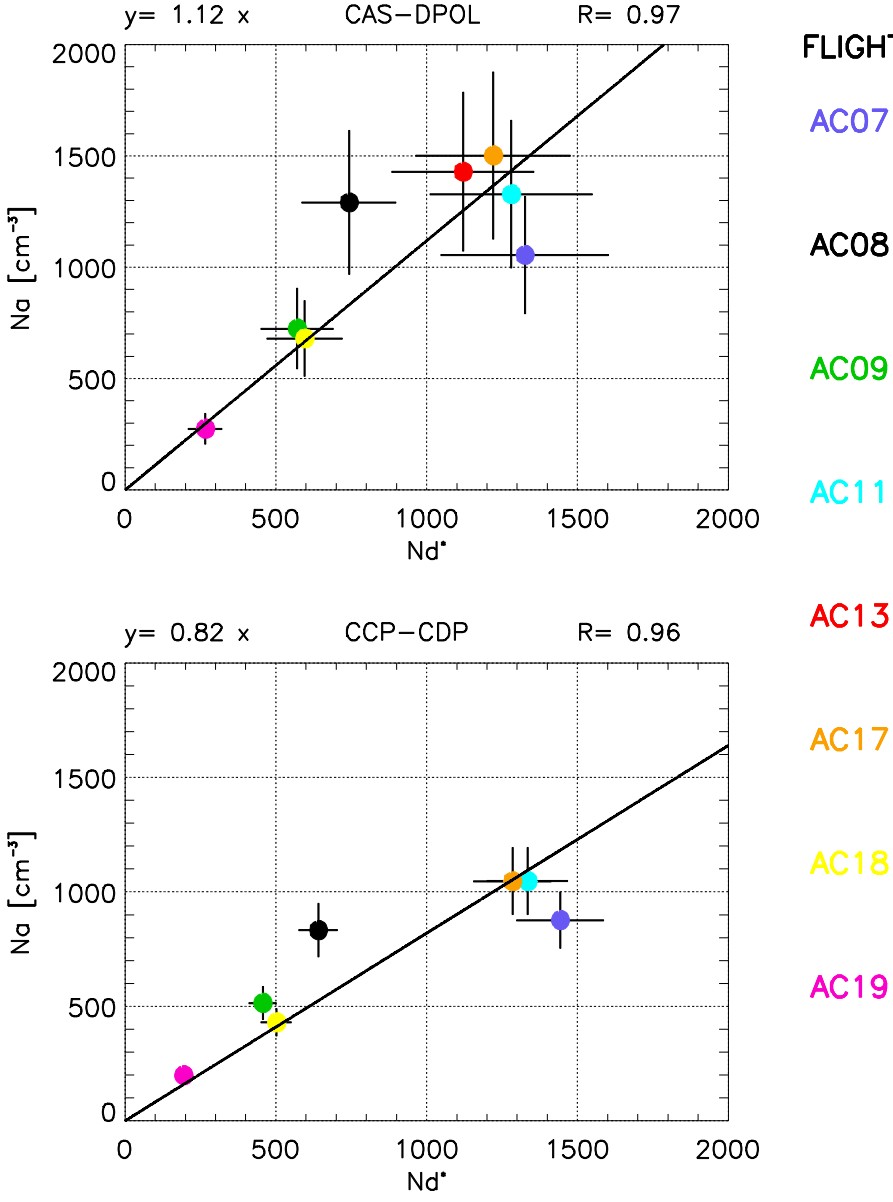

**Figure 15.** $N_d$* versus $N_a$ measured with CAS-DPOL and CCP-CDP (indicated on the top of panels) for profile flights during the ACRIDICON-CHUVA campaign. The color of the dots is associated with the flight number shown at the right side of the panels. Error bars indicates the uncertainties of variables estimates. The linear regression equation and the correlation coefficient *R* are shown in the top of each panel.