# Peer review of "Comparing calculated microphysical properties of tropical convective clouds at cloud base with measurements during the ACRIDICON-CHUVA campaign"

_Atmospheric Chemistry and Physics, 2016_

## Referee Comment (RC1) · Anonymous Referee #3 · 20 Jan 2017

Title: Comparing calculated microphysical properties of tropical convective clouds at cloud base with measurements during the ACRIDICON-CHUVA campaign

Author(s): Ramon Braga et al.

The paper compares calculated with measured microphysical properties of convective liquid clouds in the tropics.

Unfortunately, calculations are not performed within a microphysical model taking into account important spatiotemporal fluctuations of dynamical and thermodynamical properties, turbulence, entrainment, etc... In this study solely a comparison of calculating cloud properties from analytical equations and respective measurements has been performed, which represents a considerable work, however with rather limited outcome.

Conclusions of this comparison study are disappointing and do not gain new insights in liquid convective cloud microphysical processes. The paper barely presents new and noteworthy concepts. As it stands, the work is solely a rather qualitative affirmation of existing parametrizations. Taking these issues into account, the study may better carve out the uncertainties of used cloud parametrizations (equations 1, 2, 3? . . .) based on the uncertainties of measured cloud parameters from the ACRIDICON-CHUVA dataset. Also taking into account missed features and uncertainties stemming from turbulence (and more complex droplet activation) and entrainment not captured in this study. Would this be possible at all? The uncertainties of your instruments and derived measurements have been discussed rather honestly in this manuscript. This is why I encourage authors to develop this manuscript into that direction. Otherwise, I would recommend rejection of this manuscript due to its poor contribution to scientific progress.

The manuscript shows some striking and unexplained differences between calculated and measured microphysical parameters (Nd versus NdT, NdT* versus NdCCN* for a series of flights). Is this a principal problem of performed measurements within an environment of complex processes and limited degree of complexity of calculations that are hardly comparable: calculations do not capture measurement data features like turbulence, entrainment, etc...? At least above mentioned differences are more important for higher Wb values!

Specific comments related to above general statement:

Line 36: What is the impact of Wb uncertainty of 0.2 ms-1 on Nd calculation?

Line 98: What is the cumulative impact of Wb and Nd uncertainties on Smax calculation and then N0 and k?

Line 194: CDP sample area has not been calibrated before, during, after flight campaign? In this case you may not claim only 10% of uncertainty in SA?

Do you correct King probe LWC (seems not to be the case), knowing that sensitivity below 10$\mu$m and above may be 30-40 $\mu$m is reduced. You are using this probe for LWC reference, however Strapp (2003) demonstrated large deviations of King probe LWC also for larger drop diameters of 40 $\mu$m (may be already 30$\mu$m?). Your effective drop diameters reach 26 $\mu$m. . . . Uncertainty of solely 5% in LWC is difficult to believe.

Line 309: And what if King probe and CAS DPOL are both wrong and CDP is right?

Line 339: Why don't you correct CAS DPOL data for your calibrations? Consequently, in your data the CAS DPOL instrument undersizes large droplets! (40 $\mu$m in diameter appear as 35$\mu$m drops?). In case your effective diameter droplets of 26 $\mu$m would have been 30$\mu$m droplets in reality, you are underestimating LWC by 50% for these droplet sizes. . . Likewise, the King probe is underestimating LWC for other reasons as mentioned above.

Line 386-394: What is the uncertainty in N0 and k calculation and finally the uncertainty in equation (2) calculated droplet number (calculated each second) when averaging CCN2 per time step normalized by FA (with two other averages of mCCN1 per time step and TmCCN1 average of all mCCN1 time steps or may be even all CCN1 data)?

Line 400: equation (3) does not pretend Smax depending on NCCN(S). Please detail how Nd can be used to achieve a closure for NdCCN estimate.

Section 5.2.1.: Gray solid/dashed lines difficult to see in Figs 11 and 12!

Fig 11a & 11c show very weak overlap of NdCCN and NdT including both uncertainties. In addition, real Nd measurements can be considerably outside NdT uncertainties and particularly outside the overlap region. Why? What is the value of this study when measurements are not better matching the calculations with their uncertainties? Are the already large uncertainties still underestimated? Are measurements and calculations comparable in their complexity of the respective environments? I don't think so. . ..

Fig 12: Color difference of Nd curves (red) and Nd in legend (blue). Can you also show results for AC13 and AC16? Fig 12b and 12c as well as 11c show Nd that significantly exceed NdT for higher Wb. Explanation? The problem stems basically from NCCN2 calculation?

Line 513: Change 10% to at least 15% if not 20% (AC14!).

Line 513-517: and a factor of 1.5 for other cases AC11 and again AC17. Solely AC 13 and AC16 data points ok. Therefore I don't agree with that improper statement.

Line 566-567: I would call a factor of 2 in NdCCN* to Nd* comparison a pretty bad result rather than a good agreement.
* * *

---

## Referee Comment (RC2) · Anonymous Referee #1 · 27 Jan 2017

**Review of "comparing calculated cloud microphysical properties of tropical convective clouds at cloud base with measurements during the ACRIDICON-CHUVA campaign" by _Ramon Campos Braga et al._**

Braga et al., use airborne measurements aboard HALO from a CCP, CAS-DPOL and CCN counter to derive cloud drop size distributions (DSDs) and cloud water content from various instruments via an inter–comparison. In this study parameterizations for liquid cloud formation in tropical convection are validated, but for instance comparing the directly measured cloud drop concentrations ($N_d$) near cloud base to inferred values that are derived by combining the cloud base updraft velocity, CCN vs SS (supersaturation) spectra. In addition, $N_d$ from cloud base was also compared to drop concentrations ($N_a$) derived by assuming adiabatic expansion for vertical evolution of cloud drop effective radius above cloud base.

Overall, this paper presents a good summary but it lacks a significant scientific finding or discovery. Rather it is verifying previous formulated parameterizations, which is valuable. However, the authors could do a better job of comparing the differences they observe between the parameterizations validated here with previous studies.

Perhaps the paper can be re-worked to demonstrate the novelty of the work, which is lacking in the current version of the manuscript. Specific comments below should help achieve this. After such revisions have been made, the paper maybe considered for publication.

There are small editorial issues and some grammatical errors throughout the manuscript, of which I have pointed out a few, but will leave it to the authors to check that more carefully upon submission of the revised version.

**Specific comments:**

Line 29: Why not introduce CWC here like all the other acronyms in the abstract?

Line 46: "pursue" replace this word with something more suitable like "cloud microphysical models "aim" to reproduce or "The goal of cloud microphysical models is to reproduce…."

Line 137 "account" should be "accounted"

The discussion in line 132 to 137 can be expanded upon to make the paper more scientifically novel. State in more detail what was unique about these measurements, are the convective clouds here unique? Related to this but later in the paper, are the results obtained here the same as other convective regions in the world? Could the authors comment or discuss this? If indeed this is the case, that the results are similar to other locations of convection globally, the authors may consider discussing this point and stressing this aspect.

Line 149: should read "Manaus City" not "Manaus city"

Line 193-194: Delete "was used additionally considering" and Line 194: add "was considered" after 10%. In total the sentence should read "For the CDP sample area of 0.22 mm$^2$, an uncertainty of about 10% was considered (Molleker et al., 2014)."

Line 205: Delete extra periods

Line 267: "maximal" should be "maximum"

Line 269: should "probes" have an apostrophe after it i.e. probes'? it sounds like it is being used in the possessive.

Line 297: Why these specific flights being used (AC08 and AC20) for the CWC, why not data from the entire campaign? Also, why not use the same flights as were used in the effective radius comparison (line 278)?

Line 309-314: Why compare only with one hot wire probe when three of them were on board the aircraft?

Line 319: insert "the" before "hot-wire"

Line 320: Can you make it clearer that this is a decreasing number concentration with *increasing* effective radius

Line 322: insert "the" ahead of "CAS-DPOL" in general the grammar is really poor from lines 320-325, please rectify

Line 326-333: Why not consider using only particles less than 40 microns in your CWC comparison?

Line 406: replace "greater than or equal" with the symbol "≥"

Line 471-479: Are these values presented here similar to literature values from other locations in the world? Can there be a comparison and discussion of this?

Line 520: Figure 14a shows the LWC? The Na that is stated in Figure 14a is also mentioned here in Line 523, not sure why the reference to Figure 14a is needed here.

Line 530-534: The scaling of 1.3 works quite well, perhaps mention it here since this is a new data set.

Line 558: Here the authors should make a case for why their work was novel, interesting or what is new about their work.

Line 570-574: Was there any doubt about the validity of the parameterization prior to this study? What is new about the work here other than the fact that the measurements were all taken during this campaign on one/the same aircraft?

Figures

Fig 4: Consider editing the plot so that the legend matches the sub=plot where the quantities are shown

Fig 4 (lower left panel for CWC): Why is it necessary to have a log scale? The data just cover one order of magnitude and are all squeezed to the bottom half of the panel. There is no need for the scale to extend to 10. And no need for a log scale either. This artificially downplays some of the differences between the probes.

Fig 6a and 6b: is it necessary to have zeros in front of the micron sizes, i.e. 05 instead of 5. Also, can both scales be made linear for consistency and clarity? It is hard to compare presented in the manner here.

Line 1134: Italicize "m"

Fig 12 (all panels): Shouldn't $N_d$ be in red?

Fig 13 (line 1190): reference to Fig 7-8 is not consistent with text, should be Figure 11 and 12

---

## Referee Comment (RC3) · Anonymous Referee #2 · 10 Feb 2017

The Manuscript is a little vague in its objectives, but it appears that it is attempting to validate aircraft observations by performing closure with either different instruments or between CCN measurements made at different supersaturations and cloud droplet numbers measured at different updraft speeds. It does this using below cloud and in cloud measurements made during the ACRIDICON-CHUVA campaign and combining these with activation models.

The other reviewers have already made comments regarding the models used so I will focus here mostly on the measurements and the analysis that goes along with those.

Unfortunately this paper needs significant extra work in order to make it of publishable quality. However I think the type of analysis that has been performed here is valuable and is not undertaken enough. This is the type of paper that can be used to assure the quality of the measurements being made and that other papers in the project can reference to avoid repeating this analysis by multiple groups and authors. It is also the type of paper that can highlight the limits of the instruments. This is good as it can provide insight to a modeller who is using the data perhaps without an in-depth knowledge of its limits and it also means that it becomes clear what science cannot be performed with the data and therefore where we need to improve our instruments, calibration methods and analysis techniques. However the work is only valuable in this sense if the analysis is performed in an incredibly rigorous manner. I applaud the author's attempt to write this paper, but I would suggest that he needs to pull in more input from coauthors - there are many well respected coauthors on the paper and I am surprised that their instrument knowledge does not show through in this paper. There are certainly other people who work full time within the aircraft instrument community who could have input.

I would suggest that the manuscript needs a full rewrite and I would suggest that the author goes back to basics in terms of deciding exactly what the objectives are (are they validating instruments or validating the cloud models), then doing a thorough uncertainty analysis of the instruments. This must include details of calibration methods used and the uncertainty derived from those, plus things that cannot or have not been calibrated and the reason why and what the expected uncertainty for these things might be. Based on the uncertainty analysis the author can then decide if the objectives are achievable and can present appropriate uncertainties in the conclusions. Based on the general comments above I recommend the paper be rejected in its current form. Some more detailed comments follow

Introduction - In general the author should be familiar with the calibration methods used with the instruments used and this should be reflected in the references.

line 75 - Previous analysis (Strapp et al 1992, Journal of Atmospheric and Oceanic Technology Vol 9 p 548) has indicated that the PCASP dries it's sample through ram heating during measurements. The author should familiarise himself with this work, and understand why this drying may not be happening.

Line 85 - If CCN measured at constant S is not constant then either N0 or k in (1) are changing. I.e., either the total number is changing but everything else remains the same or the hygroscospicity or size distribution of the aerosol is changing. Or of course there could be a combination of these factors. The author must show which are occurring.

Line 95-100 - Total number totally cancels from effective radius calculations and adiabatic calculations reveal expectations for mass of condensed water not number. Dividing adiabatic water content by measured mass per particle would give a number concentration but even in an adiabatic regime the uncertainty on this would be larger than the measured droplet number concentration. Calibrations on the CDP operated by FAAM using the method described by Rosenberg et al 2012 (Atmospheric Measurement Techniques vol 5 p1147) provides an uncertainty around 0.5 um in sizing, but typically shows a discrepancy of around 2 um from the manufacturer's specification. If the manufacturer spec is used in this work then we can expect that at 20 um we have approximately 30% uncertainty in mass per particle measurements.

Line 190 - Has the collecting angle of the instruments been measured? This defines the location of the Mie wiggles and where the bins should be merged.

Line 246-260 - As described previously this assumes k is constant, the author needs to provide evidence this is a good assumption. The correction method means that we are correcting to a point where N0 is equal to the average N0 for the scan. This should be made clear and an estimate of how much N0 is varying must be made as this impacts how much confidence we have in a model's estimate of Nd in cloud.

Line 261-269 - When I first read this seemed entirely circular. Later it becomes clear

that this is the point. We are putting observations into a model and checking for consistency. The author should highlight in the aims of the paper that they intend to do this so that the reader knows to expect this. A better way to represent this may be a plot f Nd vs S with data points taken from measurements and derived through equation 3 (perhaps coloured by w) along with points from the scanning and static CCN instrument. If the model is correct and the obs are consistent then all points should fall on one line.

Lines284-288 This probability matching method assumes droplet number is a monatonic function of w only. I have no issue with the monotonic assumption, but the author should show that there is no other influences upon drop concentration such as entrained dry/clean air and constant aerosol/ccn concentration below cloud or at least state why this is a good assumption.

Lines 307-325 This needs a thorough uncertainty analysis to show its usefulness as described earlier.

Line 350 - You are claiming an uncertainty of 5% in N0, but as described earlier this is in the average N0 over the scan. We have seen CCN number on the constant supersaturation instrument vary from ~650 to 950 cm-3 so it seems unreasonable to claim 5% uncertainty in this parameter. . This ambiguity comes from not being clear in the first instance about what you are trying to measure. In reality I think an estimate of k is what you should be aiming for as N0 is clearly changing and is not a constant. Line 380-390 I certainly would not be alone in suggesting that the phrase "agree closely" and similar variations has very little place in scientific work. In this case there is a difference of up to 70% in fig 9a. Phrases such as "agree within the measurement uncertainties," "differ by up to x amount," or "agree to the extent that conclusion y is unaffected" are all appropriate, but "agree closely" is entirely subjective.

Line 401 - Another "good agreement" statement. Points here deviate from the 1:1 line by up to a factor of 2.

Line 440-444 - This difference is almost certainly within the expected uncertainty which

as described above is probably 30% from the mass per particle measurement, plus perhaps 10-20% from sample area and air speed through the sample volume estimates.

447-450 and figs 13/14 - I see size distributions like this all the time and often by people who work with these instruments a lot. They are unfortunately not really appropriate styles for plotting size distributions. The following changes should be made. The plot should show points and not lines. It is not appropriate to "join the dots" on a plot that has significant uncertainties. Each point should have an x and y uncertainty. Standard error is not an appropriate uncertainty to use. It assumes that we measure the same thing repeatedly and that the uncertainty is dominated by noise. Here we have concentrations that vary with time during and between the periods that contribute to these average size distributions. So the standard error becomes some combination of noise and variability and omits all systematic uncertainties. Instead the author should do a proper error analysis including contributions from sample area, air speed at the probe, bin width and counting (Poisson) uncertainty for the y error and sizing uncertainty for the x error.

Line 454-560 The sensitivity is probably not the issue, it is more likely to be the bin widths for which we see no calibration.

---

## Author Comment (AC1) · 20 Mar 2017

Review of "comparing calculated cloud microphysical properties of tropical convective clouds at cloud base with measurements during the ACRIDICON-CHUVA campaign" by Ramon Campos Braga et al.

Anonymous Referee #1

Braga et al., use airborne measurements aboard HALO from a CCP, CAS-DPOL and CCN counter to derive cloud drop size distributions (DSDs) and cloud water content from various instruments via an inter–comparison. In this study parameterizations for

liquid cloud formation in tropical convection are validated, but for instance comparing the directly measured cloud drop concentrations (Nd) near cloud base to inferred values that are derived by combining the cloud base updraft velocity, CCN vs SS (supersaturation) spectra. In addition, Nd from cloud base was also compared to drop concentrations (Na) derived by assuming adiabatic expansion for vertical evolution of cloud drop effective radius above cloud base. Overall, this paper presents a good summary but it lacks a significant scientific finding or discovery. Rather it is verifying previous formulated parameterizations, which is valuable. However, the authors could do a better job of comparing the differences they observe between the parameterizations validated here with previous studies. Perhaps the paper can be re-worked to demonstrate the novelty of the work, which is lacking in the current version of the manuscript. Specific comments below should help achieve this. After such revisions have been made, the paper maybe considered for publication. There are small editorial issues and some grammatical errors throughout the manuscript, of which I have pointed out a few, but will leave it to the authors to check that more carefully upon submission of the revised version.

General comments

The authors thank the referee for the general comments and advices. Furthermore, the advices of the referee are highly appreciated as well as the very valuable and constructive suggestions to increase the quality of the manuscript. We tried to address the points requested by the reviewer to the paper be considered for publication. Overall, we have improved the focus of the paper highlighting our objectives and the novelty of our study.

The answers for specific comments are available after "A:" for each specific line.

Specific comments:

Line 29: Why not introduce CWC here like all the other acronyms in the abstract?

[Figure]

A: OK. Changed.

Line 46: "pursue" replace this word with something more suitable like "cloud micro-physical models "aim" to reproduce or "The goal of cloud microphysical models is to reproduce. . .."

A: OK. Changed.

Line 137 "account" should be "accounted"

A: OK. Changed.

The discussion in line 132 to 137 can be expanded upon to make the paper more scientifically novel. State in more detail what was unique about these measurements, are the convective clouds here unique? Related to this but later in the paper, are the results obtained here the same as other convective regions in the world? Could the authors comment or discuss this? If indeed this is the case, that the results are similar to other locations of convection globally, the authors may consider discussing this point and stressing this aspect.

A: The text was changed to address these comments. Thanks.

New text:

"The availability of these measurements collected by the same aircraft provides a unique opportunity to compare the data with model predictions and to test the sensitivity of the results to the differences between the measurements by the cloud probes.

This study is novel in several aspects:

It is the first study that validates the methodology of retrieving the adiabatic cloud drop concentrations Na (Freud et al., 2011) from the vertical evolution of re while assuming that re is nearly adiabatic. This is important because it supports the validity of retrieving Na from satellite-retrieved vertical profile of re (Rosenfeld et al., 2014a and 2016).

[Figure]

It is the first study that tests with aircraft the measured Nd with its parameterization that is based on NCCN(S) along with cloud base spectrum of updrafts weighted by the updraft speed itself, Wb*. It is done this way to be compatible with the recently developed methodology of retrieving CCN from satellites by means of retrieving Nd and Wb* (Rosenfeld et al., 2016).

It is the first study that compares observationally the old Twomey (1959) parameterization of the dependence of Nd on Wb (Eq. 2) versus the recent Pinsky et al. (2012) analytical expression for the same (Eq. 3)."

References:

Freud, E., Rosenfeld, D. and Kulkarni, J. R.: Resolving both entrainment-mixing and number of activated CCN in deep convective clouds, Atmos. Chem. Phys., 11(24), 12887–12900, doi:10.5194/acp-11-12887-2011, 2011.

Pinsky, M., Khain, A., Mazin, I. and Korolev, A.: Analytical estimation of droplet concentration at cloud base, J. Geophys. Res. Atmos., 117(17), 1–14, doi:10.1029/2012JD017753, 2012.

Rosenfeld, D., Fischman, B., Zheng, Y., Goren, T. and Giguzin, D.: Combined satellite and radar retrievals of drop concentration and CCN at convective cloud base, Geophys. Res. Lett., 41(9), 3259–3265, doi:10.1002/2014GL059453, 2014a.

Rosenfeld D., Y. Zheng, E. Hashimshoni, M. L. Pöhlker, A. Jefferson, C. Pöhlker, X. Yu, Y. Zhu, G. Liu, Z. Yue, B. Fischman, Z. Li, D. Giguzin, T. Goren, P. Artaxoi, H. M. J. Barbosai, U. Pöschl, and Meinrat O. Andreae, 2016: Satellite retrieval of cloud condensation nuclei concentrations by using clouds as CCN chambers. Proceedings of the National Academy of Sciences, doi:10.1073/pnas.1514044113.

Twomey, S.: The nuclei of natural cloud formation part II: the supersaturation in natural clouds and the variation of cloud droplet concentration, Geofis. Pura e Appl., 43(1), 243–249, doi:10.1007/BF01993560, 1959.

[Figure]

Line 149: should read "Manaus City" not "Manaus city"

A: OK. Changed.

Line 193-194: Delete "was used additionally considering" and Line 194: add "was considered" after 10%. In total the sentence should read "For the CDP sample area of 0.22 mm2, an uncertainty of about 10% was considered (Molleker et al., 2014)."

A: OK. Changed.

Line 205: Delete extra periods

A: OK. Changed.

Line 267: "maximal" should be "maximum"

A: OK. Changed.

Line 269: should "probes" have an apostrophe after it i.e. probes'? it sounds like it is being used in the possessive.

A: OK. Changed.

Line 297: Why these specific flights being used (AC08 and AC20) for the CWC, why not data from the entire campaign? Also, why not use the same flights as were used in the effective radius comparison (line 278)?

A: The sentence is wrong. We used all flights, except AC07 which we have not hot-wire data. We have corrected the sentence.

New text: "The mean values of CWC estimated from the probes from flights AC08 to AC20 (AC07 had no hot-wire CWC data) and altitudes between 600 m and 5,000 m generally show an increase with increasing re."

Line 309-314: Why compare only with one hot wire probe when three of them were on board the aircraft?

A: There was only one hotwire operational aboard HALO. This one was mounted on the CAS-DPOL. Some DMT instruments are delivered with hot-wires. The CCP was indeed equipped with the hot-wire as well, but we do not operate it during flight as then the CCP's overall power consumption would exceed the limits. It was physically disconnected and no part of CCP anymore.

Line 319: insert "the" before "hot-wire"

A: OK. Changed.

Line 320: Can you make it clearer that this is a decreasing number concentration with increasing effective radius

A: OK. Changed.

Line 322: insert "the" ahead of "CAS-DPOL" in general the grammar is really poor from lines 320-325, please rectify

A: The text was rewritten.

Line 326-333: Why not consider using only particles less than 40 microns in your CWC comparison?

A: The amount of CWC above 40 $\mu$m is negligible. However, it illustrates the difference in the shapes of the DSDs.

Line 406: replace "greater than or equal" with the symbol "$\geq$"

A: OK. Changed.

Line 471-479: Are these values presented here similar to literature values from other locations in the world? Can there be a comparison and discussion of this?

A: Ok. We added a new text:

"For some flights the values estimated for N0 and k parameters of Eq. 1 are similar to what was found by Pöhlker et al., (2016) for ground measurements (N0 = 1469 $\pm$ 78

and k = 0.36 ± 0.06) during the dry season in the Amazon. However, in the majority of the cases N0 and k are twice or three times greater than the values from Pöhlker et al., (2016). These differences are probably related to flying selectively to areas that had high aerosol concentrations to contrast the cloud behavior with the flights with low aerosol concentrations, as shown in Fig. 2. The high CCN measured in this study is more similar to previous aircraft measurements in smoky conditions over the Amazon (Andreae et al., 2004; Freud et al., 2008)."

References

Andreae, M. O., Rosenfeld, D., Artaxo, P., Costa, A. A., Frank, G. P., Longo, K. M. and Silva-Dias, M. A. F.: Smoking rain clouds over the Amazon, Science, 303(5662), 1337–42, doi:10.1126/science.1092779, 2004.

Freud, E., Rosenfeld, D., Andreae, M. O., Costa, A. A. and Artaxo, P.: Robust relations between CCN and the vertical evolution of cloud drop size distribution in deep convective clouds, Atmos. Chem. Phys., 8(6), 1661–1675, doi:10.5194/acp-8-1661-2008, 2008.

Pöhlker, M.L., Pöhlker, C., Ditas, F., Klimach, T., Hrabe de Angelis, I., Araújo, A., Brito, J., Carbone, S., Cheng, Y., Chi, X. and Ditz, R., 2016. Long-term observations of cloud condensation nuclei in the Amazon rain forest–Part 1: Aerosol size distribution, hygroscopicity, and new model parametrizations for CCN prediction. Atmospheric Chemistry and Physics, 16(24), pp.15709-15740.

Line 520: Figure 14a shows the LWC? The Na that is stated in Figure 14a is also mentioned here in Line 523, not sure why the reference to Figure 14a is needed here.

A: Yes, LWC for adiabatic fraction > 0.25. A: Ok.

Line 530-534: The scaling of 1.3 works quite well, perhaps mention it here since this is a new data set.

A: Ok. Thanks.

Line 558: Here the authors should make a case for why their work was novel, interesting or what is new about their work.

A: The conclusions now clearly show the novel aspects of this study.

New text:"This study is focused on testing novel parameterizations that are used for the recently developed methodology of satellite retrievals of Na, Wb*, and CCN in convective clouds, based on aircraft measurements during the ACRIDICON-CHUVA campaign in the Amazon. It is the first time that these new parameterizations are tested com-prehensively alongside old parameterizations. Liquid water content measurements from a hot-wire device were taken as a reference for the quality assessment of estimated CWC from cloud probe DSDs near cloud base. The intercomparison of the DSDs and the CWC derived from the different instruments generally shows good agreement within the instrumental uncertainties. The values of Nd near cloud base were comparable within the measurement errors with their inferred values based on the measured Wb* and NCCN(S). The values of Wb* were calculated from the measured spectrum of W* using the parameterization of Rosenfeld et al. (2014a), which is also used for retrieving cloud base updraft from satellites (Zheng et al., 2015). In addition, Nd near cloud base was favorably (within $\pm20\%$) compared with Na, obtained from the vertical evolution of cloud drop effective radius (re) above cloud base. The values of Na in this study were obtained with the same parameterization that has been recently developed for satellite calculated Na based on the satellite retrieved vertical evolution of re in convective clouds (Freud et al., 2011; Rosenfeld et al., 2014a). These results support the methodology to derive Na based on the rate of re growth with cloud depth and under the assumption that the entrainment and mixing of air into convective clouds is extremely inhomogeneous.

The measured effective droplet numbers (Nd*) at cloud base were also compared against NdT* which is its predicted value based on the old parameterization in Eq. 2 (Twomey, 1959), which uses Wb* and the NCCN(S) power law. A newer parameterization calculates NdCCN* by substituting S into the power law NCCN(S), where S

is obtained from Eq. 3 (Pinsky et al., 2012). The agreement between Nd* and Nd-CCN* was only within a factor of 2, underlying the yet unresolved challenge of aircraft measurements of S in clouds.

In summary, the measurements of NCCN(S) and Wb did reproduce the observed Nd. when using Twomey's parameteri-zation, while using measured in cloud S remains a challenge. Furthermore, the vertical evolution of re with height reproduced the observation-based adiabatic cloud base drop concentrations, Na. The combination of these results provide aircraft observational support for the various components of the satellite retrieval methodology that was recently developed to retrieve NCCN(S) below the base of convective clouds (Rosenfeld et al., 2016). This parameter-ization can now be applied more confidently and with the proper qualifications to cloud simulations and satellite retrievals."

Line 570-574: Was there any doubt about the validity of the parameterization prior to this study? What is new about the work here other than the fact that the measurements were all taken during this campaign on one/the same aircraft?

A: Most of the tested parameterizations are new, and it is the first time that they are tested comprehensively alongside old parameterizations.

Figures Fig 4: Consider editing the plot so that the legend matches the sub=plot where the quantities are shown

Fig 4 (lower left panel for CWC): Why is it necessary to have a log scale? The data just cover one order of magnitude and are all squeezed to the bottom half of the panel. There is no need for the scale to extend to 10. And no need for a log scale either. This artificially downplays some of the differences between the probes.

A: We changed the Figure 4 as you suggest. It is available at supplementary material.

Fig 6a and 6b: is it necessary to have zeros in front of the micron sizes, i.e. 05 instead of 5. Also, can both scales be made linear for consistency and clarity? It is hard to

compare presented in the manner here. A: The others referees also have requested some changes in figures 6. We have changed the figures 6 and captions to address the suggestions. Following these suggestions we have changed the DSDs figures using histograms of binned detection channels. The data points are shown with size bin limits in x-direction (to cover the Mie-ambiguity ranges, providing an approach to have the channel-wise sizing error superimposed by the size-bin limits) and uncertainty in y-direction. We also eliminate the zeros in front of micron sizes. New figures 6 are available at supplementary material.

Line 1134: Italicize "m" A: Ok. Thanks.

Fig 12 (all panels): Shouldn't Nd be in red? A: Yes. Thanks.

Fig 13 (line 1190): reference to Fig 7-8 is not consistent with text, should be Figure 11 and 12 A: Yes. Thanks.

Please also note the supplement to this comment:
http://www.atmos-chem-phys-discuss.net/acp-2016-872/acp-2016-872-AC1-
supplement.pdf

―――――――――――――――――

**Supplement:**

[Figure]

**Figure 4. Mean cloud water content from the hot-wire measurements and estimated from the cloud probes (CCP-CDP and CAS-DPOL from top to bottom, respectively) as a function of effective radius ($r_e$) size (left panel). The ratios between the hot-wire liquid water content and the cloud water content derived from each probe are shown in red (*CWCr*). The total uncertainty for each probe and the hot-wire measurements are shown by the dotted lines. The number of cases (black continuous line), hot-wire measurement standard deviations (dashed black line), and probe CWC standard deviations (dashed colored line) for each $r_e$ size are shown in the right panels.**

[Figure]

[Figure]

**Figure 6.** (left) Mean cloud droplet concentration (solid lines) and (right) cloud water content as a function of droplet diameter in the left and right panels, respectively, for a) 5 μm < $r_e$ < 6 μm; b) 8 μm < $r_e$ < 9 μm; c) 11 μm < $r_e$ < 12 μm; d) 12 μm < $r_e$ < 13 μm. The probes are identified by colors as shown at the top of the panels. The error bars indicate the uncertainty range of mean cloud droplet concentration and cloud water content values as a function of droplet diameter.

---

## Author Comment (AC3) · 20 Mar 2017

Title: Comparing calculated microphysical properties of tropical convective clouds at

cloud base with measurements during the ACRIDICON-CHUVA campaign.

Author(s): Ramon Braga et al.

The paper compares calculated with measured microphysical properties of convective liquid clouds in the tropics. Unfortunately, calculations are not performed within a microphysical model taking into account important spatiotemporal fluctuations of dynamical and thermodynamical properties, turbulence, entrainment, etc... In this study solely a comparison of calculating cloud properties from analytical equations and respective measurements has been performed, which represents a considerable work, however with rather limited outcome. Conclusions of this comparison study are disappointing and do not gain new insights in liquid convective cloud microphysical processes. The paper barely presents new and noteworthy concepts. As it stands, the work is solely a rather qualitative affirmation of existing parameterizations. Taking these issues into account, the study may better carve out the uncertainties of used cloud parameterizations (equations 1, 2, 3? . . .) based on the uncertainties of measured cloud parameters from the ACRIDICONCHUVA dataset. Also taking into account missed features and uncertainties stemming from turbulence (and more complex droplet activation) and entrainment not captured in this study. Would this be possible at all? The uncertainties of your instruments and derived measurements have been discussed rather honestly in this manuscript. This is why I encourage authors to develop this manuscript into that direction. Otherwise, I would recommend rejection of this manuscript due to its poor contribution to scientific progress.

The manuscript shows some striking and unexplained differences between calculated and measured microphysical parameters (Nd versus NdT, NdT* versus NdCCN* for a series of flights). Is this a principal problem of performed measurements within an environment of complex processes and limited degree of complexity of calculations that are hardly comparable: calculations do not capture measurement data features like turbulence, entrainment, etc...? At least above mentioned differences are more important for higher Wb values!

General comments

The authors thank the referee for the general comments and advices. Furthermore, the advices of the referee are highly appreciated as well as the very valuable and constructive suggestions to increase the quality of the manuscript. However, we disagree with the referee affirmation which highlight that the study have a poor contribution to scientific progress. As mentioned before, the objective of the paper is to validate the physical parameterizations connecting between CCN(S), Wb, Nd and Na, when tested over the important convective regime of the Amazon. This is the first paper that tests all parameterizations proposed for convective clouds regime against each other with the same dataset.

Furthermore, the parameterizations are far from being old. Rather, as now stated in the first paragraph of the abstract: "The objective of this study is to validate novel parameterizations that were recently developed for satellite retrievals of CCN(S) at cloud base alongside with more traditional parameterizations connecting CCN(S) with cloud base updrafts and drop concentrations."

The calculations of the effective updraft speed at cloud base (Wb*) provide a new capability for test the parameterizations proposed in the study and to ascribe the capability of Na estimates for convective clouds developed at different aerosol conditions over Amazon. The role idea of using Wb* is already accounting for the air turbulence (see Rosenfeld et al., 2014a) and the Na estimates for convective clouds also account for entrainment as is described at section 4.4 (also discussed at Freud et al., 2011). The study supports the use of Na estimates using satellite data of effective radius vertical profiles at convective clouds over the Amazon basin.

Regarding the possibility of use other parameterizations in our analysis, we have already highlight at introduction section some issues due to the unreasonable measurements of low hygroscopic factor values below cloud base, which prevent us to test other types of parameterizations (e.g. k-Köhler model estimates) to validate Nd at

cloud base.

In summary, the following text was added to the introduction: "This study is novel in several aspects:

a. It is the first study that validates the methodology of retrieving the adiabatic cloud drop concentrations Na (Freud et al., 2011) from the vertical evolution of re while assuming that re is nearly adiabatic. This is important because it supports the validity of retrieving Na from satellite-retrieved vertical profile of re (Rosenfeld et al., 2014a and 2016).

b. It is the first study that tests with aircraft the measured Nd with its parameterization that is based on supersaturation spectrum of CCN along with cloud base spectrum of updrafts, Wb*. It is done this way to be compatible with the recently developed methodology of retrieving CCN from satellites by means of retrieving Nd and Wb* (Rosenfeld et al., 2016).

c. It is the first study that compares observationally the old Twomey (1959) parameterization of the dependence of Nd on Wb (Eq. 2) versus the recent Pinsky et al. (2012) analytical expression for the same (Eq. 3)."

We agree to better carve out the uncertainties of used cloud parameterizations, which is available at the new version of the manuscript.

The uncertainties of Wb of HALO were recalculated for all campaign data. In the new version of the manuscript we ascribe the uncertainties of theoretical estimates assuming 0.3 ms-1 as the Wb uncertainty. This new number changes the values of variables uncertainties used for NdT, Smax and NdCCN estimates.

References:

Freud, E., Rosenfeld, D. and Kulkarni, J. R.: Resolving both entrainment-mixing and number of activated CCN in deep convective clouds, Atmos. Chem. Phys., 11(24), 12887–12900, doi:10.5194/acp-11-12887-2011, 2011.

Pinsky, M., Khain, A., Mazin, I. and Korolev, A.: Analytical estimation of droplet concentration at cloud base, J. Geophys. Res. Atmos., 117(17), 1–14, doi:10.1029/2012JD017753, 2012.

Rosenfeld, D., Fischman, B., Zheng, Y., Goren, T. and Giguzin, D.: Combined satellite and radar retrievals of drop concentration and CCN at convective cloud base, Geophys. Res. Lett., 41(9), 3259–3265, doi:10.1002/2014GL059453, 2014a.

Rosenfeld D., Y. Zheng, E. Hashimshoni, M. L. Pöhlker, A. Jefferson, C. Pöhlker, X. Yu, Y. Zhu, G. Liu, Z. Yue, B. Fischman, Z. Li, D. Giguzin, T. Goren, P. Artaxoi, H. M. J. Barbosai, U. Pöschl, and Meinrat O. Andreae, 2016: Satellite retrieval of cloud condensation nuclei concentrations by using clouds as CCN chambers. Proceedings of the National Academy of Sciences, doi:10.1073/pnas.1514044113.

Twomey, S.: The nuclei of natural cloud formation part II: the supersaturation in natural clouds and the variation of cloud droplet concentration, Geofis. Pura e Appl., 43(1), 243–249, doi:10.1007/BF01993560, 1959.

Specific comments

Line 36: What is the impact of Wb uncertainty of 0.2 ms-1 on Nd calculation?

A: The Wb uncertainty impacts on average for about 65% on NdT uncertainty.

Line 98: What is the cumulative impact of Wb and Nd uncertainties on Smax calculation and then N0 and k?

A: The cumulative impact on Smax is 22 % on average. N0 and k is not used on Smax calculation, just Wb, Nd and temperature and pressure of cloud base (used on coefficient C estimate at equation 2).

A: We have changed the manuscript to address these questions as follow: "The uncertainties regarding the Smax, NdCCN and NdT estimates for measurements at cloud base with both probes (CCP-CDP and CAS-DPOL) are on average about 22, 20 and

38 % for all flights, respectively (the uncertainty method adopted for these theoretical estimates are available at Appendix A). The Wb uncertainty of 0.3 m s-1 impacts on average for about 65% (60 %) on NdT (Smax) uncertainty, and the uncertainty from the estimated Smax contributes for most of NdCCN uncertainty (∼70% on average)."

Line 194: CDP sample area has not been calibrated before, during, after flight campaign? In this case you may not claim only 10% of uncertainty in SA?

A: The CDP sample area has been frequently measured revealing by the way 0.27mm$^2$ (not 0.22mm$^2$, which should by a typo or an outdated value and must be corrected in the text) with an uncertainty of 10%. The uncertainty +/- 0.03mm$^2$ results from repeated measurements. Unless there is no massive manipulation/disarrangement of the CDP's optics or a detectable aging of the laser diode, the sample area remains stable even if the instrument experiences regular handling during, e.g., field campaign operations.

Do you correct King probe LWC (seems not to be the case), knowing that sensitivity below 10 $\mu$m and above may be 30-40 $\mu$m is reduced. You are using this probe for LWC reference, however Strapp (2003) demonstrated large deviations of King probe LWC also for larger drop diameters of 40 $\mu$m (may be already 30 $\mu$m?). Your effective drop diameters reach 26 $\mu$m .... Uncertainty of solely 5% in LWC is difficult to believe.

A: The uncertainty of 5 % was referenced with the original paper by King et al.. In order to account for the particles <10um that are not fully detected by the hotwire, we only consider size distributions with an effective radius above 5 um to reduce the contributions from smaller particles. Since the agreement between the Hotwire and CAS-DPOL and the Hotwire and CDP doesn't change with effective diameter, this is a good indicator that the uncertainty of the measurement doesn't change with the growth of the detected particles. Figure 6d shows the size resolved contribution of LWC for effective radii of 26 um. The fraction of CWC from particles above 30 um is one to two orders of magnitude less than the maximum CWC. Therefore we believe that the uncertainty of 5% is justified for the size range considered in this analysis.

Line 309: And what if King probe and CAS DPOL are both wrong and CDP is right?

A: We believe that as they measure the same cloud volume and have good agreement the measurements are correct. This does not exclude the possibility of CDP works fine as we were measuring at very inhomogeneous convective clouds. Also, perfect agreement between both probes is not expected due differences at cloud volume and singular characteristics of each instrument (e.g. sample area, inlet configuration etc.)

Line 339: Why don't you correct CAS DPOL data for your calibrations? Consequently, in your data the CAS DPOL instrument undersizes large droplets! (40 $\mu$m in diameter appear as 35 $\mu$m drops?). In case your effective diameter droplets of 26 $\mu$m would have been 30 $\mu$m droplets in reality, you are underestimating LWC by 50% for these droplet sizes: : : Likewise, the King probe is underestimating LWC for other reasons as mentioned above.

A: In Figure 6d (available at supplementary material), we show size distributions and CWC distributions of both cloud probes for large effective diameters. Even for the large effective diameter, only few drops > 40 $\mu$m were measured by both instruments CDP and CAS-DPOL. The difference between CAS-DPOL and the CDP above 35um is less than a factor of two, though at very low concentrations. Thus, the difference/error in diameter of these large particles does not contribute significantly to the calculated effective radius. The large droplets contribute an order of magnitude less to the total water content than droplets in the size range around 25 um. Changing the calibration would change the result of the intercomparison of CWC insignificantly. According to Strapp et al. 2003, hotwire probes in general compared well to other LWC instruments up to MVDs of about 32 um, which comprises the largest fraction of the CWC measured during ACRIDICON-CHUVA. To sum up, the disagreement between the CWC of hot wire and the cloud probes represent an upper limit of the cloud probe inaccuracy.

Line 386-394: What is the uncertainty in N0 and k calculation and finally the uncertainty in equation (2) calculated droplet number (calculated each second) when averaging

CCN2 per time step normalized by FA (with two other averages of mCCN1 per time step and TmCCN1 average of all mCCN1 time steps or may be even all CCN1 data)?

A: We have changed the manuscript to address these questions as follow:

" The calculated NCCN(S) errors for these flight segments are a function of the measured particle number (i.e. 10% of NCCN(S) for large concentrations and the mean of the error is around 20% of NCCN(S)). The estimated standard error (STDE) for the N0 and k parameters and CCN estimates were calculated (as described in Appendix B) for each flight segment and are shown in Table 2. The table shows that the STDE associated to the Twomey's equation fit is about 5% for the N0 and k parameters. The changes in the air mass assumed to correct the CCN2 for FA during the flight segments were up to 24 % for all flights. As long as the cloud segment compared with this data are not at exactly the same location as the measurements was performed, the mean (i.e. TmCCN1) is a good measure for this comparison. The standard error was used for the error propagation calculations and the resulting error in NCCN(S) is 15 % of NCCN(S) estimates on average. The resulting error of N0 (k slope) was also calculated and is 23 % (20 %) of N0 (k) values on average, associated to the Twomey's equation fit and the NCCN(S) error."

"The uncertainties regarding the Smax, NdCCN and NdT estimates for measurements at cloud base with both probes (CCP-CDP and CAS-DPOL) are on average about 22, 20 and 38 % for all flights, respectively (the uncertainty method adopted for these theoretical estimates are available at Appendix A). The Wb uncertainty of 0.3 m s-1 impacts on average for about 65% (60 %) on NdT (Smax) uncertainty, and the uncertainty from the estimated Smax contributes for most of NdCCN uncertainty ($\sim$70% on average)."

Line 400: equation (3) does not pretend Smax depending on NCCN(S). Please detail how Nd can be used to achieve a closure for NdCCN estimate.

A: Ok. The following text was added: " The value of Smax at cloud base can be estimated from Eq. 3 based on the vertical velocity at cloud base and Nd values measured with the cloud probes CCP-CDP and CAS-DPOL (Ncdp and Ncas, respectively). Therefore, the estimated Smax near cloud base can be used in Eq. 1, producing the NdCCN estimates to achieve a closure for Nd measurements at cloud base."

Section 5.2.1.: Gray solid/dashed lines difficult to see in Figs 11 and 12!

A: Ok. We made it thicker.

Fig 11a & 11c show very weak overlap of NdCCN and NdT including both uncertainties. In addition, real Nd measurements can be considerably outside NdT uncertainties and particularly outside the overlap region. Why? What is the value of this study when measurements are not better matching the calculations with their uncertainties? Are the already large uncertainties still underestimated? Are measurements and calculations comparable in their complexity of the respective environments? I don't think so. . ..

A: NdT show a great overlap at Figures 11a and 11c, whilst NdCCN indeed have a weaker overlap (see Figures 11a and 11c at supplementary material). The following text was added to the manuscript: "Both values of Ncas and Ncdp are within the range of the theoretical expectation of NdT and NdCCN, except for occasional deviations at the extreme percentiles. For example, the maximum NdT versus maximum Nd are outside the error interval for Nd. This is so because extreme percentiles are much more prone to random variations than the middle range, such as the median. The lines of NdT mostly agreed quite well with the lines of Nd with only small deviations. The NdCCN mostly underestimates Nd by down to a factor of 0.5 for reasons that we could not identify. Entrainment is not a likely cause, because it would dilute Nd and thus incur NdCCN to be biased positively with respect to Nd. It appears that measuring S in clouds is still a great challenge, even indirectly by using Eq. 3. Remarkably, Eq. 2 (Twomey, 1959), which avoids an explicit usage of S, still performs better when limited within the observed bounds of Wb and k within the cloud."

Fig 12: Color difference of Nd curves (red) and Nd in legend (blue). Can you also show

results for AC13 and AC16? Fig 12b and 12c as well as 11c show Nd that significantly exceed NdT for higher Wb. Explanation? The problem stems basically from NCCN2 calculation?

A: The response to the previous comment addresses this comment too.

Line 513: Change 10% to at least 15% if not 20% (AC14!).

A: Ok. Changed.

Line 513-517: and a factor of 1.5 for other cases AC11 and again AC17. Solely AC 13 and AC16 data points ok. Therefore I don't agree with that improper statement. Line 566-567: I would call a factor of 2 in NdCCN* to Nd* comparison a pretty bad result rather than a good agreement.

A: The following text was added to the manuscript regarding these comments: "…Figure 13a shows the values of Nd* and NdT* for the different cloud base measurements shown in Figs. 11 and 12. The NdT* agrees with Nd* within the measurements uncertainties, as shown by the error bars. The bias of NdT* with respect to Nd* for the CAS -DPOL is 1.00 with a standard deviation ±0.17 around it. The respective result for the CDP is 0.84 ±0.12. A weaker agreement is observed for comparisons between NdCCN* and Nd* (see Fig. 13b), A factor of ∼2 can be observed for some cases (AC14 and AC17). The bias of NdCCN* with respect to Nd* for the CAS-DPOL is 0.80 ±0.07. The respective result for the CDP is 0.76 ±0.1."

Please also note the supplement to this comment:
http://www.atmos-chem-phys-discuss.net/acp-2016-872/acp-2016-872-AC3-supplement.pdf

**Supplement:**

[Figure]

[Figure]

**Figure 6. (left) Mean cloud droplet concentration (solid lines) and (right) cloud water content as a function of droplet diameter in the left and right panels, respectively, for a) 5 μm < $r_e$ < 6 μm; b) 8 μm < $r_e$ < 9 μm; c) 11 μm < $r_e$ < 12 μm; d) 12 μm < $r_e$ < 13 μm. The probes are identified by colors as shown at the top of the panels. The error bars indicate the uncertainty range of mean cloud droplet concentration and cloud water content values as a function of droplet diameter.**

[Figure]

[Figure]

[Figure]

c) FLIGHT AC14 TIME: 15:06(184 s) CCN=1509.7 · $S^{0.973}$

[Figure]

d) FLIGHT AC16 TIME: 20:10(58 s) CCN=1966.2 · $S^{0.672}$

[Figure]

**Figure 11a-f.** $N_{dCCN}$, $S$, $N_{dT}$ and $N_d$ values are presented as a function of the cloud base updrafts ($W_b$). This plot is based on the 'probability matching method' (PMM), using same percentiles for $W_b$ and $N_d$ ($N_{dCCN}$ or $N_{dT}$). The values of $N_{dCCN}$, $N_{dT}$ and $N_d$ are shown the left y-axis, those of $S$ on the right y-axis. The black dashed lines are the $N_{dT}$ uncertainties. The gray solid (dashed) lines are the $N_{dCCN}$ values (uncertainties). The effective updraft $W_b$* for each flight segment is shown by the cyan line. The data are based on the CAS-DPOL probe. The time, period of measurements (sample size in seconds), and $N_{CCN}(S)$ equation are shown on the top of the figures.

---

## Author Response (AR1)

**Review of "comparing calculated cloud microphysical properties of tropical convective clouds at cloud base with measurements during the ACRIDICON-CHUVA campaign" by *Ramon Campos Braga et al.***

**Anonymous Referee #1**

Braga et al., use airborne measurements aboard HALO from a CCP, CAS-DPOL and CCN counter to derive cloud drop size distributions (DSDs) and cloud water content from various instruments via an inter–comparison. In this study parameterizations for liquid cloud formation in tropical convection are validated, but for instance comparing the directly measured cloud drop concentrations (Nd) near cloud base to inferred values that are derived by combining the cloud base updraft velocity, CCN vs SS (supersaturation) spectra. In addition, Nd from cloud base was also compared to drop concentrations (Na) derived by assuming adiabatic expansion for vertical evolution of cloud drop effective radius above cloud base.

Overall, this paper presents a good summary but it lacks a significant scientific finding or discovery. Rather it is verifying previous formulated parameterizations, which is valuable. However, the authors could do a better job of comparing the differences they observe between the parameterizations validated here with previous studies.

Perhaps the paper can be re-worked to demonstrate the novelty of the work, which is lacking in the current version of the manuscript. Specific comments below should help achieve this. After such revisions have been made, the paper maybe considered for publication.

There are small editorial issues and some grammatical errors throughout the manuscript, of which I have pointed out a few, but will leave it to the authors to check that more carefully upon submission of the revised version.

*General comments*

The authors thank the referee for the general comments and advices. Furthermore, the advices of the referee are highly appreciated as well as the very valuable and constructive suggestions to increase the quality of the manuscript. We tried to address the points requested by the reviewer to the

paper be considered for publication. Overall, we have improved the focus of the paper highlighting our objectives and the novelty of our study.

**Specific comments:**

Line 29: Why not introduce CWC here like all the other acronyms in the abstract?
A: OK. Changed.

Line 46: "pursue" replace this word with something more suitable like "cloud microphysical models "aim" to reproduce or "The goal of cloud microphysical models is to reproduce…."
A: OK. Changed.

Line 137 "account" should be "accounted"
A: OK. Changed.

The discussion in line 132 to 137 can be expanded upon to make the paper more scientifically novel. State in more detail what was unique about these measurements, are the convective clouds here unique? Related to this but later in the paper, are the results obtained here the same as other convective regions in the world? Could the authors comment or discuss this? If indeed this is the case, that the results are similar to other locations of convection globally, the authors may consider discussing this point and stressing this aspect.
A: The text was changed to address these comments. Thanks.

New text:

"The availability of these measurements collected by the same aircraft provides a unique opportunity to compare the data with model predictions and to test the sensitivity of the results to the differences between the measurements by the cloud probes.
This study is novel in several aspects:

a. It is the first study that validates the methodology of retrieving the adiabatic cloud drop concentrations $N_a$ (Freud et al., 2011) from the vertical evolution of $r_e$ while assuming that $r_e$ is nearly adiabatic. This is important because it supports the validity of retrieving $N_a$ from satellite-retrieved vertical profile of $r_e$ (Rosenfeld et al., 2014a and 2016).

b. It is the first study that tests with aircraft the measured $N_d$ with its parameterization that is based on $N_{CCN}(S)$ along with cloud base spectrum of updrafts weighted by the updraft speed itself, $W_b^*$. It is done this way to be compatible with the recently developed methodology of retrieving CCN from satellites by means of retrieving $N_d$ and $W_b^*$ (Rosenfeld et al., 2016).

c. It is the first study that compares observationally the old Twomey (1959) parameterization of the dependence of $N_d$ on $W_b$ (Eq. 2) versus the recent Pinsky et al. (2012) analytical expression for the same (Eq. 3)."

References:

Freud, E., Rosenfeld, D. and Kulkarni, J. R.: Resolving both entrainment-mixing and number of activated CCN in deep convective clouds, Atmos. Chem. Phys., 11(24), 12887–12900, doi:10.5194/acp-11-12887-2011, 2011.

Pinsky, M., Khain, A., Mazin, I. and Korolev, A.: Analytical estimation of droplet concentration at cloud base, J. Geophys. Res. Atmos., 117(17), 1–14, doi:10.1029/2012JD017753, 2012.

Rosenfeld, D., Fischman, B., Zheng, Y., Goren, T. and Giguzin, D.: Combined satellite and radar retrievals of drop concentration and CCN at convective cloud base, Geophys. Res. Lett., 41(9), 3259–3265, doi:10.1002/2014GL059453, 2014a.

Rosenfeld D., Y. Zheng, E. Hashimshoni, M. L. Pöhlker, A. Jefferson, C. Pöhlker, X. Yu, Y. Zhu, G. Liu, Z. Yue, B. Fischman, Z. Li, D. Giguzin, T. Goren, P. Artaxo, H. M. J. Barbosai, U. Pöschl, and Meinrat O. Andreae, 2016: Satellite retrieval of cloud condensation nuclei concentrations by using clouds as CCN chambers. Proceedings of the National Academy of Sciences, doi:10.1073/pnas.1514044113.

Twomey, S.: The nuclei of natural cloud formation part II: the supersaturation in natural clouds and the variation of cloud droplet concentration, Geofis. Pura e Appl., 43(1), 243–249, doi:10.1007/BF01993560, 1959.

Line 149: should read "Manaus City" not "Manaus city"

A: OK. Changed.

Line 193-194: Delete "was used additionally considering" and Line 194: add "was considered" after 10%. In total the sentence should read "For the CDP sample area of 0.22 mm2, an uncertainty of about 10% was considered (Molleker et al., 2014)."

A: OK. Changed.

Line 205: Delete extra periods

A: OK. Changed.

Line 267: "maximal" should be "maximum"

A: OK. Changed.

Line 269: should "probes" have an apostrophe after it i.e. probes'? it sounds like it is being used in the possessive.

A: OK. Changed.

Line 297: Why these specific flights being used (AC08 and AC20) for the CWC, why not data from the entire campaign? Also, why not use the same flights as were used in the effective radius comparison (line 278)?

A: The sentence is wrong. We used all flights, except AC07 which we have not hotwire data. We have corrected the sentence.

Line 309-314: Why compare only with one hot wire probe when three of them were on board the aircraft?

A: There was only one hotwire operational aboard HALO. This one was mounted on the CAS-DPOL.

Some DMT instruments are delivered with hot-wires. The CCP was indeed equipped with the hot-wire as well, but we do not operate it during flight as then

the CCP's overall power consumption would exceed the limits. It was physically disconnected and no part of CCP anymore.

Line 319: insert "the" before "hot-wire"
A: OK. Changed.

Line 320: Can you make it clearer that this is a decreasing number concentration with *increasing* effective radius
A: OK. Changed.

Line 322: insert "the" ahead of "CAS-DPOL" in general the grammar is really poor from lines 320-325, please rectify
A: The text was rewritten.

Line 326-333: Why not consider using only particles less than 40 microns in your CWC comparison?
A: The amount of CWC above 40 µm is negligible. However, it illustrates the difference in the shapes of the DSDs.

Line 406: replace "greater than or equal" with the symbol "≥"
A: OK. Changed.

Line 471-479: Are these values presented here similar to literature values from other locations in the world? Can there be a comparison and discussion of this?
A: Ok. We added a new text:

"For some flights the values estimated for $N_0$ and $k$ parameters of Eq. 1 are similar to what was found by Pöhlker et al., (2016) for ground measurements ($N_0 = 1469 \pm 78$ and $k = 0.36 \pm 0.06$) during the dry season in the Amazon. However, in the majority of the cases $N_0$ and $k$ are twice or three times greater than the values from Pöhlker et al., (2016). These differences are probably related to flying selectively to areas that had high aerosol concentrations to contrast the cloud behavior with the flights with low aerosol concentrations, as shown in Fig. 2. The high CCN measured in this study is more similar to

previous aircraft measurements in smoky conditions over the Amazon (Andreae et al., 2004; Freud et al., 2008)."

Table from Pöhlker et al., 2016

**Table 5.** Twomey fit parameters describing CCN spectra $N_{CCN}(S)$ vs. $S$ as parametrization input data (compare to Figs. 10 and 11a, c). Fit parameters are provided for annually averaged CCN spectra and resolved by seasons.

| Time period | $N_{CCN}(1\%)$ (cm$^{-3}$) | $k$ | $R^2$ |
|---|---|---|---|
| Annual | $998 \pm 60$ | $0.36 \pm 0.04$ | 0.88 |
| Wet season | $289 \pm 7$ | $0.57 \pm 0.03$ | 0.98 |
| LRT period | $378 \pm 9$ | $0.38 \pm 0.03$ | 0.94 |
| Transition | $970 \pm 40$ | $0.49 \pm 0.05$ | 0.94 |
| Dry season | $1469 \pm 78$ | $0.36 \pm 0.06$ | 0.86 |

Andreae, M. O., Rosenfeld, D., Artaxo, P., Costa, A. A., Frank, G. P., Longo, K. M. and Silva-Dias, M. A. F.: Smoking rain clouds over the Amazon, Science, 303(5662), 1337–42, doi:10.1126/science.1092779, 2004.

Freud, E., Rosenfeld, D., Andreae, M. O., Costa,  A. A. and Artaxo, P.: Robust relations between CCN and the vertical evolution of cloud drop size distribution in deep convective clouds, Atmos. Chem. Phys., 8(6), 1661–1675, doi:10.5194/acp-8-1661-2008, 2008.

Pöhlker, M.L., Pöhlker, C., Ditas, F., Klimach, T., Hrabe de Angelis, I., Araújo, A., Brito, J., Carbone, S., Cheng, Y., Chi, X. and Ditz, R., 2016. Long-term observations of cloud condensation nuclei in the Amazon rain forest–Part 1: Aerosol size distribution, hygroscopicity, and new model parametrizations for CCN prediction. *Atmospheric Chemistry and Physics,* *16*(24), pp.15709-15740.

Line 520: Figure 14a shows the LWC? The Na that is stated in Figure 14a is also mentioned here in Line 523, not sure why the reference to Figure 14a is needed here.

A: Yes, LWC for adiabatic fraction > 0.25.

A: Ok.

Line 530-534: The scaling of 1.3 works quite well, perhaps mention it here since this is a new data set.

A: Ok. Thanks.

Line 558: Here the authors should make a case for why their work was novel, interesting or what is new about their work.

A: The conclusions now clearly show the novel aspects of this study.

Line 570-574: Was there any doubt about the validity of the parameterization prior to this study? What is new about the work here other than the fact that the measurements were all taken during this campaign on one/the same aircraft?

A: Most of the tested parameterizations are new, and it is the first time that they are tested comprehensively alongside old parameterizations.

Figures

Fig 4: Consider editing the plot so that the legend matches the sub=plot where the quantities are shown

Fig 4 (lower left panel for CWC): Why is it necessary to have a log scale? The data just cover one order of magnitude and are all squeezed to the bottom half of the panel. There is no need for the scale to extend to 10. And no need for a log scale either. This artificially downplays some of the differences between the probes.

A: We changed the Figure 4 as you suggest. See below.

[Figure]

**Figure 4.** Mean cloud water content from the hot-wire measurements and estimated from the cloud probes (CCP-CDP and CAS-DPOL from top to bottom, respectively) as a function of effective radius ($r_e$) size (left panel). The ratios between the hot-wire liquid water content and the cloud water content derived from each probe are shown in red (*CWCr*). The total uncertainty for each probe and the hot-wire measurements are shown by the dotted lines. The number of cases (black continuous line), hot-wire measurement standard deviations (dashed black line), and probe CWC standard deviations (dashed colored line) for each $r_e$ size are shown in the right panels.

Fig 6a and 6b: is it necessary to have zeros in front of the micron sizes, i.e. 05 instead of 5. Also, can both scales be made linear for consistency and clarity? It is hard to compare presented in the manner here.

A: The others referees also have requested some changes in figures 6. We have changed the figures 6 and captions to address the suggestions. Following these suggestions we have changed the DSDs figures using histograms of binned detection channels. The data points are shown with size bin limits in x-direction (to cover the Mie-ambiguity ranges, providing an approach to have the channel-wise sizing error superimposed by the size-bin limits) and uncertainty in y-direction. We also eliminate the zeros in front of micron sizes. See new figures below.

[Figure]

[Figure]

[Figure]

**Figure 6.** (left) Mean cloud droplet concentration (solid lines) and (right) cloud water content as a function of droplet diameter in the left and right panels, respectively, for a) 5 µm < $r_e$ < 6 µm; b) 8 µm < $r_e$ < 9 µm; c) 11 µm < $r_e$ < 12 µm; d) 12 µm < $r_e$ < 13 µm. The probes are identified by colors as shown at the top of the panels. The error bars indicate the uncertainty range of mean cloud droplet concentration and cloud water content values as a function of droplet diameter.

Line 1134: Italicize "m"

A: Ok. Thanks.

Fig 12 (all panels): Shouldn't Nd be in red?

A: Yes. Thanks.

Fig 13 (line 1190): reference to Fig 7-8 is not consistent with text, should be Figure 11 and 12

A: Yes. Thanks.
The Manuscript is a little vague in its objectives, but it appears that it is attempting to validate aircraft observations by performing closure with either different instruments or between CCN measurements made at different supersaturations and cloud droplet numbers measured at different updraft speeds. It does this using below cloud and in cloud measurements made during the ACRIDICON-CHUVA campaign and combining these with activation models.

The other reviewers have already made comments regarding the models used so I will focus here mostly on the measurements and the analysis that goes along with those.

Unfortunately this paper needs significant extra work in order to make it of publishable quality. However I think the type of analysis that has been performed here is valuable and is not undertaken enough. This is the type of paper that can be used to assure the quality of the measurements being made and that other papers in the project can reference to avoid repeating this analysis by multiple groups and authors. It is also the type of paper that can highlight the limits of the instruments. This is good as it can provide insight to a modeller who is using the data perhaps without an in-depth knowledge of its limits and it also means that it becomes clear what science cannot be performed with the data and therefore where we need to improve our instruments, calibration methods and analysis techniques. However the work is only valuable in this sense if the analysis is performed in an incredibly rigorous manner. I applaud the author's attempt to write this paper, but I would suggest that he needs to pull in more input from coauthors - there are many well respected coauthors on the paper and I am surprised that their instrument knowledge does not show through in this paper. There are certainly other

people who work full time within the aircraft instrument community who could have input.

I would suggest that the manuscript needs a full rewrite and I would suggest that the author goes back to basics in terms of deciding exactly what the objectives are (are they validating instruments or validating the cloud models), then doing a thorough uncertainty analysis of the instruments. This must include details of calibration methods used and the uncertainty derived from those, plus things that cannot or have not been calibrated and the reason why and what the expected uncertainty for these things might be. Based on the uncertainty analysis the author can then decide if the objectives are achievable and can present appropriate uncertainties in the conclusions. Based on the general comments above I recommend the paper be rejected in its current form. Some more detailed comments follow

*General Comments:*

The authors thank the referee for the general comments and advices. Furthermore, the advices of the referee are highly appreciated as well as the very valuable and constructive suggestions to increase the quality of the manuscript. We tried to address the points requested by the reviewer to the paper be considered for publication. Overall, we have improved the focus of the paper highlighting our objectives and the novelty of our study.

The availability of the measurements collected by the aircraft HALO provides a unique opportunity to compare the data with model predictions and to test the sensitivity of the results to the differences between the measurements by the cloud probes.

This study is novel in several aspects:

a. It is the first study that validates the methodology of retrieving the adiabatic cloud drop concentrations $N_a$ (Freud et al., 2011) from the vertical evolution of $r_e$ while assuming that $r_e$ is nearly adiabatic. This is important because it supports the validity of retrieving $N_a$ from satellite-retrieved vertical profile of $r_e$ (Rosenfeld et al., 2014a and 2016).

b. It is the first study that tests with aircraft the measured $N_d$ with its parameterization that is based on $N_{CCN}(S)$ along with cloud base spectrum of updrafts weighted by the updraft speed itself, $W_b^*$. It is done this way to be compatible with the recently developed methodology of retrieving CCN from satellites by means of retrieving $N_d$ and $W_b^*$ (Rosenfeld et al., 2016).

c. It is the first study that compares observationally the old Twomey (1959) parameterization of the dependence of $N_d$ on $W_b$ (Eq. 2) versus the recent Pinsky et al. (2012) analytical expression for the same (Eq. 3)."

References:

Freud, E., Rosenfeld, D. and Kulkarni, J. R.: Resolving both entrainment-mixing and number of activated CCN in deep convective clouds, Atmos. Chem. Phys., 11(24), 12887–12900, doi:10.5194/acp-11-12887-2011, 2011.

Pinsky, M., Khain, A., Mazin, I. and Korolev, A.: Analytical estimation of droplet concentration at cloud base, J. Geophys. Res. Atmos., 117(17), 1–14, doi:10.1029/2012JD017753, 2012.

Rosenfeld, D., Fischman, B., Zheng, Y., Goren, T. and Giguzin, D.: Combined satellite and radar retrievals of drop concentration and CCN at convective cloud base, Geophys. Res. Lett., 41(9), 3259–3265, doi:10.1002/2014GL059453, 2014a.

Rosenfeld D., Y. Zheng, E. Hashimshoni, M. L. Pöhlker, A. Jefferson, C. Pöhlker, X. Yu, Y. Zhu, G. Liu, Z. Yue, B. Fischman, Z. Li, D. Giguzin, T. Goren, P. Artaxoi, H. M. J. Barbosai, U. Pöschl, and Meinrat O. Andreae, 2016: Satellite retrieval of cloud condensation nuclei concentrations by using clouds as CCN chambers. Proceedings of the National Academy of Sciences, doi:10.1073/pnas.1514044113.

Twomey, S.: The nuclei of natural cloud formation part II: the supersaturation in natural clouds and the variation of cloud droplet concentration, Geofis. Pura e Appl., 43(1), 243–249, doi:10.1007/BF01993560, 1959.

*Specific comments:*

Introduction - In general the author should be familiar with the calibration methods used with the instruments used and this should be reflected in the references.

line 75 - Previous analysis (Strapp et al 1992, Journal of Atmospheric and Oceanic Technology Vol 9 p 548) has indicated that the PCASP dries it's sample through ram heating during measurements. The author should familiarise himself with this work, and understand why this drying may not be happening.

A: There is no mention of PCASP near line 75 of the ACPD paper.
We are well aware of the claim that PCASP is drying the aerosols in the inlet. The observations show that this assumption is clearly not valid. This disqualified the use of PCASP for this study. This is explained in lines 111-117 of the ACPD paper.

Line 85 - If CCN measured at constant S is not constant then either N0 or k in (1) are changing. I.e., either the total number is changing but everything else remains the same or the hygroscospicity or size distribution of the aerosol is changing. Or of course there could be a combination of these factors. The author must show which are occurring.

A: We highlight that CCN concentration changes as a function of *S* or CCN load. Therefore, we correct CCN concentrations measured with different *S* with the CCN loaded observed for measurements with fixed *S*. The CCN load is calculated by calculating the difference between the measured CCN concentrations with the mean CCN concentration measured for a fixed *S*.

Line 95-100 - Total number totally cancels from effective radius calculations and adiabatic calculations reveal expectations for mass of condensed water not number. Dividing adiabatic water content by measured mass per particle would give a number concentration but even in an adiabatic regime the uncertainty on

this would be larger than the measured droplet number concentration. Calibrations on the CDP operated by FAAM using the method described by Rosenberg et al 2012 (Atmospheric Measurement Techniques vol 5 p1147) provides an uncertainty around 0.5 um in sizing, but typically shows a discrepancy of around 2 um from the manufacturer's specification. If the manufacturer spec is used in this work then we can expect that at 20 um we have approximately 30% uncertainty in mass per particle measurements.

Line 190 - Has the collecting angle of the instruments been measured? This defines the location of the Mie wiggles and where the bins should be merged.

A: Given the uncertainty of the sample area, the Probe Air Speed (PAS), article losses, deviations and maybe coincidence (not negligible but likely not a significant issue) the uncertainty in concentration ranges below 20% and likely approaches or exceeds 20% only in cases of tight curve maneuvers à this might be the most prominent case when the "collecting angle" comes into play. For level flight (straight heading) I would quantify this issue to be comparatively small given the flight speeds of generally larger 170m/s and up to 240m/s – i.e. the direction of particle penetration may be predominantly perfectly in-line with the flight direction (unless there was a systematic deviation of cloud elements due to flow disturbances induced by the aircraft structure or the neighbored instrument, etc. for which at current state no clear evidence is given, yet).

Line 246-260 - As described previously this assumes k is constant, the author needs to provide evidence this is a good assumption. The correction method means that we are correcting to a point where N0 is equal to the average N0 for the scan. This should be made clear and an estimate of how much N0 is varying must be made as this impacts how much confidence we have in a model's estimate of Nd in cloud.

A: The correction method assumes that the variability of $CCN_1$ at each flight step can be corrected by the average measurements of $CCN_1$ ($TmCCN_1$). Indeed, there is a variation on $TmCCN_1$ for each flight segment. The calculated standard deviation for $TmCCN_1$ in all flight segments was up to 24 %, indicating a small impact on the parameterization proposed to fit the Twomey equation (Eq. 1). We calculated the uncertainty impacts from the adjusted $N_0$ and $k$ and it contributes to about 35 % of $N_{dCCN}$ and $N_{DT}$ uncertainties on average.
This was added to the text.

Line 261-269 - When I first read this seemed entirely circular. Later it becomes clear that this is the point. We are putting observations into a model and checking for consistency. The author should highlight in the aims of the paper that they intend to do this so that the reader knows to expect this. A better way to represent this may be a plot f Nd vs S with data points taken from measurements and derived through equation 3 (perhaps coloured by w) along with points from the scanning and static CCN instrument. If the model is correct and the obs are consistent then all points should fall on one line.

A: Unfortunately $S$ measurements within clouds are not accurate, and then we could not compare with $S_{max}$ estimates.

Lines 284-288 This probability matching method assumes droplet number is a monatonic function of w only. I have no issue with the monotonic assumption, but the author should show that there is no other influences upon drop concentration such as entrained dry/clean air and constant aerosol/ccn concentration below cloud or at least state why this is a good assumption.

A: We assume that for a given CCN(S) spectrum below cloud base the droplet number measured at cloud base is a function of W, as stated by Twomey equation (Eq. 2). The analysis show that for most of cases that the theoretical estimates do not reproduce the measured $N_d$ the degree of entrainment should be high (because we have a large dispersion of $N_d$ values). This is an issue that we should highlight in the new version of the manuscript.

New text at manuscript:

"A suitable method to analyze the relationship between $W_b$ and $N_d$ measurements is the 'probability matching method' (PMM) (Haddad and Rosenfeld, 1997), which requires that the two related variables will be increasing monotonically with each other. For a set of measurements of $W_b$ and $N_d$ at cloud base, it is expected that larger $W_b$ would produce larger $N_d$ for a given $N_{CCN}(S)$. Therefore, it is assumed also that $N_d$ is produced uniquely by $W_b$ for a given $N_{CCN}(S)$ spectrum as calculated from the measurements below cloud base. It is further assumed that entrainment does not change systematically with $W_b$ in a way that would reverse the monotonic increase of $W_b$ with $N_{CCN}(S)$."

Lines 307-325 This needs a thorough uncertainty analysis to show its usefulness as described earlier.

A: The line numbers do not match anything that can be relevant to the comment in the manuscript.

Line 350 - You are claiming an uncertainty of 5% in N0, but as described earlier this is in the average N0 over the scan. We have seen CCN number on the constant supersaturation instrument vary from 650 to 950 cm-3 so it seems unreasonable to claim 5% uncertainty in this parameter. . This ambiguity comes from not being clear in the first instance about what you are trying to measure. In reality I think an estimate of k is what you should be aiming for as N0 is clearly changing and is not a constant.

A: We have calculated an uncertainty of about 20% for large $N_{CCN}(S)$ and about 10 % for smaller $N_{CCN}(S)$. On average, $N_{CCN}(S)$ have an uncertainty of 15 %, $N_0$ and $k$ 20 and 23 %, respectively. This is better described at the new version.

Line 380-390 I certainly would not be alone in suggesting that the phrase "agree closely" and similar variations has very little place in scientific work. In this case there is a difference of up to 70% in fig 9a. Phrases such as "agree within the

measurement uncertainties," "differ by up to x amount," or "agree to the extent that conclusion y is unaffected" are all appropriate, but "agree closely" is entirely subjective.

A: Ok. changed.

Line 401 - Another "good agreement" statement. Points here deviate from the 1:1 line by up to a factor of 2.

A: We rewrote the sentence.

New text:

"…Figure 13a shows the values of $N_d$* and $N_{dT}$* for the different cloud base measurements shown in Figs. 11 and 12. The $N_{dT}$* agrees with $N_d$* within the measurements uncertainties, as shown by the error bars. The bias of $N_{dT}$* with respect to $N_d$* for the CAS -DPOL is 1.00 with a standard deviation ±0.17 around it. The respective result for the CDP is 0.84 ±0.12. A weaker agreement is observed for comparisons between $N_{dCCN}$* and $N_d$* (see Fig. 13b), A factor of ~2 can be observed for some cases (AC14 and AC17). The bias of $N_{dCCN}$* with respect to $N_d$* for the CAS-DPOL is 0.80 ±0.07. The respective result for the CDP is 0.76 ±0.1. "

Line 440-444 - This difference is almost certainly within the expected uncertainty which as described above is probably 30% from the mass per particle measurement, plus perhaps 10-20% from sample area and air speed through the sample volume estimates.

447-450 and figs 13/14 - I see size distributions like this all the time and often by people who work with these instruments a lot. They are unfortunately not really appropriate styles for plotting size distributions. The following changes should be made. The plot should show points and not lines. It is not appropriate to "join the dots" on a plot that has significant uncertainties. Each point should have an x and y uncertainty. Standard error is not an appropriate uncertainty to use. It assumes that we measure the same thing repeatedly and that the uncertainty is dominated by noise. Here we have concentrations that vary with

time during and between the periods that contribute to these average size distributions. So the standard error becomes some combination of noise and variability and omits all systematic uncertainties. Instead the author should do a proper error analysis including contributions from sample area, air speed at the probe, bin width and counting (Poisson) uncertainty for the y error and sizing uncertainty for the x error.

A: The CDP'S sizing uncertainty is calibrated regularly before, during and after flights with mono-sized glass beads or Poly Styrol Latex (PSL) of various sizes. The uncertainty of these calibrations mainly results from the uncertainty in size of the test aerosol and refractive index resulting in an uncertainty for specific particle diameter of at most 10%. The CIPgs sizing may imply an general uncertainty of +/- 15 µm which is the instruments resolution. In the way we treated the data for spherical bodies, the uncertainty should not be larger for the CIPgs sizing as the particles are mainly in PAS speed or faster (latter causes an image squeezing in flight direction which is compensated by choosing the diameter in diode array direction for the image sizing). Furthermore, in the droplet regime, the reproduction of the Fresnel-diffraction may cause a non-systematic uncertainty in the sizing. However, all droplet sizes may more or less be influenced in the airborne state by deformation or shrinking (very smallest drops - due to congestion heating) in the compressed flow regime upstream of the probes at highest flight speeds which at current state is not quantifiable but may be small.

We agree with the referee that the DSDs on Figures 6 should not be presented as line plots. The line between data points suggests a course of the DSDs that is not real. Instead, we have changed the DSDs figures using histograms of binned detection channels. The data points are shown with size bin limits in x-direction (to cover the Mie-ambiguity ranges, providing an approach to have the channel-wise sizing error superimposed by the size-bin limits) and uncertainty in y-direction.

The new figures 6a-d are available below.

[Figure]

[Figure]

**Figure 6.** (left) Mean cloud droplet concentration (solid lines) and (right) cloud water content as a function of droplet diameter in the left and right panels, respectively, for a) 5 μm < $r_e$ < 6 μm; b) 8 μm < $r_e$ < 9 μm; c) 11 μm < $r_e$ < 12 μm; d) 12 μm < $r_e$ < 13 μm. The probes are identified by colors as shown at the top of the panels. The error bars indicate the uncertainty range of mean cloud droplet concentration and cloud water content values as a function of droplet diameter.

Line 454-560 The sensitivity is probably not the issue, it is more likely to be the bin widths for which we see no calibration.

A: OK.

**Interactive comment on "Comparing calculated microphysical properties of tropical convective**
**clouds at cloud base with measurements during the ACRIDICON-CHUVA campaign" by Ramon Campos Braga et al.**

**Anonymous Referee #3**

Title: Comparing calculated microphysical properties of tropical convective clouds at cloud base with measurements during the ACRIDICON-CHUVA campaign.

Author(s): Ramon Braga et al.

The paper compares calculated with measured microphysical properties of convective liquid clouds in the tropics.

Unfortunately, calculations are not performed within a microphysical model taking into account important spatiotemporal fluctuations of dynamical and thermodynamical properties, turbulence, entrainment, etc... In this study solely a comparison of calculating cloud properties from analytical equations and respective measurements has been performed, which represents a considerable work, however with rather limited outcome.

Conclusions of this comparison study are disappointing and do not gain new insights in liquid convective cloud microphysical processes. The paper barely presents new and noteworthy concepts. As it stands, the work is solely a rather qualitative affirmation of existing parameterizations. Taking these issues into account, the study may better carve out the uncertainties of used cloud parameterizations (equations 1, 2, 3? …) based on the uncertainties of measured cloud parameters from the ACRIDICONCHUVA dataset. Also taking into account missed features and uncertainties stemming from turbulence (and more complex droplet activation) and entrainment not captured in this study. Would this be possible at all? The uncertainties of your instruments and derived measurements have been discussed rather honestly in this manuscript. This is

why I encourage authors to develop this manuscript into that direction. Otherwise, I would recommend rejection of this manuscript due to its poor contribution to scientific progress.

The manuscript shows some striking and unexplained differences between calculated and measured microphysical parameters (Nd versus NdT, NdT* versus NdCCN* for a series of flights). Is this a principal problem of performed measurements within an environment of complex processes and limited degree of complexity of calculations that are hardly comparable: calculations do not capture measurement data features like turbulence, entrainment, etc...? At least above mentioned differences are more important for higher Wb values!

*General comments*

The authors thank the referee for the general comments and advices. Furthermore, the advices of the referee are highly appreciated as well as the very valuable and constructive suggestions to increase the quality of the manuscript. However, we disagree with the referee affirmation which highlight that the study have a poor contribution to scientific progress. As mentioned before, the objective of the paper is to validate the physical parameterizations connecting between CCN(S), $W_b$, $N_d$ and $N_a$, when tested over the important convective regime of the Amazon. This is the first paper that tests all parameterizations proposed for convective clouds regime against each other with the same dataset.

Furthermore, the parameterizations are far from being old. Rather, as now stated in the first paragraph of the abstract: "The objective of this study is to validate novel parameterizations that were recently developed for satellite retrievals of CCN(S) at cloud base alongside with more traditional parameterizations connecting CCN(S) with cloud base updrafts and drop concentrations."

The calculations of the effective updraft speed at cloud base ($W_b$*) provide a new capability for test the parameterizations proposed in the study and to

ascribe the capability of $N_a$ estimates for convective clouds developed at different aerosol conditions over Amazon. The role idea of using $W_b^*$ is already accounting for the air turbulence (see Rosenfeld et al., 2014a) and the $N_a$ estimates for convective clouds also account for entrainment as is described at section 4.4 (also discussed at Freud et al., 2011). The study supports the use of $N_a$ estimates using satellite data of effective radius vertical profiles at convective clouds over the Amazon basin.

Regarding the possibility of use other parameterizations in our analysis, we have already highlight at introduction section some issues due to the unreasonable measurements of low hygroscopic factor ($\kappa$) values below cloud base, which prevent us to test other types of parameterizations (e.g. *k-Köhler* model estimates) to validate $N_d$ at cloud base.

In summary, the following text was added to the introduction:

"This study is novel in several aspects:

a. It is the first study that validates the methodology of retrieving the adiabatic cloud drop concentrations $N_a$ (Freud et al., 2011) from the vertical evolution of $r_e$ while assuming that $r_e$ is nearly adiabatic. This is important because it supports the validity of retrieving $N_a$ from satellite-retrieved vertical profile of $r_e$ (Rosenfeld et al., 2014a and 2016).

b. It is the first study that tests with aircraft the measured $N_d$ with its parameterization that is based on supersaturation spectrum of CCN along with cloud base spectrum of updrafts, $W_b^*$. It is done this way to be compatible with the recently developed methodology of retrieving CCN from satellites by means of retrieving $N_d$ and $W_b^*$ (Rosenfeld et al., 2016).

c. It is the first study that compares observationally the old Twomey (1959) parameterization of the dependence of $N_d$ on $W_b$ (Eq. 2) versus the recent Pinsky et al. (2012) analytical expression for the same (Eq. 3)."

We agree to better carve out the uncertainties of used cloud parameterizations, which will be available at the new version of the manuscript.

The uncertainties of $W_b$ of HALO were recalculated for all campaign data. In the new version of the manuscript we ascribe the uncertainties of theoretical estimates assuming 0.3 ms$^{-1}$ as the $W_b$ uncertainty. This new number changes the values of variables uncertainties used for $N_{dT}$, $S_{max}$ and $N_{dCCN}$ estimates.

***Specific answers***

Specific comments related to above general statement:

Line 36: What is the impact of $W_b$ uncertainty of 0.2 ms-1 on $N_d$ calculation?

A: The $W_b$ uncertainty impacts on average for about 65% on $N_{dT}$ uncertainty.

Line 98: What is the cumulative impact of $W_b$ and $N_d$ uncertainties on $S_{max}$ calculation and then $N_0$ and $k$?

A: The cumulative impact on $S_{max}$ is 22 % on average. $N_0$ and $k$ is not used on $S_{max}$ calculation, just $W_b$, $N_d$ and temperature and pressure of cloud base (used on coefficient $C$ estimate at equation 2).

A: We have changed the manuscript to address these questions as follow:
"The uncertainties regarding the $S_{max}$, $N_{dCCN}$ and $N_{dT}$ estimates for measurements at cloud base with both probes (CCP-CDP and CAS-DPOL) are on average about 22, 20 and 38 % for all flights, respectively (the uncertainty method adopted for these theoretical estimates are available at Appendix A). The $W_b$ uncertainty of 0.3 m s$^{-1}$ impacts on average for about 65% (60 %) on $N_{dT}$ ($S_{max}$) uncertainty, and the uncertainty from the estimated $S_{max}$ contributes for most of $N_{dCCN}$ uncertainty (~70% on average)."

Line 194: CDP sample area has not been calibrated before, during, after flight campaign? In this case you may not claim only 10% of uncertainty in SA?

A: The CDP sample area has been frequently measured revealing by the way 0.27mm² (not 0.22mm², which should by a typo or an outdated value and must be corrected in the text) with an uncertainty of 10%. The uncertainty +/- 0.03mm² results from repeated measurements. Unless there is no massive manipulation/disarrangement of the CDP's optics or a detectable aging of the laser diode, the sample area remains stable even if the instrument experiences regular handling during, e.g., field campaign operations.

Do you correct King probe LWC (seems not to be the case), knowing that sensitivity below 10 µm and above may be 30-40 µm is reduced. You are using this probe for LWC reference, however Strapp (2003) demonstrated large deviations of King probe LWC also for larger drop diameters of 40 µm (may be

already 30 μm?). Your effective drop diameters reach 26 μm .... Uncertainty of solely 5% in LWC is difficult to believe.

A: The uncertainty of 5 % was referenced with the original paper by King et al.. In order to account for the particles <10um that are not fully detected by the hotwire, we only consider size distributions with an effective radius above 5 um to reduce the contributions from smaller particles. Since the agreement between the Hotwire and CAS-DPOL and the Hotwire and CDP doesn't change with effective diameter, this is a good indicator that the uncertainty of the measurement doesn't change with the growth of the detected particles. Figure 6d shows the size resolved contribution of LWC for effective radii of 26 um. The fraction of CWC from particles above 30 um is one to two orders of magnitude less than the maximum CWC. Therefore we believe that the uncertainty of 5% is justified for the size range considered in this analysis.

Line 309: And what if King probe and CAS DPOL are both wrong and CDP is right?

A: We believe that as they measure the same cloud volume and have good agreement the measurements are correct. This does not exclude the possibility of CDP works fine as we were measuring at very inhomogeneous convective clouds. Also, perfect agreement between both probes is not expected due differences at cloud volume and singular characteristics of each instrument (e.g. sample area, inlet configuration etc.)

Line 339: Why don't you correct CAS DPOL data for your calibrations? Consequently, in your data the CAS DPOL instrument undersizes large droplets! (40 μm in diameter appear as 35 μm drops?). In case your effective diameter droplets of 26 μm would have been 30 μm droplets in reality, you are underestimating LWC by 50% for these droplet sizes: : : Likewise, the King probe is underestimating LWC for other reasons as mentioned above.

A: In Figure 6d (available below), we show size distributions and CWC distributions of both cloud probes for large effective diameters. Even for the large effective diameter, only few drops > 40 μm were measured by both instruments CDP and CAS-DPOL. The difference between CAS-DPOL and the CDP above 35um is less than a factor of two, though at very low concentrations. Thus, the difference/error in diameter of these large particles does not contribute significantly to the calculated effective radius. The large droplets contribute an order of magnitude less to the total water content than droplets in the size range around 25 um. Changing the calibration would change the result of the intercomparison of CWC insignificantly. According to Strapp et al. 2003, hotwire probes in general compared well to other LWC instruments up to MVDs of about 32 um, which comprises the largest fraction of the CWC measured during ACRIDICON-CHUVA. To sum up, the disagreement between the CWC of hot wire and the cloud probes represent an upper limit of the cloud probe inaccuracy.

[Figure]

Figure 6. (left) Mean cloud droplet concentration (solid lines) and (right) cloud water content as a function of droplet diameter in the left and right panels, respectively, for a) 5 μm < $r_e$ < 6 μm; b) 8 μm < $r_e$ < 9 μm; c) 11 μm < $r_e$ < 12 μm; d) 12 μm < $r_e$ < 13 μm. The probes are identified by colors as shown at the top of the panels. The error bars indicate the uncertainty range of mean cloud droplet concentration and cloud water content values as a function of droplet diameter.

Line 386-394: What is the uncertainty in $N_0$ and $k$ calculation and finally the uncertainty in equation (2) calculated droplet number (calculated each second) when averaging $CCN_2$ per time step normalized by FA (with two other averages of $mCCN_1$ per time step and $TmCCN_1$ average of all $mCCN_1$ time steps or may be even all $CCN_1$ data)?

A: We have changed the manuscript to address these questions as follow:

" The calculated $N_{CCN}(S)$ errors for these flight segments are a function of the measured particle number (i.e. 10% of $N_{CCN}(S)$ for large concentrations and the mean of the error is around 20% of $N_{CCN}(S)$). The estimated standard error (STDE) for the $N_0$ and $k$ parameters and CCN estimates were calculated (as described in Appendix B) for each flight segment and are shown in Table 2. The table shows that the STDE associated to the Twomey's equation fit is about 5% for the $N_0$ and $k$ parameters. The changes in the air mass assumed to correct the $CCN_2$ for $FA$ during the flight segments were up to 24 % for all flights. As long as the cloud segment compared with this data are not at exactly the same location as the measurements was performed, the mean (i.e. $TmCCN_1$) is a good measure for this comparison. The standard error was used for the error propagation calculations and the resulting error in $N_{CCN}(S)$ is 15 % of $N_{CCN}(S)$ estimates on average. The resulting error of $N_0$ ($k$ slope) was also calculated and is 23 % (20 %) of $N_0$ ($k$) values on average, associated to the Twomey's equation fit and the $N_{CCN}(S)$ error."

"The uncertainties regarding the $S_{max}$, $N_{dCCN}$ and $N_{dT}$ estimates for measurements at cloud base with both probes (CCP-CDP and CAS-DPOL) are on average about 22, 20 and 38 % for all flights, respectively (the uncertainty method adopted for these theoretical estimates are available at Appendix A). The $W_b$ uncertainty of 0.3 m s$^{-1}$ impacts on average for about 65% (60 %) on $N_{dT}$ ($S_{max}$) uncertainty, and the uncertainty from the estimated $S_{max}$ contributes for most of $N_{dCCN}$ uncertainty (~70% on average)."

Line 400: equation (3) does not pretend Smax depending on $N_{CCN}(S)$. Please detail how $N_d$ can be used to achieve a closure for $N_{dCCN}$ estimate.

A: Ok. The following text was added:
" The value of $S_{max}$ at cloud base can be estimated from Eq. 3 based on the vertical velocity at cloud base and $N_d$ values measured with the cloud probes CCP-CDP and CAS-DPOL (*Ncdp* and *Ncas,* respectively). Therefore, the estimated $S_{max}$ near cloud base can be used in Eq. 1, producing the $N_{dCCN}$ estimates to achieve a closure for $N_d$ measurements at cloud base."

Section 5.2.1.: Gray solid/dashed lines difficult to see in Figs 11 and 12!

A: Ok. We made it thicker.

Fig 11a & 11c show very weak overlap of $N_{dCCN}$ and $N_{dT}$ including both uncertainties. In addition, real $N_d$ measurements can be considerably outside $N_{dT}$ uncertainties and particularly outside the overlap region. Why? What is the value of this study when measurements are not better matching the calculations with their uncertainties? Are the already large uncertainties still underestimated? Are measurements and calculations comparable in their complexity of the respective environments? I don't think so….

A: $N_{dT}$ show a great overlap at Figures 11a and 11c, whilst $N_{dCCN}$ indeed have a weaker overlap (see Figures 11a and 11c below).
The following text was added to the manuscript:
"Both values of *Ncas* and *Ncdp* are within the range of the theoretical expectation of $N_{dT}$ and $N_{dCCN}$, except for occasional deviations at the extreme percentiles. For example, the maximum $N_{dT}$ versus maximum $N_d$ are outside the error interval for $N_d$. This is so because extreme percentiles are much more prone to random variations than the middle range, such as the median.
The lines of $N_{dT}$ mostly agreed quite well with the lines of $N_d$ with only small deviations. The $N_{dCCN}$ mostly underestimates $N_d$ by down to a factor of 0.5 for reasons that we could not identify. Entrainment is not a likely cause, because it would dilute $N_d$ and thus incur $N_{dCCN}$ to be biased positively with respect to $N_d$.

It appears that measuring *S* in clouds is still a great challenge, even indirectly by using Eq. 3. Remarkably, Eq. 2 (Twomey, 1959), which avoids an explicit usage of *S*, still performs better when limited within the observed bounds of $W_b$ and *k* within the cloud."

[Figure]

[Figure]

Fig 12: Color difference of $N_d$ curves (red) and $N_d$ in legend (blue). Can you also show results for AC13 and AC16? Fig 12b and 12c as well as 11c show $N_d$ that significantly exceed $N_{dT}$ for higher $W_b$. Explanation? The problem stems basically from $N_{CCN2}$ calculation?

A: The response to the previous comment addresses this comment too.

Line 513: Change 10% to at least 15% if not 20% (AC14!).

A: Ok. Changed.

Line 513-517: and a factor of 1.5 for other cases AC11 and again AC17. Solely AC 13 and AC16 data points ok. Therefore I don't agree with that improper statement.
Line 566-567: I would call a factor of 2 in $N_{dCCN}*$ to $N_d*$ comparison a pretty bad result rather than a good agreement.

A: The following text was added to the manuscript regarding these comments:

[revised manuscript text omitted]

a) FLIGHT AC11   TIME: 17:52(56 s)   CCN=2927.4 · $S^{1.137}$

1230

b) FLIGHT AC13   TIME: 16:50(72 s)   CCN=4145.4 · $S^{0.922}$

c) FLIGHT AC14   TIME: 15:06(184 s)   CCN=1509.7 · S^0.973

d) FLIGHT AC16   TIME: 20:10(58 s)   CCN=1966.2 · S^0.672

1235

[Figure]

**Figure 11a-f.** $N_{dCCN}$, $S$, $N_{dT}$ and $N_d$ values are presented as a function of the cloud base updrafts ($W_b$). This plot is based on the 'probability matching method' (PMM), using same percentiles for $W_b$ and $N_d$ ($N_{dCCN}$ or $N_{dT}$). The values of $N_{dCCN}$, $N_{dT}$ and $N_d$ are shown the left y-axis, those of $S$ on the right y-axis. The black dashed lines are the $N_{dT}$ uncertainties. The gray solid (dashed) lines are the $N_{dCCN}$ values (uncertainties). The effective updraft $W_b$* for each flight segment is shown by the cyan line. The data are based on the CAS-DPOL probe. The time, period of measurements (sample size in seconds), and $N_{CCN}(S)$ equation are shown on the top of the figures.

[Figure]

a) FLIGHT AC11   TIME: 17:52(61 s)   CCN=2927.4 · $S^{1.137}$

[Figure]

b) FLIGHT AC14   TIME: 15:06(111 s)   CCN=1509.7 · $S^{0.973}$

[Figure]

[Figure]

**Figure 12a-d. Same as Figure 11 for the CCP-CDP probe. No data were available for flight AC16. The CCP-CDP malfunctioned in flight AC13 during the cloud base measurements.**

1250

1255

[Figure]

**Figure 13. a)** $N_d^*$ versus $N_{dT}^*$ calculated with $W_b^*$ from cloud base data shown in Figures 11-12. The CAS-DPOL values are indicated by plus symbols (+) and the CCP-CDP values are indicate by circles (o). The colors indicate each flight segment (legend in the right side of the plot). Error bars indicates the uncertainties of variables estimates. Lines show the 1:1 and 1:2 relationships between $N_{dT}^*$ versus $N_d^*$ for each probe; **b)** Same for $N_d^*$ versus $N_{dCCN}^*$.

[Figure]

1265   **Figure 14 a). Mean volume drop mass ($M_v$) versus liquid water content from the CCP-CDP measurements for adiabatic fraction greater than 0.25 ($LWC_a$). Values are shown with different colors labeled as a function of height in kilometers above sea level (indicated by the colorbar on the right side of the graphic). The slope of the linear equation is the estimated $N_a$ (i.e., 1496 cm$^{-3}$); b) $M_v$ versus $r_e$ as a function of height in kilometers above sea level (indicated by the colorbar on the right side of the graphic).**

1270

[Figure]

**Figure 15.** $N_d$* versus $N_a$ measured with CAS-DPOL and CCP-CDP (indicated on the top of panels) for profile flights during the ACRIDICON-CHUVA campaign. The color of the dots is associated with the flight number shown at the right side of the panels. Error bars indicates the uncertainties of variables estimates. The linear regression equation and the correlation coefficient $R$ are shown in the top of each panel.

1275

---

## Author Response (AR2)

**Co-Editor Decision: Reconsider after minor revisions (Editor review)** (03 May 2017) by Ulrich Schumann

We appreciate and thank the Editor for his considerable effort and care in the complex review of this paper.

Comments to the Author:

Thank you for submitting the paper. I am glad to see that the ACRIDICON measurements are scientifically useful.

Sorry for the long time it took to come to a decision on this paper.

The ACPD version of your paper got mixed ratings (poor to good) ; one reviewers suggested rejection, the two other asked for major revisions.

Unfortunately, none of these reviewers was ready to review the revised paper.

That caused the large time delay.

I got a short review from a new reviewer (reviewer # 4), recommending acceptance subject to technical corrections.

I also looked through the paper, again, myself.

In addition, I informally and confidentially discussed my tentative decision with one of the previous reviewers.

I also addressed several times a second previous reviewer but got no response.

Based on this limited input, I decide as follows:

The paper is accepted subject to minor revision to be checked by me, the co-editor, including reactions to the reviewer #4 and the following recommendations:

Abstract, line 43: please delete "more confidently and" because the paper also shows up various difficulties to verify the relationships.

A: Ok. Changed

Lines 14 to 152: Please avoid the term" it is the first study that" because this claim was not part of the ACPD paper and, hence, was not available for public discussion, and I am not in the position myself to judge that this is indeed the first study.

Instead start the point "a" as follows." It validates the …" and similar for points b and c.

A: Ok. Changed

Line 251: I know of a recent Nature paper by Moore et al. (2017; doi: 10.1038/nature21420) that could be cited in connection with ACCESS-II by now. Please check for available references for ECLIF and DACCIAW and provide explanations to all abbreviations used.

A: Ok. Changed. The explanations of abbreviations are corrected now.

Line 268: "For a long sequence of measurements at cloud base (> 20 s) these uncertainties become negligible." Please add an explanation for why the uncertainties get smaller for a longer sequence. I do not see that the velocity measurement errors are purely random. Hence, I have doubts that this sentence is strictly correct. Unless you have good arguments, I suggest deleting this sentence.
A: Ok. Deleted.

Line 257 and 284: write 125°C and 0°C without blank before °C.

A: Ok. Changed

Line 507: why is there a blank after "measured particle number". Is there a symbol missing? Or should the blank be deleted?

A: The blank is deleted.

Line 606: I missed to understand where you introduced "novel" parametrization: I thought the parameterizations are as suggested earlier. Hence, I ask to skip "novel" or to provide a good explanation to me.
Line 607, for reasons given above, I ask to skip "It is the first time that these" and adjust the remainder of the sentence accordingly (there are neither new nor old parametrizations). An alternative could be to call the "novel" ones "recent".
A: Ok. Change  "novel" to "recent".

Line 633: The last sentence should be changed in consistency with the changes recommended for the abstract. I am still not sure that you got much more confidence in the usefulness of the parameterization but got more evidence to quantify the application range and the uncertainties of the models. You may adjust your wording to this impression, or change otherwise, or omit this sentence.
A: We omit this sentence.

Line 1128, Caption Figure 5. I assume the reference to Figure 3 needs to be replaced by reference to Figure 4.

A: Ok. Thanks.

**Referee number 4 comments.**

Braga et al., use airborne measurements during the ARCRIDOICON-CHUVA campaign to verify and further the applicability of the recently proposed satellite retrieval algorithm for CCNC below cloud base. Their validation involves an in-depth cloud probe comparison. As well as a comparison between observed cloud properties and previously proposed cloud parameterizations for predicting cloud droplet number concentration based on updraft velocity and CCNC as a function of supersaturation.

Specific comments:

Line 216: "article losses" should be, "particle losses"

A: Ok. Changed.

Line 217: consider adding a comma after the parentheses so that it is "issue), the"

A: Ok. Changed.

Line 217: make droplets singular, it should read "cloud droplet concentration"

A: Ok. Changed.

Line 248: replace "the" to "in", the sentence should be "up to 20 µm in size"

A: Ok. Changed.

Line 292: Consider moving "maximum" before uncertainty so that it read, "an overall maximum uncertainty of 16%"

A: Ok. Changed.

Line 298: Please specify the probe used for the DWC to make the sentence a bit clearer

A: Ok. Changed.

Line 342: The word "data", is plural, please change "is" to "are"

A: Ok. Changed.

Line 342: please remove the extra "used" so that the sentence reads "The data are the same as used for the"

A: Ok. Changed.

Line 533: There should be a space between "Ndt" and "and"

A: Ok. Changed.

Line 575: The texts suggests that dividing by a factor 1.3 for this case helps brings the values closer together. However, in this case not dividing by 1.3 would actually keep the values closer. Please clarify this or note that this is an exception since the division works well for the majority of the remaining flights.

A: We don't agree. The decrease of 30 % is essential to bring the values closer in every case. Actually, it is not a factor of ~1.3 but 30 % (we have corrected it in the text now), i.e., a factor

of 0.7. In this case, without applying this factor the difference is 289 cm$^{-3}$, while applying the factor the difference is 160 cm$^{-3}$ (almost the double).

Line 615: Consider replacing "was favorably compared" with "compared well"

A: Ok. Changed.

Line 627: Remove the period after Nd

A: Ok. Changed.

Line 628: Consider changing "in cloud" to "in-cloud"

A: Ok. Changed.

Line 1163: Please remove the "(left)" and "(right)" as they are described at the end of the sentence.

A: Ok. Changed.

Figure 7: Please make the Cumulative probability axis the same bold as the other axes in the figure

A: Ok. Changed.

Figure 15: Are the correlation coefficients for both probes .97? In the text it says greater than .90

A: Ok. We have changed in the text to 0.97.